# Oja's Algorithm for Streaming Sparse PCA

**Syamantak Kumar**[1]     **Purnamrita Sarkar**[2]

[1]Department of Computer Science, UT Austin
[2]Department of Statistics and Data Sciences, UT Austin
syamantak@utexas.edu, purna.sarkar@austin.utexas.edu

## Abstract

Oja's algorithm for Streaming Principal Component Analysis (PCA) for $n$ data-points in a $d$ dimensional space achieves the same sin-squared error $O(r_{\mathsf{eff}}/n)$ as the offline algorithm in $O(d)$ space and $O(nd)$ time and a single pass through the datapoints. Here $r_{\mathsf{eff}}$ is the effective rank (ratio of the trace and the principal eigenvalue of the population covariance matrix $\Sigma$). Under this computational budget, we consider the problem of sparse PCA, where the principal eigenvector of $\Sigma$ is $s$-sparse, and $r_{\mathsf{eff}}$ can be large. In this setting, to our knowledge, *there are no known single-pass algorithms* that achieve the minimax error bound in $O(d)$ space and $O(nd)$ time without either requiring strong initialization conditions or assuming further structure (e.g., spiked) of the covariance matrix. We show that a simple single-pass procedure that thresholds the output of Oja's algorithm (the Oja vector) can achieve the minimax error bound under some regularity conditions in $O(d)$ space and $O(nd)$ time. We present a nontrivial and novel analysis of the entries of the unnormalized Oja vector, which involves the projection of a product of independent random matrices on a random initial vector. This is completely different from previous analyses of Oja's algorithm and matrix products, which have been done when the $r_{\mathsf{eff}}$ is bounded.

## 1  Introduction

Principal Component Analysis (PCA) [Pea01, Jol03] is a classical statistical method for data analysis and visualization. Given a dataset $\{X_i\}_{i=1,\ldots,n}$ where $X_i \in \mathbb{R}^d$, sampled independently from a distribution with mean zero and covariance matrix $\Sigma$, the goal in PCA is to find the directions that explain most of the variance in the data. It is well known [Wed72, JJK+16, Ver10] that the leading eigenvector, $\hat{v}$, of the empirical covariance matrix, $\hat{\Sigma}$, provides an optimal error rate under suitable tail conditions on the datapoints.

Computing $\hat{v}$ can be inefficient for large sample sizes, $n$, and dimensions $d$. Oja's algorithm [Oja82a] offers a comparable error rate in $O(nd)$ time and $O(d)$ space. Going back to the Canadian psychologist Donald Hebb's research [Heb49], it has attracted a lot of attention in theoretical Statistics and Computer Science communities [JJK+16, AZL17, CYWZ18, YHW18, HW19a, HNW21, MP22, LSW21, Mon22, HNWW21]. In these works, the error metric is the $\sin^2$ error between the estimated vector and the principal eigenvector of $\Sigma$ (true population eigenvector $v_1$). Notably, [JJK+16], [AZL17], and [HNW21] establish that Oja's algorithm achieves the same $O(r_{\mathsf{eff}}/n)$ sin-squared error as the *offline algorithm* that estimates the top eigenvector of the empirical covariance matrix.

However, when the effective rank, $r_{\mathsf{eff}}$, of $\Sigma$ (defined as $\operatorname{Tr}(\Sigma)/\|\Sigma\|$) is large, PCA has been shown to be inconsistent [Pau07, JM09, JL09]. This setting comes up in sparse PCA problems, when $v_1$ is $s$-sparse (i.e. has only $s$ nonzero entries). Let $\|.\|_0$ denote the $l_0$ norm, i.e, the count of non-zero vector entries. Then, sparse PCA can be formally framed as the optimization problem:

$$\widehat{v}_{\mathsf{sparse}} := \arg\max_{w \in \mathbb{R}^d} \sum_i \left(X_i^T w\right)^2, \text{ under constraints } \|w\|_2 = 1, \ \|w\|_0 = s \tag{1}$$

38th Conference on Neural Information Processing Systems (NeurIPS 2024).

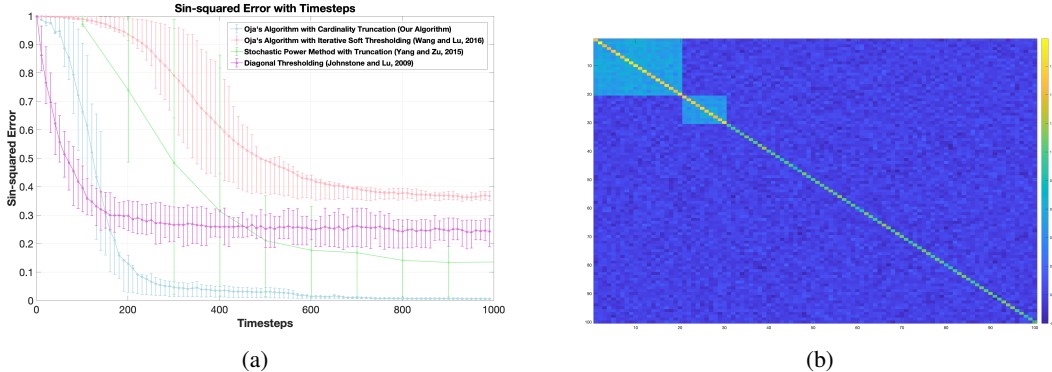

(a)                                                      (b)

Figure 1: Comparison of Sparse PCA algorithms for identifying leading eigenvector, $v_1$, operating in $O(d)$ space and $O(nd)$ time with population covariance matrix specified in [QLR19], Section 5.1. Figure (a) plots [JL09] (Purple), [YX15] (Black), [WL16] (Orange) and our proposed Algorithm 2 (Blue) for $n = d = 1000$, with error bars over 100 random runs. Figure (b) shows an image of the covariance matrix with $n = d = 100$.

In general, without further assumptions, Problem (1) is non-convex and NP-hard [MWA06], as it reduces to subset selection in ordinary least squares regression.

[VL12, CMW13] showed a $O\left(\sigma_*^2 s \log(d)/n\right)$ minimax lower bound for the $\sin^2$ error $1 - (v_1^T \widehat{v})^2$, where $\sigma_*^2 := \frac{\lambda_1 \lambda_2}{(\lambda_1 - \lambda_2)^2}$. Here $\lambda_1 > \lambda_2 \geq \ldots \lambda_d$ are the eigenvalues of $\Sigma$. Extensive research has been conducted on optimal offline algorithms for sparse PCA, some of which are convex relaxation-based [BR13, dBEG08, VCLR13, STL07, ZX18, DMMW17, AW08, Ma13, CMW13]. Others involve iterated thresholding [JNRS10, Ma13, YZ13], where a truncated power-method is analyzed along to achieve sparsity. For brevity, we only describe algorithms that fit within the computation budget in consideration, i.e., $O(nd)$ time, $O(d)$ space. For a detailed comparison, see Table 1 and Appendix Section A.1.

**Support recovery algorithms in $O(nd)$ time, $O(d)$ space:** Consider the spiked covariance model

$$\Sigma = \sum_{i \in [r]} \nu_i v_i v_i^T + I_d \tag{2}$$

where $I_d$ is the identity matrix, $\nu_i > 0$, and $v_i$ are sparse. For the general case, we only assume $v_1$ is $s$-sparse. When $r = 1$, $\Sigma_{ii}$ are the largest for $i \in S$. Diagonal thresholding essentially estimates $\Sigma_{ii}$ within our computational budget and uses thresholding to recover the support [JL09, AW08]. However, as we will show, without knowing the support sizes in each eigenvector and the number of spikes, this algorithm can fail, even in a spiked setting with $r > 1$. Also, for $r = 1$, [BPP18] show how to adapt a black-box algorithm for sparse linear regression for support recovery.

**Sparse PCA algorithms in $O(nd)$ time, $O(d)$ space:** The streaming sparse PCA algorithms proposed by [YX15] and [WL16] require an initialization $u_0$ with a sufficiently large $|u_0^T v_1| = \Omega(1)$ (local convergence), which can be hard to find for large $d$ and a general $\Sigma$. See Table 1 for details.

In light of this lack of $O(nd)$ time, $O(d)$ space globally convergent algorithms for sparse PCA, we ask the following question in this work:

**Goal:** *Is there a single-pass algorithm that, under a general $\Sigma$ with $s$-sparse $v_1$, outputs $\hat{v}$ achieving the minimax $\sin^2$ error $(1 - \langle \widehat{v}, v_1 \rangle^2)$ with $O(d)$ space, $O(nd)$ time, without a strong initialization?*

We provide a surprisingly simple answer to the above question:

**Theorem 1.1** (Informal). *For a suitable range of the effective rank $r_{\text{eff}}$ and the ratio $\lambda_1/\lambda_2$, there exists a single pass algorithm $\mathcal{A}$ that recovers the support of $v_1$ using Oja's algorithm, operates under $O(d)$ space, $O(nd)$ time and returns $\hat{v}$ with the minimax optimal $\sin^2$ error, $O\left(\sigma_*^2 s \log(d)/n\right)$, for a general covariance matrix.*

**Our contributions:**

1. **Support recovery:** We show, for a *general* $\Sigma$ with the only constraint of a $s$-sparse $v_0$ that the top $k$ entries of the Oja vector in magnitude include the true support with high probability. The Oja vector is initialized by a random unit vector.

2. **Sparse PCA:** We use the recovered support to achieve a minimax optimal sparse PCA algorithm.

3. **Entrywise analysis:** Our analysis is nontrivial and novel because it deviates from all existing analyses of matrix products and streaming PCA [HNWTW20, HW19b, LSW21, Lia23] which require $\|X_i\|^2/\lambda_1$ or $r_{\text{eff}}$ to be bounded to obtain the $O(1/n)$ $\sin^2$ error rate.

| Paper(s) | $\lambda_1/\lambda_2$ | $\Sigma$ | Global conv.? | Space | Time | $\sin^2$ error |
|---|---|---|---|---|---|---|
| Johnstone and Lu [JL09] | $1 + o(1)$ | Spiked | Y | $O(d)$ | $O(nd)$ | $o(1)$ |
| SDP-based [VCLR13] [dBEG08] | $1 + o(1)$ | General | Y | $O(d^2)$ | $O(n^\omega + d^\omega)$ | $O\left(\dfrac{s^2 \log(d)}{n}\right)$ |
| Shen et al. [SSM13] | $\Omega(d^\epsilon), \epsilon > 0$ | General | Y | $O(d^2)$ | $O(nd^2)$ | $o(1)$ |
| Ma, Cai et al. [Ma13] [CMW13] | $1 + \Omega(1)$ | Spiked | Y | $O(d^2)$ | $O(nd^2)$ | $O\left(\dfrac{s \log(d)}{n}\right)$ |
| Yuan and Zhang [YZ13] | $1 + \Omega(1)$ | General | N | $O(d^2)$ | $O(nd^2)$ | $O\left(\dfrac{s \log(d)}{n}\right)$ |
| Yang and Xu [YX15] | $1 + \Omega(1)$ | Spiked | N | $O(d)$ | $O(nd)$ | $O\left(\dfrac{s \log(d)}{n}\right)$ |
| Wang and Lu [WL16] | $1 + \Omega(1)$ | Spiked | N | $O(d)$ | $O(nd)$ | $o(1)$ |
| Oja's Algorithm [JJK+16] | $1 + o(1)$ | General | Y | $O(d)$ | $O(nd)$ | $O\left(\dfrac{r_{\text{eff}}}{n}\right)$ |
| Deshp et al. [DM+16] | $1 + o(1)$ | Spiked | Y | $O(d^2)$ | $O(nd^2)$ | $O\left(\dfrac{s^2 \log(d)}{n}\right)$ |
| Qiu et al.(Cor. 2) [QLR19] | $1 + o(1)$ | General | Y | $O(d^2)$ | $\Omega(nd^2)$ [1] | $O\left(\dfrac{s^2 \log(d)}{n}\right)$ |
| Qiu et al. (Th. 4) [QLR19] | $1 + o(1)$ | General | Y | $O(d^2)$ | $O(nd^2)$ | $O\left(\dfrac{d^2 \log(d)}{\sqrt{n}}\right)$ |
| Gataric et al. (Th. 2) [GWS20] | $1 + o(1)$ [2] | Spiked | Y | $O(d^2)$ | $O(nd^2)$ [3] | $O\left(\dfrac{s \log(d)}{n}\right)$ |
| Our work | $1 + \Omega(1)$ | General | Y | $O(d)$ | $O(nd)$ | $O\left(\dfrac{s \log(d)}{n}\right)$ |

Table 1: Comparison of sparse PCA algorithms for estimating $v_1$, based on various parameters. We require Assumptions 1 and 2. The other algorithms may be valid under weaker assumptions. For ease of comparison, we fix $\frac{\lambda_1}{\lambda_2} = 1 + \Omega(1)$ and $\frac{r_{\text{eff}} \log(n)}{n} = O(1)$ for our results in this table.

In Figure 1b, for a simple spiked model with $r = 2$, we show the relative performances of all $O(nd)$ time and $O(d)$ algorithms in Table 1. Our thresholded and renormalized Oja algorithm outperforms all other algorithms operating under the same computational budget. The diagonal thresholding algorithm ( [JL09]), which is successful for the special case of $r = 1$, has a large error in the general case.

We now present an outline of our paper. We start by describing the problem setup and assumptions in Section 2. Then we present our main results in Section 3, which includes our results for Support

---

[1] The authors do not state the runtime explicitly. The algorithm, as stated, requires at least $\Omega(nd^2)$ computation.

[2] The authors require $\lambda_1/\lambda_2 \geq 1 + O(\sqrt{s^3 \log d/n})$

[3] When there are $m$ spikes, Thm 2 of [GWS20] requires $A = \Omega(\frac{\nu_m^2}{\nu_1^2} d^2 \log(d))$. When $\nu_1$ and $\nu_m$ are the same order, storing the empirical covariance matrix is computationally more efficient.

Recovery (Section 3.1), Sparse PCA (Section 3.3) and Entrywise Deviation bounds (Section 3.5) for the Oja vector. Finally, we provide a sketch of the proof along with the techniques used in Section 4.

## 2    Problem setup and preliminaries

**Notation**. We use $\mathbb{E}\left[.\right]$ to denote expectation and $[n]$ for $\{1,\dots,n\}$. The matrix multiplication constant is denoted as $\omega \approx 2.372$. $X \perp\!\!\!\perp Y$ represents statistical independence between random variables $X$ and $Y$. The $\ell^2$ norm for vectors and operator norm for matrices is $\|.\|_2$, the count of nonzero vector elements ($\ell^0$ norm) is $\|.\|_0$, and the Frobenius norm for matrices is $\|.\|_F$. For $v \in \mathbb{R}^d, R \subseteq [d], \lfloor v \rfloor_R \in \mathbb{R}^d$ is the truncated vector with entries outside $R$ set to 0. $I_d \in \mathbb{R}^{d \times d}$ is the identity matrix, with $i^{th}$ column $e_i \in \mathbb{R}^{d \times 1}$. For any set $T \subseteq [d]$, $I_T \in \mathbb{R}^{d \times d}$ is defined as $I_T\left(i,j\right) = \mathbb{1}(i, j \in T)\mathbb{1}(i = j)$, where $\mathbb{1}(.)$ is the indicator random variable. $\langle A, B \rangle := \text{Tr}(A^T B)$ represents the matrix inner product. $\widetilde{O}$ and $\widetilde{\Omega}$ represent order notations with logarithmic factors. We start by defining subgaussianity for multivariate distributions.

**Definition 2.1.** *A random mean-zero vector $X \in \mathbb{R}^d$ with covariance matrix $\Sigma$ is a $\sigma-$subgaussian random vector ($\sigma > 0$) if for all vectors $v \in \mathbb{R}^d$, we have $\mathbb{E}\left[\exp\left(v^T X\right)\right] \leq \exp\left(\sigma^2 v^T \Sigma v/2\right)$. Equivalently, $\exists\, L > 0$, such that $\forall p \geq 2$, $\left(\mathbb{E}\left[|v^T X|^p\right]\right)^{\frac{1}{p}} \leq L\sigma\sqrt{p}\sqrt{v^T \Sigma v}$.* [4]

This definition of subgaussianity has been used in contemporary works on PCA and covariance estimation (See for example [MZ20, JLT20, DKPP23] and Theorem 4.7.1 in [Ver18]). We operate under the following two assumptions, unless otherwise specified,

**Assumption 1** (Subgaussianity). *$\{X_i\}_{i \in [n]}$ are of independent and identically distributed $\sigma$-subgaussian vectors in $\mathbb{R}^d$ with covariance matrix $\Sigma := \mathbb{E}\left[X_i X_i^T\right]$.*

We denote the eigenvectors of $\Sigma$ as $v_1, v_2, \cdots v_d$ and the corresponding eigenvalues as $\lambda_1 > \lambda_2 \geq \cdots \lambda_d$. Define $V_\perp := [v_2, v_3, \cdots v_d] \in \mathbb{R}^{d \times (d-1)}$ and $\Lambda_2 \in \mathbb{R}^{(d-1) \times (d-1)} = \text{diag}\left(\lambda_2, \lambda_3, \cdots \lambda_d\right)$.

**Assumption 2** (Sparsity and Spectral gap). *We assume that $\max\left\{1, \frac{\lambda_2}{\lambda_1 - \lambda_2}\right\} \frac{\text{Tr}(\Lambda_2)}{\lambda_1 - \lambda_2} \leq \frac{cn}{\log(n)}$ and $\frac{\lambda_1}{\lambda_1 - \lambda_2} \leq c\sqrt{\frac{n}{\log^2(n)}}$ for an absolute constant $c > 0$. The leading eigenvector, $v_1$, satisfies $\|v\|_0 \leq s$ with support set $S := \{i : v_1\left(i\right) \neq 0\}$.*

**Remark 2.2.** *We note that Assumption 2 allows for $r_{\text{eff}}$ to be as large as $d$, given a sufficient eigengap. This can be observed by setting $\lambda_1 = \lambda_2(1 + g_n)$ for some $g_n > 0$. Note that $\frac{\text{Tr}(\Lambda_2)}{\lambda_1 - \lambda_2} \leq \min\left(\frac{1 + g_n}{g_n} r_{\text{eff}}, \frac{1}{g_n} d\right)$. If $g_n \leq 1$, then,*

$$\max\left\{1, \frac{1}{g_n}\right\} \frac{\text{Tr}(\Lambda_2)}{\lambda_1 - \lambda_2} \leq \frac{1}{g_n}\min\left(\frac{1 + g_n}{g_n} r_{\text{eff}}, \frac{1}{g_n} d\right) = 2\frac{1}{g_n}\min\left(\frac{1 + g_n}{2g_n} r_{\text{eff}}, \frac{1}{2g_n} d\right) \leq 2r_{\text{eff}}/g_n^2$$

*If $g_n \gg 1$, then,*

$$\max\left\{1, \frac{1}{g_n}\right\} \frac{\text{Tr}(\Lambda_2)}{\lambda_1 - \lambda_2} \leq \frac{\text{Tr}(\Lambda_2)}{\lambda_1 - \lambda_2} \leq \frac{d}{g_n}$$

*therefore, in both cases, as long as $d \leq \frac{ng_n}{\log n}$, $r_{\text{eff}}$ can be as large as $d$, while allowing for Assumption 2 to hold.*

**Oja's algorithm with constant learning rate.** With a constant learning rate, $\eta$, and initial vector, $u_0$, Oja's algorithm [Oja82b], denoted as $\text{Oja}\left(\{X_t\}_{t \in [n]}, \eta, u_0\right)$, performs the updates, $u_t \leftarrow (I + \eta X_t X_t^T)u_{t-1}$, $u_t \leftarrow \frac{u_t}{\|u_t\|_2}$. For convenience of analysis, we also define $\forall t \in [n]$,

$$B_t := \left(I + \eta X_t X_t^T\right)\left(I + \eta X_{t-1} X_{t-1}^T\right) \cdots \left(I + \eta X_1 X_1^T\right), \ B_0 = I \tag{3}$$

---

[4]The results developed in this work follow if instead of subgaussianity, the moment bound holds $\forall\, p \leq 8$.

# 3 Main results

We present our main contributions in two stages. Firstly, in Section 3.1, we demonstrate that with an upper bound on the support size, the top elements of the Oja vector include the support with constant probability, which can be enhanced using a boosting procedure (SuccessBoost) described in Section 3.4. Secondly, in Section 3.3, we use the support to extract the eigenvector and provide a high-probability $\sin^2$ error guarantee. Section 3.5 details our results on bounding the entrywise deviation of the Oja vector, which are crucial to our proofs and of independent interest. Detailed proofs are in the Appendix, Sections A.4 and A.5, with the learning rate, $\eta$, specified in Lemma A.2.4.

## 3.1 Support recovery

---
**Algorithm 1** OjaSupportRecovery $\left( \{X_i\}_{i \in [n]}, k, \eta \right)$

---
1: **Input** : Dataset $\{X_i\}_{i \in [n]}$, Cardinality parameter $k \geq s$, learning rate $\eta > 0$
2: $u_0 \sim \mathcal{N}(0, I)$
3: $\hat{v} \leftarrow \mathsf{Oja}\left( \{X_i\}_{i \in [n]}, \eta, y_0 \right)$
4: $\widehat{S} \leftarrow$ Indices of $k$ largest values of $|\hat{v}|$
5: **return** $\widehat{S}$

---

Algorithm 1 provides an estimate, $\widehat{S}$, of the true support set, $S$. It computes the Oja vector and returns the set of indices corresponding to its $k$ largest entries in absolute value. Our key result in Lemma 3.1 discusses the recovery of the support set, $S$, for any $k \geq s$, without requiring exact knowledge of the sparsity parameter $s$. Using Algorithm 1, it provides a set $\widehat{S} \supseteq S$ with probability at least 0.9.

**Lemma 3.1** (s-Agnostic Recovery). *Under Assumptions 1,2, for* $\min_i |v_1(i)| = \widetilde{\Omega}\left( \frac{\lambda_1}{\lambda_1 - \lambda_2} \left( \frac{d}{n^2} \right)^{\frac{1}{4}} \right)$, $\widehat{S} \leftarrow$ OjaSupportRecovery $\left( \{X_i\}_{i \in [n]}, k, \eta := \frac{3 \log(n)}{n(\lambda_1 - \lambda_2)} \right)$ *with* $k \geq s$ *satisfies,* $\mathbb{P}\left( S \subseteq \widehat{S} \right) \geq 0.9$.

If $k = s$, i.e, the size of the support is exactly known, then we can improve the result of Lemma 3.1 to obtain an estimator, $\widehat{S}$, of the support set with high probability. Theorem 3.2 provides the corresponding guarantees. The SuccessBoost algorithm uses geometric aggregation on subsets returned from Algorithm 1 run on $\log(1/\delta)$ disjoint subsets of the data and is described in Section 3.4.

**Theorem 3.2** (High probability support recovery). *Let Assumptions 1, 2 hold. For dataset* $\mathcal{D} := \{X_i\}_{i \in [n]}$, *let* $\mathcal{A}$ *be the randomized algorithm which computes* $\widehat{S} \leftarrow$ OjaSupportRecovery $\left( \{X_i\}_{i \in [n]}, k, \eta \right)$, *where* $\eta := \frac{3 \log(n)}{n(\lambda_1 - \lambda_2)}$ *and* $k = s$. *Then, for* $\delta \in (0, 1)$, $\min_i |v_1(i)| = \widetilde{\Omega}\left( \frac{\lambda_1}{\lambda_1 - \lambda_2} \left( \frac{d}{n^2} \right)^{\frac{1}{4}} \right)$, $\tilde{S} \leftarrow$ SuccessBoost $\left( \{X_i\}_{i \in [n]}, \mathcal{A}, \delta \right)$ *satisfies,*

$$\mathbb{P}\left( \tilde{S} = S \right) \geq 1 - \delta$$

For comparison, existing support recovery algorithms for general $\Sigma$ are known for convex-relaxation-based algorithms like the SDP-based algorithm of [LV15]. These require a much larger computational budget than ours. In Section 3.3, we show how to use the $s$-agnostic support recovery in Lemma 3.1 to perform Sparse PCA and obtain a $\sin^2$ error guarantee, where the final high probability error bound is obtained using a similar probability-boosting argument. For the learning rate, we follow the convention in related work ([BDF13, XHS+18, JJK+16, AZL17, HNWW21]) and choose the optimal value of the learning rate, which requires the knowledge of $\lambda_1 - \lambda_2$. We believe an educated guess of $\eta$ would lead to consistency at the cost of a suboptimal error bound.

## 3.2 Comparison with other support recovery algorithms

We note that (see Table 1), for the spiked model with $r = 1$ (Eq 2) [JL09] and [AW08] provide a diagonal thresholding algorithm for support recovery using $O(d)$ space and $O(nd)$ time.[5] To achieve a high-probability guarantee for the estimated support set, $\widehat{S}$, of the form $\mathbb{P}\left( \widehat{S} = S \right) \geq 1 - \delta$, they

---
[5]The algorithm proposed in [AW08] allows for a slight generalization of the spiked model in Eq 2.

require (Proposition 1, [AW08]) $n = \Omega\left(s^2 \log(d) + \log\left(\frac{1}{\delta}\right)\right)$. In comparison, Theorem 3.2 requires a larger sample size.

**Remark 3.3.** *In practice, we will not know whether $\Sigma$ is spiked or general. So, we can always augment our support recovery algorithm by taking a union of the support from the Oja vector and diagonal thresholding, still maintaining $O(nd)$ time and $O(d)$ space.*

However, diagonal thresholding only works for the spiked model with a single spike, which our results do not require. It is easy to construct a $\Sigma$ where the elements in the support of an eigenvector with a small eigenvalue have a larger magnitude than those of $v_1$ (Eq A.13). Here the diagonal thresholding method fails (see Figure 1a and Proposition 3.4). The explicit construction and the proof of Proposition 3.4 are available in the Appendix Section A.2. Figure 1 a) plots the $\sin^2$ error due to different Sparse PCA algorithms operating in $O(d)$ space and $O(nd)$ time on such a covariance matrix, $\Sigma$, which is visualized in Figure 1 a).

**Proposition 3.4** (Lower bound for diagonal thresholding). *Let Assumption 1 hold. For any diagonal-thresholding algorithm, $\mathcal{A}$, performing support recovery with sparsity parameter $s$ such that $n = \Omega\left(\sigma^4 s^2 \log(d)\right)$, there exists a covariance matrix $\Sigma$ with principal eigenvector, $v_1$, $\|v_1\|_0 = s$, such that, $\mathbb{P}\left(\left|\widehat{S}\bigcap S\right| = 0\right) \geq 1 - d^{-10}$.*

It may seem that if $r$ in Eq 2 is small, and the sparsity parameters of each $v_i$, $i \leq r$ are known, then diagonal thresholding would work. However, in general, $r$ can be as large as $d$ and the union of supports of $v_i$ can be $[d]$.

## 3.3 Sparse PCA

In this section, we describe our results for Sparse PCA, which use the support recovery guarantees developed in Section 3.1. For the results in this section, we split the dataset $D := \{X_i\}_{i\in[n]}$ into two halves and estimate the support using the first half as $\widehat{S} \leftarrow$ OjaSupportRecovery $\left(\{X_i\}_{i\in[\frac{n}{2}]}, k, \eta := \frac{3\log(n)}{n(\lambda_1 - \lambda_2)}\right)$ and input, $k \geq s$. The second half of the samples are then used to compute the estimated sparse eigenvector. Algorithm 2 describes a general procedure for Sparse PCA given access to an estimated support set, $\widehat{S}$. We start with an intuitive procedure in Theorem 3.5, which runs Oja's algorithm on the data and then uses the support to truncate the estimated eigenvector.

---

**Algorithm 2** TruncateOja $\left(\{X_i\}_{i\in[n]}, \widehat{S}, \mathcal{A}, \Theta\right)$

---

1: **Input** : Dataset $\{X_i\}_{i\in[n]}$, , estimated support set $\widehat{S} \subseteq [d]$, Algorithm $\mathcal{A}$, Parameters $\Theta$
2: $\hat{v} \leftarrow \mathcal{A}\left(\{X_i\}_{i\in[n]}, \Theta\right)$
3: $\hat{v}_{\mathsf{truncvec}} \leftarrow \dfrac{\lfloor \hat{v}\rfloor_{\hat{S}}}{\left\|\lfloor\hat{v}\rfloor_{\hat{S}}\right\|_2}$
4: **return** $\hat{v}_{\mathsf{truncvec}}$

---

**Theorem 3.5** (Vector Truncation). *Let Assumptions 1 and 2 hold and $k \geq s$. For dataset $D := \{X_i\}_{i\in[n]}$ and $w_0 \sim \mathcal{N}(0, I)$, let $\mathcal{A}$ be the randomized algorithm which computes $\hat{v}_{\mathsf{truncvec}} \leftarrow$ TruncateOja $\left(\{X_i\}_{i\in\left(\frac{n}{2}, n\right]}, \widehat{S}, \mathsf{Oja}, \{\eta, w_0\}\right)$, where $\eta := \frac{3\log(n)}{n(\lambda_1 - \lambda_2)}$. Then, for $\min_i |v_1(i)| = \widetilde{\Omega}\left(\left(\frac{d}{n^2}\right)^{\frac{1}{8}}\right)$, $\tilde{v} \leftarrow$ SuccessBoost $\left(\{X_i\}_{i\in[n]}, \mathcal{A}, d^{-10}\right)$ satisfies,*

$$\sin^2(\tilde{v}, v_1) \leq C''\left(\frac{\lambda_1}{\lambda_1 - \lambda_2}\right)^2 \frac{k\log^2(d)}{n}$$

*with probability at least $1 - d^{-10}$, where $C'' \geq 0$ is an absolute constant.*

**Remark 3.6** (Limitation). *Existing inconsistency results on PCA [JL09] provide a threshold for signal strength $(\lambda_1 - \lambda_2)/\lambda_1$, below which, the principal eigenvector of $\hat{\Sigma}$ is asymptotically orthogonal to $v_1$. We believe a similar result may hold for the Oja vector, which leads to the signal strength condition in Assumption 2.*

Note that the rate obtained in Theorem 3.5 nearly matches the minimax lower bound proved in [VL12, CMW13], up to a factor of $\frac{\lambda_2}{\lambda_1}$ and $\log(d)$ and has optimal dependence on $s$, and $n$. A limitation of Algorithm 2 is that it uses the estimated support, $\widehat{S}$, at the very end after computing the estimated eigenvector to enhance the signal by truncation. Instead, one may run Oja's algorithm on datapoints restricted to the recovered support in the beginning.

To this end, we use the algorithm in [Lia23] (denoted by OptimalOja, see Proposition A.5.3) for subgaussian data, which uses an iteration-dependent sequence of step-sizes $\{\eta_i\}_{i\in[n]}$. We run Algorithm 2 with OptimalOja as the procedure to do sparse PCA. This leads to the minimax error rate, shown in Theorem 3.7. The high probability bounds in Theorem 3.5 and 3.7 both use the support recovery guarantees derived in Section 3.1 and the boosting procedure described in Section 3.4. Detailed proofs for both results can be found in Appendix Section A.5.

**Theorem 3.7** (Data Truncation). *Let Assumptions 1 and 2 hold and $k \geq s$. For dataset $\mathcal{D} := \{X_i\}_{i\in[n]}$ and $w_0 \sim \mathcal{N}(0, I)$, let $\mathcal{A}$ be the randomized algorithm which computes $\hat{v}_{\mathsf{truncvec}} \leftarrow$* TruncateOja $\left( \left\{ \lfloor X_i \rfloor_{\widehat{S}} \right\}_{i\in\left(\frac{n}{2},n\right]}, \widehat{S}, \mathsf{OptimalOja}, \{\{\eta_t\}_{t\in[\frac{n}{2}]}, w_0\} \right)$. *Then for* $\min_i |v_1(i)| = \widetilde{\Omega}\left( \frac{\lambda_1}{\lambda_1-\lambda_2} \left(\frac{d}{n^2}\right)^{\frac{1}{4}} \right)$, $\tilde{v} \leftarrow$ SuccessBoost $\left( \{X_i\}_{i\in[n]}, \mathcal{A}, d^{-10} \right)$ *satisfies,*

$$\sin^2(\tilde{v}, v_1) \leq C'' \frac{\lambda_1 \lambda_2}{(\lambda_1 - \lambda_2)^2} \frac{k \log(d)}{n}$$

*with probability at least $1 - d^{-10}$, where $C'' \geq 0$ is an absolute constant.*

**Remark 3.8.** *Algorithm 2, with both* Oja *and* OptimalOja *as input procedures require a simple initialization vector $w_0 \sim \mathcal{N}(0, I)$. In contrast, the block stochastic power method-based algorithm presented in [YX15] provides local convergence guarantees (see Theorem 1) requiring a block size of $O(s \log(d))$. They provide an initialization procedure, but the theoretical guarantees to achieve such an initialization require block size $\Omega(d)$. [YZ13] also require a close enough initialization. In the particular setting of a single-spiked covariance model, they require $|w_0^T v_1| = \Omega(1)$. In comparison for Algorithm 2, 3.7, it suffices to have $|w_0^T v_1| \geq \frac{\delta}{\sqrt{e}}$ with probability at least $1 - \delta$ (see Lemma A.2.1).*

### 3.4 Probabilistic boosting

In this section, we describe a generic procedure for boosting the success probability of a given randomized algorithm, $\mathcal{A}$ (also see [KLL$^+$23]). If $\mathcal{A}$ satisfies Definition 3.9, then its probability can be boosted using this procedure. The formal guarantees of the boosting procedure are provided in Lemma 3.10 (proof in Appendix Section A.2). It divides the data evenly into $\log\left(\frac{1}{\delta}\right)$ buckets[6], runs $\mathcal{A}$ on each bucket and aggregates the results via pairwise comparisons.

**Definition 3.9.** *Let $\mathcal{T}$ be a set with metric $\rho$ and $\mathcal{A}$ be a randomized algorithm which takes as input $n$ i.i.d datapoints $D := \{X_i\}_{i\in[d]}$ and possibly additional statistically independent parameters $\theta$, and returns an estimate $q \in \mathcal{T}$, which satisfies $\mathbb{P}(\rho(q, q_*) \geq \epsilon) \leq \frac{1}{3}$ for a fixed $q_* \in \mathcal{T}$. Then, $\mathcal{A}$ is said to be a constant success oracle with parameters $(D, \theta, \mathcal{T}, \rho, q_*, \epsilon)$, denoted as $\mathcal{A} := \mathsf{ConstantSuccessOracle}(D, \theta, \mathcal{T}, \rho, q_*, \epsilon)$.*

---

**Algorithm 3** SuccessBoost $\left( \{X_i\}_{i\in[n]}, \mathcal{A}, \delta \right)$

---

1: **Input** : Dataset $D := \{X_i\}_{i\in[n]}$, $\mathcal{A} := \mathsf{ConstantSuccessOracle}(D, \theta, \mathcal{T}, \rho, q_*, \epsilon)$, Required failure probability $\delta$
2: **Return** : An estimate $\tilde{q} \in \mathcal{T}$ such that $\mathbb{P}(\rho(\tilde{q}, q_*) \leq 3\epsilon) \geq 1 - \delta$
3: $S \leftarrow 30 \log\left(\frac{1}{\delta}\right)$, $B \leftarrow n/S$
4: $\forall t \in [S], q_t \leftarrow \mathcal{A}\left( \{X_{B(t-1)+i}\}_{i\in[B]}, \theta, \mathcal{T}, \rho, q_*, \epsilon \right)$, $\mathcal{C}_t \leftarrow \{t' \in [S] : \rho(q_t, q_{t'}) \leq 2\epsilon\}$
5: **If** $\exists q_t$ such that $|\mathcal{C}_t|/S \geq 0.4$ **Return** $q_t$ **Else Return** $\perp$

---

[6]For simplicity, we assume $S, B$ in Algorithm 3 are integers.

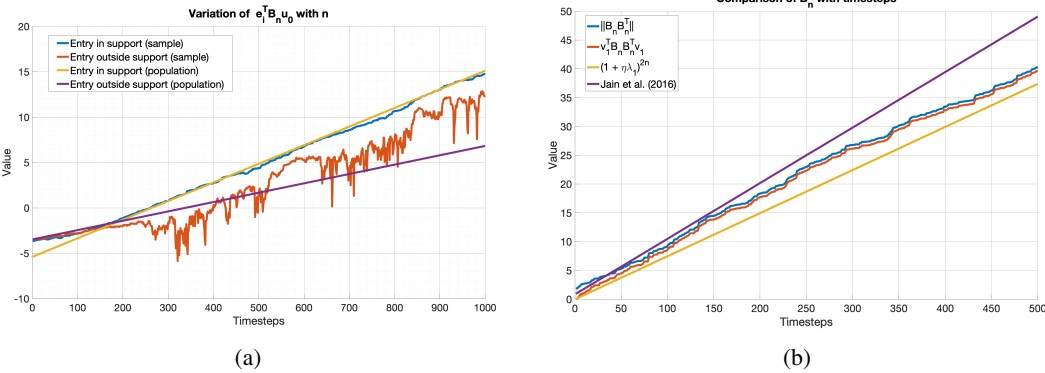

(a)                          (b)

Figure 2: We use $\Sigma$ used in [QLR19], Section 5.1. (a) Variation of $\log\left(|e_i^\top B_n u_0|\right)$ for $i \in S$ and $i \notin S$ ($y$-axis) with $n$ ($x$-axis) for a fixed unit vector $u_0$. $\eta$ is set as Theorem 3.5 and $n$ grows from 1 to 1000. The lines labelled "sample" plot $\log(|e_i^\top B_n u_0|)$, whereas the "population" curves plot $\log(|\mathbb{E}\left[e_i^\top B_n u_0\right]|)$. (b) Variation of $\log\left(\left\|B_n B_n^T\right\|\right)$ and $\log\left(v_1^T B_n B_n^T v_1\right)$ ($y$-axis) with $n \in [300]$ ($x$-axis). We also plot $\log$ of the bound of $\left\|B_n B_n^T\right\|$ as in [JJK$^+$16] and $2n \log\left(1 + \eta\lambda_1\right)$ for comparison.

**Lemma 3.10** (Geometric Aggregation for Boosting). *Let $\mathcal{A} \quad :=$* ConstantSuccessOracle $(D, \theta, \mathcal{T}, \rho, q_*, \epsilon)$ *(Definition 3.9) for dataset* $D := \{X_i\}_{i \in [n]}$. *Then for $\delta \in (0, 1)$, $\tilde{q} \leftarrow$* SuccessBoost $\left(\{X_i\}_{i \in [n]}, \mathcal{A}, \delta\right)$ *satisfies* $\mathbb{P}\left(\rho\left(\tilde{q}, q_*\right) \leq 3\epsilon\right) \geq 1 - \delta$.

### 3.5 Entrywise deviation of the Oja vector

To analyze the success probability of recovering the indices in $S$, we will define the following event, $\mathcal{E} := \{S \subseteq \widehat{S}\}$. We now upper bound $\mathbb{P}\left(\mathcal{E}^c\right)$. Define an element of the unnormalized Oja vector as $r_i := e_i^T B_n u_0, \quad i \in [d]$. Here $u_0 \sim \mathcal{N}\left(0, I\right)$ is the initialization used in Algorithm 1. Observe that

$$\mathcal{E} \iff \exists \tau_n > 0 \text{ such that } \{\forall i \in S, |r_i| \geq \tau_n\} \bigcap \{|\{i : i \notin S, |r_i| \geq \tau_n\}| \leq k - s\}$$

or equivalently,

$$\mathcal{E}^c \iff \forall \tau_n > 0, \{\exists i \in S, |r_i| \leq \tau_n\} \bigcup \{|\{i : i \notin S, |r_i| \geq \tau_n\}| > k - s\}$$

Therefore, for any fixed $\tau_n > 0$, $\mathcal{E}^c \implies \{\exists i \in S, |r_i| \leq \tau_n\} \bigcup \{|\{i : i \notin S, |r_i| \geq \tau_n\}| > k - s\}$. We will, therefore, be interested in the tail behavior of $r_i$ for $i \in S$ and $i \notin S$. Before presenting our theorems, we will use Figure 2 to emphasize the daunting nature of what we aim to prove. Consider the quantity $\mathbb{E}\left[r_i | u_0\right] = \mathbb{E}\left[e_i^T B_n u_0 | u_0\right]$. We use $X = C \pm \Delta$ to denote $|X - C| \leq \Delta$.

$$\mathbb{E}\left[r_i | u_0\right] = e_i^T \mathbb{E}\left[B_n\right] v_1 v_1^T u_0 + e_i^T \mathbb{E}\left[B_n\right] V_\perp V_\perp^T u_0 \tag{4}$$

$$= \begin{cases} e_i^T v_1 v_1^T u_0 (1 + \eta\lambda_1)^n \pm \left|e_i^T V_\perp V_\perp^T u_0\right| (1 + \eta\lambda_2)^n & \text{For } i \in S \\ \pm \left|e_i^T V_\perp V_\perp^T u_0\right| (1 + \eta\lambda_2)^n & \text{For } i \notin S \end{cases}$$

Thus, traditional wisdom would make us hope that the elements, $r_i$, will concentrate around their respective expectations, whose absolute values are off by a ratio $|v_1(i)||u_0^T v_1| \exp(n\eta(\lambda_1 - \lambda_2))$.

However, Figure 2(a) shows that while the elements in the support seem *close* to their expectation, those not in support are, on average, *much larger* than their expectation. First, note that elementwise analysis of the Oja vector has not been done even in the low dimensional regime where $r_{\text{eff}}/n \to 0$. In this regime, there is very recent related work for eigenvectors of the empirical covariance matrix $\hat{\Sigma}$ [AFW22] which are not applicable here. In the high-dimensional case, an analog can be drawn with elements $\hat{\Sigma}$, which concentrate around their mean individually. Yet, $\|\hat{\Sigma} - \Sigma\|$ is not small. Thus, thresholding $\hat{\Sigma}$ obtains consistent estimates of $\Sigma$ under sparsity assumptions [BL09, DM$^+$16, Nov23].

A similar principle is applied by [SSM11] where the eigenvector of $\hat{\Sigma}$ is truncated. They assume that $n$ is fixed, and $\lambda_1/\lambda_2 = d^\alpha \to \infty$ as $d \to \infty$. In comparison, our analysis is about products of random matrices, not sums, and hence, completely different. We will show that $\hat{v}_1(i)$, even when $i \in S$, do not concentrate. But for a suitably chosen threshold, They are large with high probability,

whereas those outside $S$ are much lower with high probability. Proving this is also difficult because the analysis involves the concentration of the projection of a product of independent high dimensional matrices on some initial random vector. Lemma 3.11 establishes exactly that for elements in the support.

**Lemma 3.11** (Tail bound in support). *Fix a $\delta \in (0.1, 1)$. Define the event $\mathcal{G} := \left\{ |v_1^T u_0| \geq \frac{\delta}{\sqrt{e}} \right\}$ and threshold $\tau_n := \frac{\delta}{\sqrt{2e}} \min_{i \in S} |v_1(i)| (1 + \eta \lambda_1)^n$. Let the learning rate be set as in Lemma 3.1. Then, for an absolute constant $C_H > 0$,*

$$\forall i \in S, \quad \mathbb{P}\left( |r_i| \leq \tau_n \Big| \mathcal{G} \right) \leq C_H \left[ \eta \lambda_1 \log(n) + \eta \lambda_1 \left( \frac{\lambda_1}{\lambda_1 - \lambda_2} \right) \frac{1}{v_1(i)^2} \right]$$

Our next result provides a bound for $i \notin S$.

**Lemma 3.12** (Tail bound outside support). *Fix a $\delta \in (0.1, 1)$. Let the learning rate be set as in Lemma 3.1 and define the threshold $\tau_n := \frac{\delta}{\sqrt{2e}} \min_{i \in S} |v_1(i)| (1 + \eta \lambda_1)^n$. Then, for $\min_i |v_1(i)| = \widetilde{\Omega}\left( \frac{\lambda_1}{\lambda_1 - \lambda_2} \left( \frac{d}{n^2} \right)^{\frac{1}{4}} \right)$ and an absolute constant $C_T > 0$ we have,*

$$\forall i \notin S, \quad \mathbb{P}(|r_i| > \tau_n) \leq C_T \left[ \eta^2 \lambda_1^2 \left( \frac{\lambda_1}{\lambda_1 - \lambda_2} \right)^2 \left( \frac{1}{\delta^2 \min_{i \in S_{h_i}} v_1(i)^2} \right)^2 \right]$$

The proofs of Lemmas 3.11 and 3.12 are based on tail-bounds involving the second and fourth moments of $r_i := e_i^T B_n u_0$. The details of obtaining the tail bounds are deferred to the Appendix Section A.3. The results developed in this section are used to analyze the support recovery and $\sin^2$ error guarantees provided in Section 3.1 and 3.3. We provide a brief proof sketch in Section 4.

## 4 Proof technique

In this section, we outline the proof techniques for the entrywise deviation bounds in Lemmas 3.11 and 3.12. These bounds are crucial for analyzing both the support recovery results (Lemma 3.1 and Theorem 3.2) and the sparse PCA results (Theorems 3.5 and 3.7). The proof involves deriving bounds on the expectation and second moment of $u_0^T B_n U U^T B_n u_0$, where $U \in \mathbb{R}^{d \times k}$ is a fixed matrix and $u_0 \sim \mathcal{N}(0, I)$. It then applies Chebyshev's inequality to obtain the tail bound. For the proof sketch, we use $U = e_i$, but we maintain general notation for broader applicability in Theorem 3.5. For our results, we also need to bound this quantity with $U = I_S$ (see Lemma A.5.1 for details). Our techniques to bound $\mathbb{E}[u_0^T B_n U U^T B_n u_0]$ are detailed in Section 4.1.

### 4.1 Solving a linear system of recursions

One can show that (see Lemma A.2.11 in Appendix),

$$\mathbb{E}\left[ u_0^T B_n^T U U^T B_n u_0 \right] = \underbrace{\mathbb{E}\left[ v_1^T B_n^T U U^T B_n v_1 \right]}_{=: \alpha_n} + \underbrace{\mathbb{E}\left[ \text{Tr}\left( V_\perp^T B_n^T U U^T B_n V_\perp \right) \right]}_{=: \beta_n} \quad (5)$$

We start by showing how to bound $\alpha_n$ and $\beta_n$. Before we dive into our techniques, we note that the analysis of Oja's algorithm [JJK$^+$16] in the non-sparse setting provides some tools that we could potentially use here. Using the recursion from Lemma 9 in [JJK$^+$16], we get

$$\left\| \mathbb{E}\left[ B_n U U^T B_n^T \right] \right\| \leq \exp(2n\eta\lambda_1 + n\eta^2 \mathcal{V}) \left\| U U^T \right\|, \quad (6)$$

where $\mathcal{V}$ is a variance parameter defined as $\left\| \mathbb{E}\left[ (A_1 - \Sigma)(A_1 - \Sigma)^T \right] \right\|$. Lemma A.2.3 shows that for $\sigma$-subgaussian $X$ (definition 2.1),

$$\mathcal{V} := \left\| \mathbb{E}\left[ (A_1 - \Sigma)(A_1 - \Sigma)^T \right] \right\| = \left\| \mathbb{E}\left[ A_1 A_1^T \right] - \Sigma^2 \right\| \leq 2L^4 \sigma^4 \lambda_1 \text{Tr}(\Sigma) + \lambda_1^2$$

This provides an upper bound on $\alpha_n \leq \left\| \mathbb{E}\left[ B_n U U^T B_n^T \right] \right\|$. While this bound is tight when $r_{\text{eff}}$ is bounded by a constant, in the high dimensional setting (Assumption 2) considered in this work, this bound is too loose. This is evident from Figure 2(B), which plots $\left\| \mathbb{E}\left[ B_n U U^T B_n^T \right] \right\|$ for $U = I$, along with the bound achieved using Eq 6, labeled as *Jain et al. (2016)*. Note that the plots are in

the log-scale so a difference in the slopes translates to a significant multiplicative difference. This warrants a more fine-grained analysis of $\alpha_n$.

Let us examine $\alpha_n$ more closely to obtain a finer bound. Using the structure of the matrix product, $B_n$, from Eq 3, we have:

$$\alpha_n = \alpha_{n-1}\left(1 + 2\eta\lambda_1\right) + \eta^2 \mathbb{E}\left[\left(v_1^T X_n X_n^T v_1\right)\left(X_n^T B_{n-1} U U^T B_{n-1}^T X_n\right)\right]$$

Now, as a consequence of subgaussianity (see Lemma A.2.2), for $K := (2L^2\sigma^2)^2$, with the Cauchy-Schwartz inequality, the second term in the RHS can be bounded further using:

$$\mathbb{E}\left[(v_1^T X_n X_n^T v_1)^2\right] \leq K\lambda_1^2, \quad \mathbb{E}\left[\left(X_n^T B_{n-1} U U^T B_{n-1}^T X_n\right)^2 \middle| \mathcal{F}_{n-1}\right] \leq K\operatorname{Tr}(U^T B_{n-1}^T \Sigma B_{n-1} U)^2$$

Therefore, using the above bound along with the eigen-decomposition $\Sigma := \lambda_1 v_1 v_1^T + V_\perp \Lambda_2 V_\perp^T$,

$$\alpha_n \leq \left(1 + 2\eta\lambda_1 + 4L^2\eta^2\sigma^4\lambda_1^2\right)\alpha_{n-1} + 4L^2\eta^2\sigma^4\lambda_1\lambda_2\beta_{n-1} \tag{7}$$

Similarly, $\beta_n$ can also be upper bounded as follows:

$$\beta_n \leq \left(1 + 2\eta\lambda_2 + 4\eta^2 L^4\sigma^4\lambda_2 \operatorname{Tr}(\Sigma)\right)\beta_{n-1} + 4\eta^2 L^4\sigma^4\lambda_1 \operatorname{Tr}(\Sigma)\alpha_{n-1} \tag{8}$$

Note that upper bounding and eliminating $\alpha_{n-1}$ or $\beta_{n-1}$ from Eq 8, 7 respectively, would simplify the recursion but lead to a weaker bound as in Eq 6. Therefore, we solve Eq 7 and 8 as a system of linear recursions in $\alpha_n$ and $\beta_n$.

$$\begin{pmatrix} \alpha_n \\ \beta_n \end{pmatrix} = \underbrace{\begin{pmatrix} 1 + 2\eta\lambda_1 + O\left(\eta^2\lambda_1^2\right) & O\left(\lambda_1\lambda_2\right) \\ O\left(\operatorname{Tr}(\Sigma)\right) & 1 + 2\eta\lambda_2 + O\left(\eta^2 \operatorname{Tr}(\Sigma)\right) \end{pmatrix}}_{:=P} \begin{pmatrix} \alpha_{n-1} \\ \beta_{n-1} \end{pmatrix} \tag{9}$$

Estimating elements of $P^n$, where $P$ is the defined $2 \times 2$ matrix, is crucial. [Wil92] gives a compact expression for these elements using $\lambda_1(P)$ and $\lambda_2(P)$. Under our assumptions, we have $P_{11} > P_{22}$. A naive upper bound on $\lambda_1(P)$ using Weyl's inequality [Die15] is $1 + 2\eta\lambda_1 + c_3 \operatorname{Tr}(\Sigma)$, similar to Eq 6. Since recursions like Eq 9 are common in our analysis, we provide a general solution in Lemma A.2.5 (detailed in the Appendix Section A.2).

An important consequence of this is that we now have the following bounds on $\alpha_n$ for $U = I$:

$$\alpha_n \leq (1 + 2\eta\lambda_1 + c_1\eta^2\lambda_1^2)^n \left(1 + O(\eta\lambda_1)\right) \tag{10}$$

which is much tighter than Eq 6 in our high-dimensional regime. Furthermore, observing Figure 2(b), we see that Eq 10 presents a much tighter upper bound, matching $(1 + \eta\lambda_1)^{2n}$ up to constant factors.

Recall that the bounds obtained in this section deal with $\alpha_n, \beta_n$ defined in Eq 5. A similar system of recursions can be obtained to get tight bounds on $\mathbb{E}\left[\left(v_1^T B_n^T U U^T B_n v_1\right)^2\right]$ and $\mathbb{E}\left[\operatorname{Tr}\left(V_\perp^T B_n^T U U^T B_n V_\perp\right)^2\right]$, details of which we defer to the Appendix in Lemmas A.2.9, A.2.10.

## 5 Conclusion

Oja's algorithm for streaming PCA has been extensively studied in the recent theoretical literature, typically assuming that $\|X_i\|^2/\lambda_1$ is bounded or a slowly growing covariance matrix effective rank $r_{\text{eff}}$. This paper addresses the high-dimensional sparse PCA setting where the effective rank $r_{\text{eff}}$ can be as large as $n/\log n$ while $v_1$ is $s$-sparse. In this context, while there has been a vast body of work that achieves minimax error bounds, we are unaware of any single-pass algorithm that works in $O(nd)$ time, $O(d)$ space, on a general $\Sigma$, without any strong initialization. Surprisingly, our thresholded estimator achieves the minimax error bound of $O(s\log d/n)$, whereas the error rate of Oja's algorithm is $O(r_{\text{eff}}/n)$. Empirically, the elements of the unnormalized Oja vector do not concentrate in this regime. Through an analysis that uncouples the projection of a product of independent random matrices on $v_1$ and its orthogonal subspace, we show that the entries of the Oja vector within the support of $v_1$ are large, while those outside are much smaller.

## Acknowledgments and Disclosure of Funding

We gratefully acknowledge NSF grants 2217069, 2019844, and DMS 2109155. We thank Kevin Tian for his valuable insight on geometric aggregation and boosting and the anonymous reviewers for their valuable feedback.

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

# A Appendix

The Appendix is organized as follows:

1. Section A.1 provides further details about related work
2. Section A.2 provides some useful results used in subsequent analyses
3. Section A.3 provides Entrywise deviation bounds for the Oja vector (Lemmas 3.11, 3.12)
4. Section A.4 proves convergence of Support Recovery results (Lemma 3.1, Theorem 3.2)
5. Section A.5 proves convergence of Sparse PCA results (Theorems 3.5, 3.7)
6. Section A.6 provides another alternative way of truncation using a value-based thresholding (Theorem A.6.1)

## A.1 Further details on related work

There has been a lot of work on computational computational hardness of sparse PCA [GMZ17, BB19, DKWB23, BKW20].

**Minimax optimal Sparse PCA algorithms with global convergence:** These consist of SDP-based algorithms such as [AW08, VCLR13, dBEG08], which do not scale well in high-dimensions (see [BR13, Wai19]). The state-of-the-art SDP solvers [JKL$^+$20, HJS$^+$22] currently have a runtime $\Omega\left(n^\omega + d^\omega\right)$, where $\omega \approx 2.732$ is the matrix multiplication exponent. Algorithms proposed in [Ma13, CMW13, JNRS10, DM$^+$16] involve forming the entire $(d \times d)$ sample covariance matrix, which can itself be challenging from the perspective of space and time complexity. Furthermore, [Ma13, CMW13, DM$^+$16] have been analyzed under the *spiked covariance model* in Eq 2. [QLR19] propose a computationally efficient modification of the Fantope projection-based algorithm of [VCLR13], which requires $O(d^2)$ space, and $\Omega(nd^2)$ time.

**Single-pass online sparse PCA algorithms with $O(d^2)$ storage and $O(nd^2)$ time** [QLR19] also provide a single-pass online algorithm and state that this algorithm (Theorem 4) *is the first to provably obtain the global optima in a streaming setting without any initialization, under a general $\Sigma$.* However, this method requires $O(d^2)$ storage, $O(nd^2)$ time, and the estimation error is $O\left(\frac{d^2}{\sqrt{n}}\right)$ (Theorem 4, [QLR19]). The algorithm does $d$ sparse linear regression problems to achieve this.

**Support recovery algorithms with $O(d^2)$ storage and $O(nd^2)$ time** : [LV15, LSH22] use an SDP-based approach and [BPP18] use sparse linear regression for support recovery.

**More details on streaming PCA algorithms** [YX15] provides an online block version of the truncated power method in [YZ13] under the spiked model (Eq 2). They require an initialization $u_0$ with a sufficiently large $|u_0^T v_1| = \Omega\left(1\right)$ (local convergence). Their proposed initialization with streaming PCA algorithm until reaching a specific accuracy threshold, for which there is no known theoretical guarantee under the spiked high-dimensional setting. [WL16] provides an analysis of streaming sparse PCA under Eq 2 via partial differential equations (PDE), but they only prove asymptotic convergence. Similar to [YZ13], they also require $|u_0^T v_1| = \Omega\left(1\right)$ which can be hard to find in high dimensions for a general $\Sigma$. Recent results provide a black-box way to obtain the top-$k$ principal components ($k$-PCA) given an algorithm to extract the top eigenvector (see [a]) which could be employed treating our algorithm as a 1-PCA oracle (see [JKL$^+$24, Mac08]). We believe that our analysis can be extended to obtain top-k principal components simultaneously via QR decomposition and thresholding.

## A.2 Useful results

**Lemma A.2.1.** *(Fact 2.9 [DKPP23]) For any symmetric $d \times d$ matrix A, we have* $\mathrm{Var}_{z \sim \mathcal{N}(0,I)}\left[z^T A z\right] = 2\|A\|_F^2$. *If A is a PSD matrix, then for any $\beta > 0$, it holds that*

$$\mathbb{P}_{z \sim \mathcal{N}(0,I)}\left[z^T A z \geq \beta \, \mathrm{Tr}\left(A\right)\right] \geq 1 - \sqrt{e\beta}$$

*Proof.* We give a short proof here. Since A is a symmetric matrix, let $A = P \Lambda P^T$ where $P$ is an orthonormal matrix and $\Lambda$ is a diagonal matrix. Then, denoting $y := P^T z$ we note that $y \sim \mathcal{N}\left(0, I\right)$.

Therefore,

$$z^T A z = z^T P \Lambda P^T z = y^T \Lambda y = \sum_{i=1}^{d} \lambda_i y_i^2$$

Therefore,

$$\mathbb{E}_{z \sim \mathcal{N}(0,I)} \left[ z^T A z \right] = \mathbb{E}_{y \sim \mathcal{N}(0,I)} \left[ \sum_{i=1}^{d} \lambda_i y_i^2 \right] = \sum_{i=1}^{d} \lambda_i = \operatorname{Tr}(A)$$

and

$$\mathbb{E}_{z \sim \mathcal{N}(0,I)} \left[ \left( z^T A z \right)^2 \right] = \mathbb{E}_{y \sim \mathcal{N}(0,I)} \left[ \left( \sum_{i=1}^{d} \lambda_i y_i^2 \right)^2 \right] = \mathbb{E}_{y \sim \mathcal{N}(0,I)} \left[ \sum_{i=1}^{d} \lambda_i^2 y_i^4 + \sum_{i,j,i \neq j}^{d} \lambda_i \lambda_j y_i^2 y_j^2 \right]$$

$$= 3 \sum_{i=1}^{d} \lambda_i^2 + \sum_{i,j,i \neq j}^{d} \lambda_i \lambda_j = 2 \operatorname{Tr}(A^2) + \operatorname{Tr}(A)^2$$

To get the tail lower bound, note that it trivially follows if $\beta > 1$. Therefore we proceed with $\beta \in (0,1)$. We have

$$\mathbb{P}_{z \sim \mathcal{N}(0,I)} \left[ z^T A z \leq \beta \operatorname{Tr}(A) \right] = \mathbb{P}_{y \sim \mathcal{N}(0,I)} \left[ \sum_{i=1}^{d} \lambda_i y_i^2 \leq \beta \sum_{i=1}^{d} \lambda_i \right]$$

$$\leq \mathbb{E} \left[ \exp \left( t \left( \beta \sum_{i=1}^{d} \lambda_i - \sum_{i=1}^{d} \lambda_i y_i^2 \right) \right) \right], t > 0$$

$$= \exp \left( t \beta \sum_{i=1}^{d} \lambda_i \right) \mathbb{E} \left[ \exp \left( -t \sum_{i=1}^{d} \lambda_i y_i^2 \right) \right]$$

$$= \exp \left( t \beta \sum_{i=1}^{d} \lambda_i \right) \prod_{i=1}^{d} (1 + 2 \lambda_i t)^{-\frac{1}{2}}$$

Let $t = \frac{1}{2 \sum_{i=1}^{d} \lambda_i} \left( \frac{1}{\beta} - 1 \right)$. Then,

$$\mathbb{P}_{z \sim \mathcal{N}(0,I)} \left[ z^T A z \leq \beta \operatorname{Tr}(A) \right] \leq \exp \left( \frac{1-\beta}{2} \right) \prod_{i=1}^{d} \left( 1 + \frac{\lambda_i}{\sum_{i=1}^{d} \lambda_i} \left( \frac{1}{\beta} - 1 \right) \right)^{-\frac{1}{2}}$$

$$\leq \exp \left( \frac{1-\beta}{2} \right) \left( 1 + \left( \frac{1}{\beta} - 1 \right) \right)^{-\frac{1}{2}}$$

$$= \exp \left( \frac{1-\beta}{2} \right) \sqrt{\beta}$$

$$\leq \sqrt{e\beta}$$

Hence proved. $\qquad \square$

**Lemma A.2.2.** *Let $X \in \mathbb{R}^d$ be a $\sigma$-subgaussian random vector with covariance matrix $\Sigma$. Then, for any matrix $M \in \mathbb{R}^{d \times m}$ and any positive integer $p \geq 2$,*

$$\mathbb{E} \left[ (X^T M M^T X)^p \right] \leq (L^2 \sigma^2 p)^p \operatorname{Tr}(M^T \Sigma M)^p$$

*Proof.* Let the eigendecomposition of $M M^T$ be $P \Lambda P^T$. Define $Y := P^T X$. Then,

$$\mathbb{E} \left[ (X^T M M^T X)^p \right] = \mathbb{E} \left[ \left( \sum_{i=1}^{d} \lambda_i y_i^2 \right)^p \right]$$

$$= \sum_{k_1 + k_2 + \cdots k_d = p;\, k_1, k_2, \cdots k_d \geq 0} \binom{n}{k_1, k_2, \cdots k_d} \mathbb{E} \left[ \prod_{i=1}^{d} \lambda_i^{k_i} y_i^{2k_i} \right] \qquad \text{(A.11)}$$

Therefore,

$$
\mathbb{E}\left[\prod_{i=1}^{d}\lambda_i^{k_i}y_i^{2k_i}\right] = \left(\prod_{i=1}^{d}\lambda_i^{k_i}\right)\mathbb{E}\left[\prod_{i=1}^{d}y_i^{2k_i}\right]
$$

$$
\leq \left(\prod_{i=1}^{d}\lambda_i^{k_i}\right)\prod_{i=1}^{d}\left(\mathbb{E}\left[\left(y_i^{2k_i}\right)^{\frac{p}{k_i}}\right]\right)^{\frac{k_i}{p}}, \text{ using Holder's inequality since } \sum_{i=1}^{d}k_i = p,
$$

$$
= \left(\prod_{i=1}^{d}\lambda_i^{k_i}\right)\prod_{i=1}^{d}\left(\mathbb{E}\left[y_i^{2p}\right]\right)^{\frac{k_i}{p}} \tag{A.12}
$$

Using the definition of sub-gaussianity (Definition 2.1) we have,

$$
\mathbb{E}\left[y_i^{2p}\right] = \mathbb{E}\left[\left(e_i^T P^T X\right)^{2p}\right]
$$

$$
= \mathbb{E}\left[\left((Pe_i)^T X\right)^{2p}\right]
$$

$$
\leq L^{2p}\sigma^{2p}\left(\sqrt{p}\right)^{2p}\left(\|Pe_i\|_{\Sigma}\right)^{2p}
$$

$$
= L^{2p}\sigma^{2p}p^p\left(e_i^T P^T \Sigma P e_i\right)^p
$$

Susbstituting in Eq A.12 we have,

$$
\mathbb{E}\left[\prod_{i=1}^{d}\lambda_i^{k_i}y_i^{2k_i}\right] \leq \left(\prod_{i=1}^{d}\lambda_i^{k_i}\right)\left(\prod_{i=1}^{d}L^{2k_i}\sigma^{2k_i}p^{k_i}\left(e_i^T P^T \Sigma P e_i\right)^{k_i}\right)
$$

$$
= \left(L^2\sigma^2 p\right)^p\prod_{i=1}^{d}\left(\lambda_i e_i^T P^T \Sigma P e_i\right)^{k_i}
$$

Substituting in Eq A.11 we have,

$$
\mathbb{E}\left[\left(X^T MM^T X\right)^p\right] \leq \left(L^2\sigma^2 p\right)^p \sum_{k_1+k_2+\cdots k_d=p;\ k_1,k_2,\cdots k_d\geq 0}\binom{n}{k_1,k_2,\cdots k_d}\prod_{i=1}^{d}\left(\lambda_i e_i^T P^T \Sigma P e_i\right)^{k_i}
$$

$$
= \left(L^2\sigma^2 p\right)^p\left(\sum_{i=1}^{d}\lambda_i e_i^T P^T \Sigma P e_i\right)^p
$$

$$
= \left(L^2\sigma^2 p\right)^p\left(\mathrm{Tr}\left(\left(\sum_{i=1}^{d}\lambda_i P e_i e_i^T P^T\right)\Sigma\right)\right)^p = \left(L^2\sigma^2 p\right)^p\mathrm{Tr}\left(M^T \Sigma M\right)^p
$$

Hence proved. $\qquad\square$

**Lemma A.2.3.** *Let $X \in \mathbb{R}^d$ be a $\sigma$-subgaussian random vector with covariance matrix $\Sigma$. Then,*

$$
\left\|\mathbb{E}\left[\left(XX^T\right)^2\right]\right\| \leq 4L^4\sigma^4\lambda_1\mathrm{Tr}\left(\Sigma\right)
$$

*Proof.* For any fixed unit vector $u \in \mathbb{R}^d$, we have

$$
u^T\mathbb{E}\left[\left(XX^T\right)^2\right]u = \mathbb{E}\left[\left(X^T X\right)\left(X^T u\right)^2\right]
$$

$$
\leq \sqrt{\mathbb{E}\left[\left(X^T X\right)^2\right]\mathbb{E}\left[\left(X^T u\right)^4\right]}
$$

$$
= \sqrt{\mathbb{E}\left[\left(X^T X\right)^2\right]\mathbb{E}\left[\left(X^T uu^T X\right)^2\right]}
$$

$$
\leq \left(2L^2\sigma^2\right)^2\mathrm{Tr}\left(\Sigma\right)\mathrm{Tr}\left(u^T \Sigma u\right)
$$

$$
\leq \left(2L^2\sigma^2\right)^2\lambda_1\mathrm{Tr}\left(\Sigma\right)
$$

where we used Lemma A.2.2 with $p = 2$ and $M = I$. $\qquad\square$

**Proposition 3.4** (Lower bound for diagonal thresholding). *Let Assumption 1 hold. For any diagonal-thresholding algorithm, $\mathcal{A}$, performing support recovery with sparsity parameter $s$ such that $n = \Omega\left(\sigma^4 s^2 \log(d)\right)$, there exists a covariance matrix $\Sigma$ with principal eigenvector, $v_1$, $\|v_1\|_0 = s$, such that, $\mathbb{P}\left(\left|\widehat{S}\bigcap S\right| = 0\right) \geq 1 - d^{-10}$.*

*Proof.* Let $s$ be a multiple of 3 for ease of analysis. Consider a dataset with a covariance matrix,

$$\Sigma := \beta_1 v_1 v_1^\top + \beta_2 v_2 v_2^\top + \beta_3 v_2 v_2^\top + \beta_4 v_2 v_2^\top + \frac{1}{2}I, \beta_1 = 2\beta_2 = 2.1\beta_3 = 2.2\beta_4$$

$$\forall i \in [s], |v_1(i)| = \frac{1}{\sqrt{s}}, \quad \forall i \in \left(s, \frac{4s}{3}\right], |v_2(i)| = \sqrt{\frac{3}{s}}$$

$$\forall i \in \left(\frac{4s}{3}, \frac{5s}{3}\right], |v_3(i)| = \sqrt{\frac{3}{s}} \quad \forall i \in \left(\frac{5s}{3}, 2s\right], |v_4(i)| = \sqrt{\frac{3}{s}} \tag{A.13}$$

where $\beta_1 = \frac{1}{2}$. Based on Eq A.13, we have for,

$$i \in (1, s], \Sigma_{i,i} = \frac{1}{2} + \frac{\beta_1}{s}$$

$$i \in \left(s, \frac{4s}{3}\right], \Sigma_{i,i} = \frac{1}{2} + \frac{3\beta_2}{s} = \frac{1}{2} + \frac{3\beta_1}{2s}$$

$$i \in \left(\frac{4s}{3}, \frac{5s}{3}\right], \Sigma_{i,i} = \frac{1}{2} + \frac{3\beta_3}{s} = \frac{1}{2} + \frac{3\beta_1}{2.1s}$$

$$i \in \left(\frac{5s}{3}, 2s\right], \Sigma_{i,i} = \frac{1}{2} + \frac{3\beta_4}{s} = \frac{1}{2} + \frac{3\beta_1}{2.2s}$$

$$i \in (2s, d], \Sigma_{i,i} = \frac{1}{2}$$

Note that the largest eigenvalue of $\Sigma$, $\lambda_1 = \beta_1 + \frac{1}{2}$. Let $t_n := 10\sigma^2 \lambda_1 \sqrt{\frac{\log(d)}{n}}$. Using Lemma 6.26 from [Wai19], we have, for the empirical covariance matrix, $\hat{\Sigma}$,

$$\mathbb{P}\left(\max_{i,j \in [d]} \left|\hat{\Sigma}_{i,j} - \Sigma(i,j)\right| \geq t_n\right) \leq \frac{1}{d^{10}}$$

Define the event, $\mathcal{E} := \max_{i,j \in [d]} \left|\hat{\Sigma}(i,j) - \Sigma(i,j)\right| \leq t_n$ and note that due to the sample complexity bound on $n$, under event $\mathcal{E}$,

$$\min_{i \in (s, 2s]} \hat{\Sigma}_{i,i} > \max_{i \in [1,s]} \hat{\Sigma}_{i,i} \geq \min_{i \in [1,s]} \hat{\Sigma}_{i,i} \geq \max_{i > 2s} \hat{\Sigma}_{i,i}$$

Therefore, under event $\mathcal{E}$, the $s$ largest diagonal entries of $\hat{\Sigma}$ are $i \in (s, 2s]$, and therefore, $|\hat{S}\bigcap S| = 0$, which completes our proof. $\square$

**Lemma 3.10** (Geometric Aggregation for Boosting). *Let $\mathcal{A} := \mathsf{ConstantSuccessOracle}\left(D, \theta, \mathcal{T}, \rho, q_*, \epsilon\right)$ (Definition 3.9) for dataset $D := \{X_i\}_{i \in [n]}$. Then for $\delta \in (0,1)$, $\tilde{q} \leftarrow \mathsf{SuccessBoost}\left(\{X_i\}_{i \in [n]}, \mathcal{A}, \delta\right)$ satisfies $\mathbb{P}\left(\rho\left(\tilde{q}, q_*\right) \leq 3\epsilon\right) \geq 1 - \delta$.*

*Proof.* Consider the indicator random variables $\chi_i := \mathbb{1}\left(\rho(q_i, q_*) \leq \epsilon\right)$. Let $p := \frac{1}{3}$ and $r = 300 \log\left(\frac{1}{\delta}\right)$ for convenience of notation. Then, $\forall i \in [r]$, $\mathbb{P}(\chi_i = 1) \geq 1 - p$. Define the set $\mathcal{S} := \{i : i \in [r], \chi_i = 1\}$. We note that using standard Chernoff bounds for sums of independent Bernoulli random variables, for $\theta \in (0,1)$,

$$\mathbb{P}\left(|\mathcal{S}| \leq (1 - \theta)\mathbb{E}[|\mathcal{S}|]\right) \leq \exp\left(-\frac{\theta^2 \mathbb{E}[|\mathcal{S}|]}{2}\right)$$

We have, $\mathbb{E}\left[|\mathcal{S}|\right] \geq r\left(1-p\right)$ using linearity of expectation. Therefore,

$$\mathbb{P}\left(|\mathcal{S}|\leq \left(1-\theta\right)\left(1-p\right)r\right) \leq \exp\left(-\frac{\theta^2\left(1-p\right)r}{2}\right)$$

$$\implies \mathbb{P}\left(|\mathcal{S}|\leq 0.9\left(1-p\right)r\right) \leq \exp\left(-\frac{\left(1-p\right)r}{200}\right), \text{ for } \theta := \frac{1}{10} \tag{A.14}$$

Recall that Algorithm 3 defines $\tilde{q}$ as:

$$\tilde{q} := q_i, \text{ such that } \frac{|\{j \in [r] : \rho\left(q_i, q_j\right) \leq 2\epsilon\}|}{r} \geq 0.9\left(1-p\right) \tag{A.15}$$

Note that the definition of $\tilde{q}$ does not require knowledge of $q_*$ and it can be computed by calculating $\rho(.)$ error between all distinct $\binom{r}{2}$ pairs $\left(q_i, q_j\right)_{i,j\in[r], i\neq j}$.

Let $\mathcal{E}$ be the event $\{|\mathcal{S}|> 0.9\left(1-p\right)r\}$ and denote $f := 0.9\left(1-p\right)$ for convenience of notation. Let us now operate conditioned on $\mathcal{E}$. Note that conditioned on $\mathcal{E}$, such a $\tilde{q}$ always exists since any point in $\mathcal{S}$ is a valid selection of $\tilde{q}$. This is true since

$$\rho\left(q_i, q_j\right) \leq \rho\left(q_i, q_*\right) + \rho\left(q_j, q_*\right) \leq 2\epsilon$$

Here we used the property of the event $\mathcal{E}$ and the triangle inequality for $\rho$. We further have, conditioned on $\mathcal{E}$ using triangle inequality for some $i \in \mathcal{S}$,

$$\rho\left(\tilde{q}, q_*\right) \leq \rho\left(\tilde{q}, q_i\right) + \rho\left(q_i, q_*\right) \leq 3\epsilon \tag{A.16}$$

Therefore, we have

$$\begin{aligned}
\mathbb{P}\left(\rho\left(\tilde{q}, q_*\right) \geq 3\epsilon\right) &= \mathbb{P}\left(\mathcal{E}\right)\mathbb{P}\left(\rho\left(\tilde{q}, q_*\right) \geq 3\epsilon|\mathcal{E}\right) + \mathbb{P}\left(\mathcal{E}^c\right)\mathbb{P}\left(\rho\left(\tilde{q}, q_*\right) \geq 3\epsilon|\mathcal{E}^c\right) \\
&= 0 + \mathbb{P}\left(\mathcal{E}^c\right)\mathbb{P}\left(\rho\left(\tilde{q}, q_*\right) \geq 3\epsilon|\mathcal{E}^c\right) \text{ using Eq A.16} \\
&\leq \mathbb{P}\left(\mathcal{E}^c\right) \\
&\leq \exp\left(-\frac{\left(1-p\right)r}{200}\right) \text{ using Eq A.14}
\end{aligned}$$

which completes our proof. $\qquad\square$

**Lemma A.2.4** (Learning rate schedule). *Let the learning rate be set as $\eta := \dfrac{\kappa \log\left(n\right)}{n\left(\lambda_1 - \lambda_2\right)}$ for a positive constant $\kappa > 0$. For constant $c \leq \frac{1}{8\kappa}\min\left\{\frac{1}{\sqrt{C}}\frac{1}{C}\right\}$, let*

$$\max\left\{1, \frac{\lambda_2}{\lambda_1 - \lambda_2}\right\}\frac{\mathrm{Tr}\left(\Lambda_2\right)}{\lambda_1 - \lambda_2} \leq \frac{cn}{\log\left(n\right)}, \quad \frac{\lambda_1}{\lambda_1 - \lambda_2} \leq c\sqrt{\frac{n}{\log^2\left(n\right)}}$$

*If $\kappa \geq 2 + o\left(1\right)$, $n = \Omega\left(s^2\log\left(d\right)\right)$, the following hold:*

1. *$\eta \leq \frac{1}{C}\frac{\left(\lambda_1-\lambda_2\right)}{\lambda_2 \mathrm{Tr}(\Lambda_2)}$*

2. *$C\eta \leq \frac{1}{4}\min\left\{\frac{1}{\lambda_1}, \frac{1}{\mathrm{Tr}(\Lambda_2)}, \frac{1}{\sqrt{\lambda_1 \mathrm{Tr}(\Lambda_2)}}\right\}$*

3. *$C\eta^2 n\lambda_1^2 \leq \frac{1}{4}$*

4. *$\exp\left(-rn\eta\left(\lambda_1 - \lambda_2\right)\right) \leq \eta\lambda_1$ for $r \geq \frac{1}{2}$*

*where $C := 100\left(L^4\sigma^4 + L^2\sigma^2\right) + 16$. We state another useful restatement of Claim (1) used in subsequent analysis,*

$$\exists \theta \in \left(0.5, 1\right), \quad \left(1-\theta\right)\left(\lambda_1 - \lambda_2\right) + 50L^4\sigma^4\eta\lambda_1^2 = 50L^4\sigma^4\log\left(n\right)\eta\lambda_2\mathrm{Tr}\left(\Sigma\right)$$

*Proof.*

$$\eta\frac{\lambda_2\mathrm{Tr}\left(\Lambda_2\right)}{\lambda_1 - \lambda_2} = \kappa\frac{\log\left(n\right)}{n}\frac{\mathrm{Tr}\left(\Lambda_2\right)}{\lambda_1 - \lambda_2}\frac{\lambda_2}{\lambda_1 - \lambda_2} \leq \kappa c$$

Therefore, the first claim follows for $c \leq \frac{1}{\kappa C}$. For the second claim,

$$C\eta\lambda_1 = \frac{\kappa C \log(n) \lambda_1}{n(\lambda_1 - \lambda_2)} \leq \kappa Cc \frac{1}{\sqrt{n}} \leq \frac{1}{4}$$

where the last inequality holds for $c \leq \frac{1}{4\kappa C}$ and $n \geq 1$. Furthermore we have

$$C\eta \operatorname{Tr}(\Lambda_2) = C \frac{\kappa \log(n)}{n(\lambda_1 - \lambda_2)} \operatorname{Tr}(\Lambda_2) \leq \kappa Cc \leq \frac{1}{4}$$

where the last inequality holds for $\kappa Cc \leq \frac{1}{4}$. Note that $\eta C \leq \min\left\{\frac{1}{4\lambda_1}, \frac{1}{4} \frac{1}{\operatorname{Tr}(\Lambda_2)}\right\}$ imply $4\eta C \leq \frac{1}{\sqrt{\lambda_1 \operatorname{Tr}(\Lambda_2)}}$.

For the third claim, we have

$$C\eta^2 n\lambda_1^2 = \eta\lambda_1 \frac{\kappa C \log(n) \lambda_1}{(\lambda_1 - \lambda_2)} = \frac{\kappa^2 C \log^2(n)}{n}\left(\frac{\lambda_1}{\lambda_1 - \lambda_2}\right)^2 \leq \kappa^2 Cc^2 \leq \frac{1}{4}$$

where the last inequality holds when $c \leq \frac{1}{\sqrt{4\kappa^2 C}}$.

Next, we have the last claim,

$$\exp(-rn\eta(\lambda_1 - \lambda_2)) = \exp(-r\kappa \log(n)) = \frac{1}{n^{r\kappa}}$$

Therefore, it suffices to ensure

$$\frac{1}{n^{r\kappa}} \leq \frac{1}{n^{\frac{\kappa}{2}}} \leq \frac{\kappa \log(n)}{n} \overset{(iv)}{\leq} \frac{\kappa\lambda_1 \log(n)}{n(\lambda_1 - \lambda_2)}$$

where $(iv)$ follows since $\frac{\lambda_1}{(\lambda_1 - \lambda_2)} \geq 1$ as $\lambda_1 > \lambda_2$. Therefore, we require

$$\frac{1}{n^{\frac{\kappa}{2}}} \leq \frac{\kappa \log(n)}{n}$$

which holds for $\kappa = 2 + o(1)$ and sufficiently large $n$. $\qquad\square$

**Lemma A.2.5.** *For constants $c_1, c_2, c_3, c_4, c_5 > 0$, consider the following system of recursions -*

$$\alpha_n \leq \left(1 + c_1\eta\lambda_1 + c_2\eta^2\lambda_1^2\right)\alpha_{n-1} + c_3\eta^2\lambda_1\lambda_2\beta_{n-1},$$
$$\beta_n \leq \left(1 + c_1\eta\lambda_2 + c_4\eta^2\lambda_2 \operatorname{Tr}(\Sigma)\right)\beta_{n-1} + c_5\eta^2\lambda_1 \operatorname{Tr}(\Sigma)\alpha_{n-1}$$

*Let $\exists\theta \in (0.5, 1)$, which satisfies $c_1(1-\theta)(\lambda_1 - \lambda_2) + c_2\eta\lambda_1^2 = c_4\eta\lambda_2 \operatorname{Tr}(\Sigma)$ and*

$$\frac{4c_3c_5}{c_1^2}\eta^2\lambda_2 \operatorname{Tr}(\Sigma)\left(\frac{\lambda_1}{\theta(\lambda_1 - \lambda_2)}\right)^2 \leq 1, \quad 4\eta\lambda_1\left(\frac{c_2\lambda_1}{c_1(\lambda_1 - \lambda_2)}\right) \leq 1 - \theta$$

*Then we have,*

$$\alpha_n \leq \lambda_1(P)^n\left[\alpha_0 + \eta\lambda_1\left(\frac{2c_3\lambda_1}{c_1\theta(\lambda_1 - \lambda_2)}\right)\left(\beta_0 + \alpha_0\frac{c_5}{c_4}\left(\frac{1-\theta}{\theta}\right)\right)\right],$$
$$\beta_n \leq \beta_0\lambda_2(P)^n + \left[\eta\lambda_1\left(\frac{2c_5\lambda_1}{c_1\theta(\lambda_1 - \lambda_2)}\right)\left(\alpha_0\frac{\operatorname{Tr}(\Sigma)}{\lambda_1} + \beta_0\frac{c_3}{c_4}\left(\frac{1-\theta}{\theta}\right)\right)\right]\lambda_1(P)^n$$

*where*

$$\left|\lambda_1(P) - 1 - c_1\eta\lambda_1 - c_2\eta^2\lambda_1^2\right| \leq \frac{c_3c_5}{c_4}\eta^2\lambda_1^2\left(\frac{1-\theta}{\theta}\right)$$

$$\left|\lambda_2(P) - 1 - c_1\eta\lambda_2 - c_4\eta^2\lambda_2 \operatorname{Tr}(\Sigma)\right| \leq \frac{c_3c_5}{c_4}\eta^2\lambda_1^2\left(\frac{1-\theta}{\theta}\right)$$

*Proof.* Writing the recursions in a matrix form, we have

$$\begin{pmatrix} \alpha_n \\ \beta_n \end{pmatrix} = \begin{pmatrix} 1 + c_1\eta\lambda_1 + c_2\eta^2\lambda_1^2 & c_3\eta^2\lambda_1\lambda_2 \\ c_5\eta^2\lambda_1\,\mathrm{Tr}\,(\Sigma) & 1 + c_1\eta\lambda_2 + c_4\eta^2\lambda_2\,\mathrm{Tr}\,(\Sigma) \end{pmatrix} \begin{pmatrix} \alpha_{n-1} \\ \beta_{n-1} \end{pmatrix} \tag{A.17}$$

Define

$$P := \begin{pmatrix} 1 + c_1\eta\lambda_1 + c_2\eta^2\lambda_1^2 & c_3\eta^2\lambda_1\lambda_2 \\ c_5\eta^2\lambda_1\,\mathrm{Tr}\,(\Sigma) & 1 + c_1\eta\lambda_2 + c_4\eta^2\lambda_2\,\mathrm{Tr}\,(\Sigma) \end{pmatrix}$$

Then $P := I + c_1\eta M$, where

$$M := \begin{pmatrix} \lambda_1 + u\eta\lambda_1^2 & v\eta\lambda_1\lambda_2 \\ w\eta\lambda_1\,\mathrm{Tr}\,(\Sigma) & \lambda_2 + x\eta\lambda_2\,\mathrm{Tr}\,(\Sigma) \end{pmatrix}$$

and $u := \frac{c_2}{c_1}$, $v = \frac{c_3}{c_1}$, $x = \frac{c_4}{c_1}$, $w = \frac{c_5}{c_1}$. We now compute eigenvalues of $M$. The trace and determinants are given as -

$$T := \lambda_1 + \lambda_2 + u\eta\lambda_1^2 + x\eta\lambda_2\,\mathrm{Tr}\,(\Sigma)$$
$$D := \lambda_1\lambda_2 + \eta\lambda_1\lambda_2\left(x\,\mathrm{Tr}\,(\Sigma) + u\lambda_1\right) + ux\eta^2\lambda_1^2\lambda_2\,\mathrm{Tr}\,(\Sigma) - vw\eta^2\lambda_1^2\lambda_2\,\mathrm{Tr}\,(\Sigma)$$

Next we compute $\frac{T^2}{4} - D$,

$$\frac{T^2}{4} - D = \frac{(\lambda_1 - \lambda_2)^2}{4} + \eta\left(\frac{(\lambda_1 + \lambda_2)\left(u\lambda_1^2 + x\lambda_2\,\mathrm{Tr}\,(\Sigma)\right) - 2\lambda_1\lambda_2\left(x\,\mathrm{Tr}\,(\Sigma) + u\lambda_1\right)}{2}\right)$$

$$+ \frac{\eta^2\left(u\lambda_1^2 + x\lambda_2\,\mathrm{Tr}\,(\Sigma)\right)^2}{4} - (ux - vw)\eta^2\lambda_1^2\lambda_2\,\mathrm{Tr}\,(\Sigma)$$

$$= \left[\frac{(\lambda_1 - \lambda_2)^2}{4} - 2\left(\frac{x\eta\lambda_2\,\mathrm{Tr}\,(\Sigma)}{2}\right)\left(\frac{\lambda_1 - \lambda_2}{2}\right) + \left(\frac{x\eta\lambda_2\,\mathrm{Tr}\,(\Sigma)}{2}\right)^2\right]$$

$$+ \eta u\lambda_1^2\left(\frac{\lambda_1 - \lambda_2}{2} + \frac{\eta u\lambda_1^2}{4}\right) + \left(vw - \frac{ux}{2}\right)\eta^2\lambda_1^2\lambda_2\,\mathrm{Tr}\,(\Sigma)$$

$$= \frac{1}{4}\left((\lambda_1 - \lambda_2) - x\eta\lambda_2\,\mathrm{Tr}\,(\Sigma)\right)^2 + \frac{\eta u\lambda_1^2}{2}\left((\lambda_1 - \lambda_2) - x\eta\lambda_2\,\mathrm{Tr}\,(\Sigma)\right) + \frac{\eta^2\lambda_1^2}{4}\left(u^2\lambda_1^2 + 4vw\lambda_2\,\mathrm{Tr}\,(\Sigma)\right),$$

$$= \frac{1}{4}\left[(\lambda_1 - \lambda_2) - x\eta\lambda_2\,\mathrm{Tr}\,(\Sigma) + \eta u\lambda_1^2\right]^2 + vw\eta^2\lambda_1^2\lambda_2\,\mathrm{Tr}\,(\Sigma)$$

Let $(\lambda_1 - \lambda_2) - x\eta\lambda_2\,\mathrm{Tr}\,(\Sigma) + \eta u\lambda_1^2 = \theta\,(\lambda_1 - \lambda_2)$ for $\theta \in (0, 1)$,

$$\frac{T^2}{4} - D = \frac{\theta^2\,(\lambda_1 - \lambda_2)^2}{4} + vw\eta^2\lambda_1^2\lambda_2\,\mathrm{Tr}\,(\Sigma)$$

$$= \frac{\theta^2\,(\lambda_1 - \lambda_2)^2}{4}\left(1 + 4\eta^2\lambda_2\,\mathrm{Tr}\,(\Sigma)\frac{vw\lambda_1^2}{\theta^2\,(\lambda_1 - \lambda_2)^2}\right)$$

Let $\frac{\eta^2 vw\lambda_1^2\lambda_2\,\mathrm{Tr}(\Sigma)}{\theta^2(\lambda_1 - \lambda_2)^2} \leq \frac{1}{4}$. Then, using the identity $1 - \frac{x}{2} \leq \sqrt{1 + x} \leq 1 + \frac{x}{2}$ for $x \in (0, 1)$ we have,

$$\frac{\theta}{2}(\lambda_1 - \lambda_2)\left(1 - \frac{2\eta^2 vw\lambda_1^2\lambda_2\,\mathrm{Tr}\,(\Sigma)}{\theta^2\,(\lambda_1 - \lambda_2)^2}\right) \leq \sqrt{\frac{T^2}{4} - D} \leq \frac{\theta}{2}(\lambda_1 - \lambda_2)\left(1 + \frac{2\eta^2 vw\lambda_1^2\lambda_2\,\mathrm{Tr}\,(\Sigma)}{\theta^2\,(\lambda_1 - \lambda_2)^2}\right) \tag{A.18}$$

Let us simplify $\frac{\eta^2 vw\lambda_1^2\lambda_2\,\mathrm{Tr}(\Sigma)}{\theta^2(\lambda_1 - \lambda_2)^2}$ using the definition of $\theta$. We have

$$\frac{\eta^2 vw\lambda_1^2\lambda_2\,\mathrm{Tr}\,(\Sigma)}{\theta^2\,(\lambda_1 - \lambda_2)^2} = \frac{\eta vw\lambda_1^2}{x}\left(\frac{(1 - \theta)}{\theta^2\,(\lambda_1 - \lambda_2)} + \frac{\eta u\lambda_1^2}{\theta^2\,(\lambda_1 - \lambda_2)^2}\right)$$

Let $(1 - \theta) \geq \frac{4\eta u \lambda_1^2}{(\lambda_1 - \lambda_2)}$. Then,

$$\frac{\theta}{2}(\lambda_1 - \lambda_2) \times \frac{\eta^2 vw\lambda_1^2\lambda_2 \operatorname{Tr}(\Sigma)}{\theta^2(\lambda_1 - \lambda_2)^2} = \frac{\eta vw\lambda_1^2}{2x}\left(\frac{1-\theta}{\theta} + \frac{4\eta u\lambda_1^2}{\theta(\lambda_1 - \lambda_2)}\right)$$

$$= \frac{\eta vw\lambda_1^2}{2\theta x}\left(1 - \theta + \frac{4\eta u\lambda_1^2}{(\lambda_1 - \lambda_2)}\right)$$

$$\leq \eta\lambda_1^2\frac{vw}{x}\left(\frac{1-\theta}{\theta}\right) \tag{A.19}$$

Then, using Eq A.18 and A.19, the eigenvalues of $M$ are given as $\lambda_1(M) := \frac{T}{2} + \sqrt{\frac{T^2}{4} - D}$ and $\lambda_2(M) := \frac{T}{2} - \sqrt{\frac{T^2}{4} - D}$ such that

$$\left|\lambda_1(M) - \lambda_1 - u\eta\lambda_1^2\right|, \quad \left|\lambda_2(M) - \lambda_2 - x\eta\lambda_2\operatorname{Tr}(\Sigma)\right| \leq \eta\lambda_1^2\frac{vw}{x}\left(\frac{1-\theta}{\theta}\right)$$

The eigenvalues of $P$ are given as $\lambda_1(P) := 1 + c_1\eta\lambda_1(M)$ and $\lambda_2(P) := 1 + c_1\eta\lambda_2(M)$. Then we have,

$$\left|\lambda_1(P) - 1 - c_1\eta\lambda_1 - c_2\eta^2\lambda_1^2\right| = \left|\lambda_1(P) - P_{1,1}\right| \leq \frac{c_3c_5}{c_4}\eta^2\lambda_1^2\left(\frac{1-\theta}{\theta}\right) \tag{A.20}$$

$$\left|\lambda_2(P) - 1 - c_1\eta\lambda_2 - c_4\eta^2\lambda_2\operatorname{Tr}(\Sigma)\right| = \left|\lambda_2(P) - P_{2,2}\right| \leq \frac{c_3c_5}{c_4}\eta^2\lambda_1^2\left(\frac{1-\theta}{\theta}\right) \tag{A.21}$$

We then use the result from [Wil92] to compute $P^n$ and $\alpha_n, \beta_n$. To compute $P^n$, we first compute the matrices $X$ and $Y$ -

$$X = \frac{P - \lambda_2(P)I}{\lambda_1(P) - \lambda_2(P)} = \frac{1}{\lambda_1(P) - \lambda_2(P)}\begin{pmatrix} P_{1,1} - \lambda_2(P) & P_{1,2} \\ P_{2,1} & P_{2,2} - \lambda_2(P) \end{pmatrix},$$

$$Y = \frac{P - \lambda_1(P)I}{\lambda_2(P) - \lambda_1(P)} = \frac{1}{\lambda_1(P) - \lambda_2(P)}\begin{pmatrix} \lambda_1(P) - P_{1,1} & -P_{1,2} \\ -P_{2,1} & \lambda_1(P) - P_{2,2} \end{pmatrix}$$

Then, $P^n = \lambda_1(P)^n X + \lambda_2(P)^n Y$, which gives

$$P^n = \begin{pmatrix} P_{1,1}a_n - b_n & P_{1,2}a_n \\ P_{2,1}a_n & P_{2,2}a_n - b_n \end{pmatrix}$$

where

$$a_n := \left(\frac{\lambda_1(P)^n - \lambda_2(P)^n}{\lambda_1(P) - \lambda_2(P)}\right), \quad b_n := \left(\frac{\lambda_1(P)^n\lambda_2(P) - \lambda_1(P)\lambda_2(P)^n}{\lambda_1(P) - \lambda_2(P)}\right)$$

Therefore, for $y_0 = \begin{bmatrix} \alpha_0 \\ \beta_0 \end{bmatrix}$, we have

$$\alpha_n = e_1^T P^n y_0 = (\alpha_0 P_{1,1} + \beta_0 P_{1,2})\left(\frac{\lambda_1(P)^n - \lambda_2(P)^n}{\lambda_1(P) - \lambda_2(P)}\right) - \alpha_0\lambda_1(P)\lambda_2(P)\left(\frac{\lambda_1(P)^{n-1} - \lambda_2(P)^{n-1}}{\lambda_1(P) - \lambda_2(P)}\right),$$
$$\tag{A.22}$$

$$\beta_n = e_2^T P^n y_0 = (\alpha_0 P_{2,1} + \beta_0 P_{2,2})\left(\frac{\lambda_1(P)^n - \lambda_2(P)^n}{\lambda_1(P) - \lambda_2(P)}\right) - \beta_0\lambda_1(P)\lambda_2(P)\left(\frac{\lambda_1(P)^{n-1} - \lambda_2(P)^{n-1}}{\lambda_1(P) - \lambda_2(P)}\right)$$
$$\tag{A.23}$$

Therefore, using Eq A.20 and Eq A.21,

$$\alpha_n \leq \alpha_0 \lambda_1 (P)^n + \left( \beta_0 P_{1,2} + \alpha_0 \frac{c_3 c_5}{c_4} \eta^2 \lambda_1^2 \left( \frac{1-\theta}{\theta} \right) \right) \left( \frac{\lambda_1 (P)^n - \lambda_2 (P)^n}{\lambda_1 (P) - \lambda_2 (P)} \right)$$

$$= \alpha_0 \lambda_1 (P)^n + \left( \beta_0 c_3 \eta^2 \lambda_1 \lambda_2 + \alpha_0 \frac{c_3 c_5}{c_4} \eta^2 \lambda_1^2 \left( \frac{1-\theta}{\theta} \right) \right) \left( \frac{\lambda_1 (P)^n - \lambda_2 (P)^n}{\lambda_1 (P) - \lambda_2 (P)} \right), \quad \text{(A.24)}$$

$$\beta_n \leq \beta_0 \lambda_2 (P)^n + \left( \alpha_0 P_{2,1} + \beta_0 \frac{c_3 c_5}{c_4} \eta^2 \lambda_1^2 \left( \frac{1-\theta}{\theta} \right) \right) \left( \frac{\lambda_1 (P)^n - \lambda_2 (P)^n}{\lambda_1 (P) - \lambda_2 (P)} \right)$$

$$= \beta_0 \lambda_2 (P)^n + \left( \alpha_0 c_5 \eta^2 \lambda_1 \operatorname{Tr} (\Sigma) + \beta_0 \frac{c_3 c_5}{c_4} \eta^2 \lambda_1^2 \left( \frac{1-\theta}{\theta} \right) \right) \left( \frac{\lambda_1 (P)^n - \lambda_2 (P)^n}{\lambda_1 (P) - \lambda_2 (P)} \right)$$

$$\text{(A.25)}$$

Recall that using Eq A.20 and Eq A.21

$$|\lambda_1 (P) - \lambda_2 (P)| \geq |P_{1,1} - P_{2,2}| - \frac{2 c_3 c_5}{c_4} \eta^2 \lambda_1^2 \left( \frac{1-\theta}{\theta} \right)$$

$$= c_1 \eta \left( (\lambda_1 - \lambda_2) - x \eta \lambda_2 \operatorname{Tr} (\Sigma) + \eta u \lambda_1^2 \right) - \frac{2 c_3 c_5}{c_4} \eta^2 \lambda_1^2 \left( \frac{1-\theta}{\theta} \right)$$

$$= c_1 \theta \eta \left( \lambda_1 - \lambda_2 \right) - \frac{2 c_3 c_5}{c_4} \eta^2 \lambda_1^2 \left( \frac{1-\theta}{\theta} \right)$$

$$\geq \frac{1}{2} c_1 \theta \eta \left( \lambda_1 - \lambda_2 \right)$$

Substituting in Eq A.24 and Eq A.25 we have,

$$\alpha_n \leq \lambda_1 (P)^n \left[ \alpha_0 + \eta \lambda_1 \left( \frac{2 c_3 \lambda_1}{c_1 \theta (\lambda_1 - \lambda_2)} \right) \left( \beta_0 + \alpha_0 \frac{c_5}{c_4} \left( \frac{1-\theta}{\theta} \right) \right) \right],$$

$$\beta_n \leq \beta_0 \lambda_2 (P)^n + \left[ \eta \lambda_1 \left( \frac{2 c_5 \lambda_1}{c_1 \theta (\lambda_1 - \lambda_2)} \right) \left( \alpha_0 \frac{\operatorname{Tr} (\Sigma)}{\lambda_1} + \beta_0 \frac{c_3}{c_4} \left( \frac{1-\theta}{\theta} \right) \right) \right] \lambda_1 (P)^n$$

Hence proved. $\qquad\square$

**Lemma A.2.6.** *Let* $U \in \mathbb{R}^{d \times m}$ *then, for all* $t > 0$*, under subgaussianity (Definition 2.1) and the step-size* $\eta$ *satisfying* $(1 - \theta) (\lambda_1 - \lambda_2) + 2 L^4 \sigma^4 \eta \lambda_1^2 = 2 L^4 \sigma^4 \eta \lambda_2 \operatorname{Tr} (\Sigma)$ *for* $\theta \in \left( \frac{1}{2}, 1 \right)$ *then we have*

$$\lambda_1 \mathbb{E} \left[ U^T B_n^T v_1 v_1^T B_n U \right] \leq \gamma_1^n \left[ \alpha_0 + \eta \lambda_1 \left( \frac{2 \lambda_1}{\theta (\lambda_1 - \lambda_2)} \right) \left( \beta_0 + \alpha_0 \left( \frac{1-\theta}{\theta} \right) \right) \right],$$

$$\mathbb{E} \left[ \operatorname{Tr} \left( U^T B_n^T V_\perp \Lambda_2 V_\perp^T B_n U \right) \right] \leq \beta_0 \gamma_2^n + \left[ \eta \lambda_1 \left( \frac{2 \lambda_1}{\theta (\lambda_1 - \lambda_2)} \right) \left( \alpha_0 \frac{\operatorname{Tr} (\Sigma)}{\lambda_1} + \beta_0 \left( \frac{1-\theta}{\theta} \right) \right) \right] \gamma_1^n$$

*where* $B_n$ *is defined in Eq 3,* $\alpha_0 = \lambda_1 v_1^T U U^T v_1,$ $\beta_0 = \operatorname{Tr} \left( U^T V_\perp \Lambda_2 V_\perp^T U \right)$ *and*

$$\left| \gamma_1 - 1 - 2 \eta \lambda_1 - 4 L^4 \sigma^4 \eta^2 \lambda_1^2 \right| \leq 4 L^4 \sigma^4 \eta^2 \lambda_1^2 \left( \frac{1-\theta}{\theta} \right)$$

$$\left| \gamma_2 - 1 - 2 \eta \lambda_2 - 4 L^4 \sigma^4 \eta^2 \lambda_2 \operatorname{Tr} (\Sigma) \right| \leq 4 L^4 \sigma^4 \eta^2 \lambda_1^2 \left( \frac{1-\theta}{\theta} \right)$$

*Proof.* Let $\alpha_n := \lambda_1 \mathbb{E} \left[ \operatorname{Tr} \left( v_1^T B_n U U^T B_n^T v_1 \right) \right], \beta_n := \mathbb{E} \left[ \operatorname{Tr} \left( U^T B_n^T V_\perp \Lambda_2 V_\perp^T B_n U \right) \right]$ such that

$$\alpha_n + \beta_n = \mathbb{E} \left[ \operatorname{Tr} \left( U^T B_n^T \Sigma B_n U \right) \right]$$

Define $A_n := X_n X_n^T$ and let $\mathcal{F}_n$ denote the filtration for observations $i \in [n]$. Then,

$$
\begin{aligned}
\alpha_n &= \lambda_1 \mathbb{E} \left[ v_1^T B_n U U^T B_n^T v_1 \right] \\
&= \lambda_1 \mathbb{E} \left[ v_1^T \left( I + \eta A_n \right) B_{n-1} U U^T B_{n-1}^T \left( I + \eta A_n \right) v_1 \right] \\
&= \alpha_{n-1} + 2\eta \lambda_1 \mathbb{E} \left[ v_1^T A_n B_{n-1} U U^T B_{n-1}^T v_1 \right] + \eta^2 \lambda_1 \mathbb{E} \left[ v_1^T A_n B_{n-1} U U^T B_{n-1}^T A_n v_1 \right] \\
&= \alpha_{n-1} + 2\eta \lambda_1 \mathbb{E} \left[ v_1^T \Sigma B_{n-1} U U^T B_{n-1}^T v_1 \right] + \eta^2 \lambda_1 \mathbb{E} \left[ \left( v_1^T X_n X_n^T v_1 \right) \left( X_n^T B_{n-1} U U^T B_{n-1}^T X_n \right) \right] \\
&= \alpha_{n-1} \left( 1 + 2\eta \lambda_1 \right) + \eta^2 \lambda_1 \mathbb{E} \left[ \left( v_1^T X_n X_n^T v_1 \right) \left( X_n^T B_{n-1} U U^T B_{n-1}^T X_n \right) \right] \\
&= \alpha_{n-1} \left( 1 + 2\eta \lambda_1 \right) + \eta^2 \mathbb{E} \left[ \mathbb{E} \left[ \left( v_1^T X_n X_n^T v_1 \right) \left( X_n^T B_{n-1} U U^T B_{n-1}^T X_n \right) \Big| \mathcal{F}_{n-1} \right] \right] \\
&\le \alpha_{n-1} \left( 1 + 2\eta \lambda_1 \right) + \eta^2 \lambda_1 \mathbb{E} \left[ \sqrt{ \mathbb{E} \left[ \left( v_1^T X_n X_n^T v_1 \right)^2 \Big| \mathcal{F}_{n-1} \right] \mathbb{E} \left[ \left( X_n^T B_{n-1} U U^T B_{n-1}^T X_n \right)^2 \Big| \mathcal{F}_{n-1} \right] } \right] \\
&= \alpha_{n-1} \left( 1 + 2\eta \lambda_1 \right) + \eta^2 \lambda_1 \mathbb{E} \left[ \sqrt{ \mathbb{E} \left[ \left( X_n^T v_1 v_1^T X_n \right)^2 \Big| \mathcal{F}_{n-1} \right] \mathbb{E} \left[ \left( X_n^T B_{n-1} U U^T B_{n-1} X_n \right)^2 \Big| \mathcal{F}_{n-1} \right] } \right] \\
&\le \alpha_{n-1} \left( 1 + 2\eta \lambda_1 \right) + 4\eta^2 L^4 \sigma^4 \lambda_1 \operatorname{Tr} \left( \Sigma, v_1 v_1^T \right) \mathbb{E} \left[ \operatorname{Tr} \left( \Sigma, B_{n-1} U U^T B_{n-1}^T \right) \right], \text{ using Lemma A.2.2 with } p = 2 \\
&= \alpha_{n-1} \left( 1 + 2\eta \lambda_1 \right) + 4\eta^2 L^4 \sigma^4 \lambda_1^2 \left( \mathbb{E} \left[ \operatorname{Tr} \left( \lambda_1 v_1 v_1^T + V_\perp \Lambda_2 V_\perp^T, B_{n-1} U U^T B_{n-1}^T \right) \right] \right) \\
&= \left( 1 + 2\eta \lambda_1 + 4\eta^2 L^4 \sigma^4 \lambda_1^2 \right) \alpha_{n-1} + 4\eta^2 L^4 \sigma^4 \lambda_1^2 \beta_{n-1} \qquad\qquad (A.26)
\end{aligned}
$$

and similarly,

$$
\begin{aligned}
\beta_n &= \mathbb{E} \left[ \operatorname{Tr} \left( U^T B_n^T V_\perp \Lambda_2 V_\perp^T B_n U \right) \right] \\
&= \mathbb{E} \left[ \operatorname{Tr} \left( \Lambda_2^{\frac{1}{2}} V_\perp^T B_n U U^T B_n^T V_\perp \Lambda_2^{\frac{1}{2}} \right) \right] \\
&= \mathbb{E} \left[ \operatorname{Tr} \left( \Lambda_2^{\frac{1}{2}} V_\perp^T B_{n-1} U U^T B_{n-1}^T V_\perp \Lambda_2^{\frac{1}{2}} \right) \right] + 2\eta \mathbb{E} \left[ \operatorname{Tr} \left( \Lambda_2^{\frac{1}{2}} V_\perp^T A_n B_{n-1} U U^T B_{n-1}^T V_\perp \Lambda_2^{\frac{1}{2}} \right) \right] + \\
&\quad + \eta^2 \mathbb{E} \left[ \operatorname{Tr} \left( \Lambda_2^{\frac{1}{2}} V_\perp^T A_n B_{n-1} U U^T B_{n-1}^T A_n V_\perp \Lambda_2^{\frac{1}{2}} \right) \right] \\
&= \beta_{n-1} + 2\eta \mathbb{E} \left[ \operatorname{Tr} \left( \Lambda_2^{\frac{1}{2}} V_\perp^T \Sigma B_{n-1} U U^T B_{n-1}^T V_\perp \Lambda_2^{\frac{1}{2}} \right) \right] + \eta^2 \mathbb{E} \left[ \left( X_n^T V_\perp \Lambda_2 V_\perp^T X_n \right) \left( X_n^T B_{n-1} U U^T B_{n-1}^T X_n \right) \right] \\
&= \beta_{n-1} + 2\eta \mathbb{E} \left[ \operatorname{Tr} \left( \Lambda_2^2 V_\perp^T \Sigma B_{n-1} U U^T B_{n-1}^T V_\perp \right) \right] + \eta^2 \mathbb{E} \left[ \mathbb{E} \left[ \left( X_n^T V_\perp \Lambda_2 V_\perp^T X_n \right) \left( X_n^T B_{n-1} U U^T B_{n-1}^T X_n \right) \Big| \mathcal{F}_{n-1} \right] \right] \\
&\le \left( 1 + 2\eta \lambda_2 \right) \beta_{n-1} + \eta^2 \mathbb{E} \left[ \sqrt{ \mathbb{E} \left[ \left( X_n^T V_\perp \Lambda_2 V_\perp^T X_n \right)^2 \Big| \mathcal{F}_{n-1} \right] \mathbb{E} \left[ \left( X_n^T B_{n-1} U U^T B_{n-1}^T X_n \right)^2 \Big| \mathcal{F}_{n-1} \right] } \right] \\
&\le \left( 1 + 2\eta \lambda_2 \right) \beta_{n-1} + 4\eta^2 L^4 \sigma^4 \operatorname{Tr} \left( \Sigma V_\perp \Lambda_2 V_\perp^T \right) \mathbb{E} \left[ \operatorname{Tr} \left( \Sigma B_{n-1} U U^T B_{n-1}^T \right) \right], \text{ using Lemma A.2.2 with } p = 2 \\
&= \left( 1 + 2\eta \lambda_2 + 4\eta^2 L^4 \sigma^4 \operatorname{Tr} \left( \Lambda_2^2 \right) \right) \beta_{n-1} + 4\eta^2 L^4 \sigma^4 \operatorname{Tr} \left( \Lambda_2^2 \right) \alpha_{n-1} \\
&\le \left( 1 + 2\eta \lambda_2 + 4\eta^2 L^4 \sigma^4 \lambda_2 \operatorname{Tr} \left( \Lambda_2 \right) \right) \beta_{n-1} + 4\eta^2 L^4 \sigma^4 \operatorname{Tr} \left( \Lambda_2^2 \right) \alpha_{n-1} \qquad\qquad (A.27)
\end{aligned}
$$

Writing the recursions in a matrix form, we have

$$
\begin{pmatrix} \alpha_n \\ \beta_n \end{pmatrix} = \begin{pmatrix} 1 + 2\eta \lambda_1 + 4\eta^2 L^4 \sigma^4 \lambda_1^2 & 4\eta^2 L^4 \sigma^4 \lambda_1^2 \\ 4\eta^2 L^4 \sigma^4 \operatorname{Tr} \left( \Lambda_2^2 \right) & 1 + 2\eta \lambda_2 + 4\eta^2 L^4 \sigma^4 \lambda_2 \operatorname{Tr} \left( \Lambda_2 \right) \end{pmatrix} \begin{pmatrix} \alpha_{n-1} \\ \beta_{n-1} \end{pmatrix} \qquad (A.28)
$$

The result then follows by using Lemma A.2.5. $\qquad\qquad\square$

**Lemma A.2.7.** *Let $U \in \mathbb{R}^{d \times m}$ then, for all $t > 0$, under subgaussianity (Definition 2.1) and the step-size $\eta$ satisfying $(1 - \theta)(\lambda_1 - \lambda_2) + 2L^4 \sigma^4 \eta \lambda_1^2 = 2L^4 \sigma^4 \eta \lambda_2 \operatorname{Tr}(\Sigma)$ for $\theta \in \left( \frac{1}{2}, 1 \right)$ then we have*

$$
\mathbb{E} \left[ v_1^T B_n U U^T B_n^T v_1 \right] \le \gamma_1^n \left[ \alpha_0 + \eta \lambda_1 \left( \frac{2\lambda_1}{\theta (\lambda_1 - \lambda_2)} \right) \left( \beta_0 + \alpha_0 \left( \frac{1 - \theta}{\theta} \right) \right) \right],
$$

$$
\mathbb{E} \left[ \operatorname{Tr} \left( V_\perp^T B_n U U^T B_n^T V_\perp \right) \right] \le \beta_0 \gamma_2^n + \left[ \eta \lambda_1 \left( \frac{2\lambda_1}{\theta (\lambda_1 - \lambda_2)} \right) \left( \alpha_0 \frac{\operatorname{Tr}(\Sigma)}{\lambda_1} + \beta_0 \left( \frac{1 - \theta}{\theta} \right) \right) \right] \gamma_1^n
$$

*where $B_n$ is defined in Eq 3, $\alpha_0 = v_1^T U U^T v_1$, $\beta_0 = \mathrm{Tr}\left(V_\perp^T U U^T V_\perp\right)$ and*

$$\left|\gamma_1 - 1 - 2\eta\lambda_1 - 4L^4\sigma^4\eta^2\lambda_1^2\right| \le 4L^4\sigma^4\eta^2\lambda_1^2\left(\frac{1-\theta}{\theta}\right)$$

$$\left|\gamma_2 - 1 - 2\eta\lambda_2 - 4L^4\sigma^4\eta^2\lambda_2\,\mathrm{Tr}\left(\Sigma\right)\right| \le 4L^4\sigma^4\eta^2\lambda_1^2\left(\frac{1-\theta}{\theta}\right)$$

*Proof.* Let $\alpha_n := \mathbb{E}\left[\mathrm{Tr}\left(v_1^T B_n U U^T B_n^T v_1\right)\right]$, $\beta_n := \mathbb{E}\left[\mathrm{Tr}\left(V_\perp^T B_n U U^T B_n^T V_\perp\right)\right]$. Define $A_n := X_n X_n^T$ and let $\mathcal{F}_n$ denote the filtration for observations $i \in [n]$. Then,

$$
\begin{aligned}
\alpha_n &= \mathbb{E}\left[v_1^T B_n U U^T B_n^T v_1\right]\\
&= \mathbb{E}\left[v_1^T \left(I + \eta A_n\right) B_{n-1} U U^T B_{n-1}^T \left(I + \eta A_n\right) v_1\right]\\
&= \alpha_{n-1} + 2\eta\mathbb{E}\left[v_1^T A_n B_{n-1} U U^T B_{n-1}^T v_1\right] + \eta^2\mathbb{E}\left[v_1^T A_n B_{n-1} U U^T B_{n-1}^T A_n v_1\right]\\
&= \alpha_{n-1} + 2\eta\mathbb{E}\left[v_1^T \Sigma B_{n-1} U U^T B_{n-1}^T v_1\right] + \eta^2\mathbb{E}\left[\left(v_1^T X_n X_n^T v_1\right)\left(X_n^T B_{n-1} U U^T B_{n-1}^T X_n\right)\right]\\
&= \alpha_{n-1}\left(1 + 2\eta\lambda_1\right) + \eta^2\mathbb{E}\left[\left(v_1^T X_n X_n^T v_1\right)\left(X_n^T B_{n-1} U U^T B_{n-1}^T X_n\right)\right]\\
&= \alpha_{n-1}\left(1 + 2\eta\lambda_1\right) + \eta^2\mathbb{E}\left[\mathbb{E}\left[\left(v_1^T X_n X_n^T v_1\right)\left(X_n^T B_{n-1} U U^T B_{n-1}^T X_n\right)\Big|\mathcal{F}_{n-1}\right]\right]\\
&\le \alpha_{n-1}\left(1 + 2\eta\lambda_1\right) + \eta^2\mathbb{E}\left[\sqrt{\mathbb{E}\left[\left(v_1^T X_n X_n^T v_1\right)^2\Big|\mathcal{F}_{n-1}\right]\mathbb{E}\left[\left(X_n^T B_{n-1} U U^T B_{n-1}^T X_n\right)^2\Big|\mathcal{F}_{n-1}\right]}\right]\\
&= \alpha_{n-1}\left(1 + 2\eta\lambda_1\right) + \eta^2\mathbb{E}\left[\sqrt{\mathbb{E}\left[\left(X_n^T v_1 v_1^T X_n\right)^2\Big|\mathcal{F}_{n-1}\right]\mathbb{E}\left[\left(X_n^T B_{n-1} U U^T B_{n-1} X_n\right)^2\Big|\mathcal{F}_{n-1}\right]}\right]\\
&\le \alpha_{n-1}\left(1 + 2\eta\lambda_1\right) + 4\eta^2 L^4\sigma^4\,\mathrm{Tr}\left(\Sigma, v_1 v_1^T\right)\mathbb{E}\left[\mathrm{Tr}\left(\Sigma, B_{n-1} U U^T B_{n-1}^T\right)\right], \text{ using Lemma A.2.2 with } p = 2\\
&= \alpha_{n-1}\left(1 + 2\eta\lambda_1\right) + 4\eta^2 L^4\sigma^4\lambda_1\left(\mathbb{E}\left[\mathrm{Tr}\left(\lambda_1 v_1 v_1^T + V_\perp \Lambda_2 V_\perp^T, B_{n-1} U U^T B_{n-1}^T\right)\right]\right)\\
&\le \left(1 + 2\eta\lambda_1 + 4\eta^2 L^4\sigma^4\lambda_1^2\right)\alpha_{n-1} + 4\eta^2 L^4\sigma^4\lambda_1\lambda_2\beta_{n-1} \quad\quad\quad\text{(A.29)}
\end{aligned}
$$

and similarly,

$$
\begin{aligned}
\beta_n &= \mathbb{E}\left[\mathrm{Tr}\left(V_\perp^T B_n U U^T B_n^T V_\perp\right)\right]\\
&= \mathbb{E}\left[\mathrm{Tr}\left(V_\perp^T \left(I + \eta A_n\right) B_{n-1} U U^T B_{n-1}^T \left(I + \eta A_n\right) V_\perp\right)\right]\\
&= \mathbb{E}\left[\mathrm{Tr}\left(V_\perp^T B_{n-1} U U^T B_{n-1}^T V_\perp\right)\right] + 2\eta\mathbb{E}\left[\mathrm{Tr}\left(V_\perp^T A_n B_{n-1} U U^T B_{n-1}^T V_\perp\right)\right] +\\
&\quad + \eta^2\mathbb{E}\left[\mathrm{Tr}\left(V_\perp^T A_n B_{n-1} U U^T B_{n-1}^T A_n V_\perp\right)\right]\\
&= \beta_{n-1} + 2\eta\mathbb{E}\left[\mathrm{Tr}\left(V_\perp^T \Sigma B_{n-1} U U^T B_{n-1}^T V_\perp\right)\right] + \eta^2\mathbb{E}\left[\left(X_n^T V_\perp V_\perp^T X_n\right)\left(X_n^T B_{n-1} U U^T B_{n-1}^T X_n\right)\right]\\
&= \beta_{n-1} + 2\eta\mathbb{E}\left[\mathrm{Tr}\left(V_\perp^T \Sigma B_{n-1} U U^T B_{n-1}^T V_\perp\right)\right] + \eta^2\mathbb{E}\left[\mathbb{E}\left[\left(X_n^T V_\perp V_\perp^T X_n\right)\left(X_n^T B_{n-1} U U^T B_{n-1}^T X_n\right)\Big|\mathcal{F}_{n-1}\right]\right]\\
&\le \left(1 + 2\eta\lambda_2\right)\beta_{n-1} + \eta^2\mathbb{E}\left[\sqrt{\mathbb{E}\left[\left(X_n^T V_\perp V_\perp^T X_n\right)^2\Big|\mathcal{F}_{n-1}\right]\mathbb{E}\left[\left(X_n^T B_{n-1} U U^T B_{n-1}^T X_n\right)^2\Big|\mathcal{F}_{n-1}\right]}\right]\\
&\le \left(1 + 2\eta\lambda_2\right)\beta_{n-1} + 4\eta^2 L^4\sigma^4\,\mathrm{Tr}\left(\Sigma V_\perp V_\perp^T\right)\mathbb{E}\left[\mathrm{Tr}\left(\Sigma B_{n-1} U U^T B_{n-1}^T\right)\right], \text{ using Lemma A.2.2 with } p = 2\\
&= \left(1 + 2\eta\lambda_2 + 4\eta^2 L^4\sigma^4\lambda_2\,\mathrm{Tr}\left(\Lambda_2\right)\right)\beta_{n-1} + 4\eta^2 L^4\sigma^4\lambda_1\,\mathrm{Tr}\left(\Lambda_2\right)\alpha_{n-1} \quad\quad\quad\text{(A.30)}
\end{aligned}
$$

The result then follows by using Lemma A.2.5. $\qquad\square$

**Lemma A.2.8.** *For all $t > 0$, under subgaussianity (Definition 2.1), let $U \in \mathbb{R}^{d\times m}$. Let the step-size $\eta$ be set according to Lemma A.2.4 then we have,*

$$\mathbb{E}\left[\left(v_1^T B_n U U^T B_n^T v_1\right)^2\right] \le \mu_1^{2n}\left[\alpha_0 + \eta\lambda_1\left(\frac{2\lambda_1}{\theta\left(\lambda_1 - \lambda_2\right)}\right)\left(\beta_0 + \alpha_0\left(\frac{1-\theta}{\theta}\right)\right)\right]^2,$$

$$\mathbb{E}\left[\mathrm{Tr}\left(V_\perp^T B_n U U^T B_n^T V_\perp\right)^2\right] \le \left(\beta_0 \mu_2^n + \left[\eta\lambda_1\left(\frac{2\lambda_1}{\theta\left(\lambda_1 - \lambda_2\right)}\right)\left(\alpha_0\frac{\mathrm{Tr}\left(\Sigma\right)}{\lambda_1} + \beta_0\left(\frac{1-\theta}{\theta}\right)\right)\right]\mu_1^n\right)^2$$

*where $B_n$ is defined in Eq 3, $\alpha_0 = v_1^T U U^T v_1$, $\beta_0 = \text{Tr}\left(V_\perp^T U U^T V_\perp\right)$ and*

$$\left|\mu_1 - 1 - 2\eta\lambda_1 - 50\eta^2 L^4 \sigma^4 \lambda_1^2\right| \leq 50 L^4 \sigma^4 \eta^2 \lambda_1^2 \left(\frac{1-\theta}{\theta}\right)$$

$$\left|\mu_2 - 1 - 2\eta\lambda_2 - 50\eta^2 L^4 \sigma^4 \lambda_2 \text{Tr}\left(\Sigma\right)\right| \leq 50 L^4 \sigma^4 \eta^2 \lambda_1^2 \left(\frac{1-\theta}{\theta}\right)$$

*Proof.* Let $\alpha_n := \mathbb{E}\left[\left(v_1^T B_n U U^T B_n^T v_1\right)^2\right]$, $\beta_n := \mathbb{E}\left[\text{Tr}\left(V_\perp^T B_n U U^T B_n^T V_\perp\right)^2\right]$. Then, using Lemma A.2.9 we have,

$$\alpha_n \leq \left(1 + 4\eta\lambda_1 + 100\eta^2 L^4 \sigma^4 \lambda_1^2\right)\alpha_{n-1} + 100\eta^2 L^4 \sigma^4 \lambda_1^2 \sqrt{\alpha_{n-1}\beta_{n-1}} + 600\eta^4 L^8 \sigma^8 \lambda_1^4 \beta_{n-1}$$

and using Lemma A.2.10 we have,

$$\beta_n \leq \left(1 + 4\eta\lambda_2 + 100\eta^2 L^4 \sigma^4 \lambda_2 \text{Tr}\left(\Sigma\right)\right)\beta_{n-1} + 100\eta^2 L^2 \sigma^2 \lambda_1 \text{Tr}\left(\Sigma\right)\sqrt{\alpha_{n-1}\beta_{n-1}}$$
$$+ 600\eta^2 L^4 \sigma^4 \lambda_1^2 \alpha_{n-1}$$

Define $a_n := \sqrt{\alpha_n}$ and $b_n := \sqrt{\beta_n}$. Then using $\sqrt{1+x} \leq 1 + \frac{x}{2}$, we have

$$a_n \leq \sqrt{1 + 4\eta\lambda_1 + 100\eta^2 L^4 \sigma^4 \lambda_1^2} a_{n-1} + 25\eta^2 L^2 \sigma^2 \lambda_1^2 b_{n-1},$$
$$\leq \left(1 + 2\eta\lambda_1 + 50\eta^2 L^4 \sigma^4 \lambda_1^2\right)a_{n-1} + 25\eta^2 L^2 \sigma^2 \lambda_1^2 b_{n-1}$$
$$b_n \leq \sqrt{1 + 4\eta\lambda_2 + 100\eta^2 L^4 \sigma^4 \lambda_2 \text{Tr}\left(\Sigma\right)} b_{n-1} + 25\eta^2 L^2 \sigma^2 \lambda_1^2 a_{n-1},$$
$$\leq \left(1 + 2\eta\lambda_2 + 50\eta^2 L^4 \sigma^4 \lambda_2 \text{Tr}\left(\Sigma\right)\right)b_{n-1} + 25\eta^2 L^2 \sigma^2 \lambda_1^2 a_{n-1}$$

The result then follows from Lemma A.2.5. $\qquad\square$

**Lemma A.2.9.** *For all $t > 0$, under subgaussianity (Definition 2.1), let $U \in \mathbb{R}^{d \times m}$, $\alpha_n := \mathbb{E}\left[\left(v_1^T B_n U U^T B_n^T v_1\right)^2\right]$, $\beta_n := \mathbb{E}\left[\text{Tr}\left(V_\perp^T B_n U U^T B_n^T V_\perp\right)^2\right]$ and $\eta L^2 \sigma^2 \lambda_1 \leq \frac{1}{4}$ then*

$$\alpha_n \leq \left(1 + 4\eta\lambda_1 + 100\eta^2 L^4 \sigma^4 \lambda_1^2\right)\alpha_{n-1} + 100\eta^2 L^4 \sigma^4 \lambda_1^2 \sqrt{\alpha_{n-1}\beta_{n-1}} + 600\eta^4 L^8 \sigma^8 \lambda_1^4 \beta_{n-1}$$

*where $B_n$ is defined in Eq 3.*

*Proof.* Let $A_n := X_n X_n^T$ and $\mathcal{F}_n$ denote the filtration for observations $i \in [n]$. Then,

$$\alpha_n = \mathbb{E}\left[\left(v_1^T B_n U U^T B_n^T v_1\right)^2\right]$$
$$= \mathbb{E}\left[\left(v_1^T B_{n-1} U U^T B_{n-1}^T v_1 + 2\eta v_1^T A_n B_{n-1} U U^T B_{n-1}^T v_1 + \eta^2 v_1^T A_n B_{n-1} U U^T B_{n-1}^T A_n v_1\right)^2\right]$$
$$= \alpha_{n-1}\left(1 + 4\eta\lambda_1\right) + 4\eta^2 \underbrace{\mathbb{E}\left[\left(v_1^T A_n B_{n-1} U U^T B_{n-1}^T v_1\right)^2\right]}_{T_1}$$
$$+ 2\eta^2 \underbrace{\mathbb{E}\left[\left(v_1^T B_{n-1} U U^T B_{n-1}^T v_1\right)\left(v_1^T A_n B_{n-1} U U^T B_{n-1}^T A_n v_1\right)\right]}_{T_2}$$
$$+ 4\eta^3 \underbrace{\mathbb{E}\left[\left(v_1^T A_n B_{n-1} U U^T B_{n-1}^T v_1\right)\left(v_1^T A_n B_{n-1} U U^T B_{n-1}^T A_n v_1\right)\right]}_{T_3}$$
$$+ \eta^4 \underbrace{\mathbb{E}\left[\left(v_1^T A_n B_{n-1} U U^T B_{n-1}^T A_n v_1\right)^2\right]}_{T_4} \qquad (A.31)$$

For $T_1$,

$$
\begin{aligned}
T_1 &= \mathbb{E}\left[\left(v_1^T A_n B_{n-1} U U^T B_{n-1}^T v_1\right)^2\right] \\
&= \mathbb{E}\left[\left(X_n^T v_1 v_1^T X_n\right)\left(X_n^T B_{n-1} U U^T B_{n-1}^T v_1 v_1^T B_{n-1} U U^T B_{n-1}^T X_n\right)\right] \\
&= \mathbb{E}\left[\mathbb{E}\left[\left(X_n^T v_1 v_1^T X_n\right)\left(X_n^T B_{n-1} U U^T B_{n-1}^T v_1 v_1^T B_{n-1} U U^T B_{n-1}^T X_n\right)\bigg|\mathcal{F}_{n-1}\right]\right] \\
&\le \mathbb{E}\left[\sqrt{\mathbb{E}\left[\left(X_n^T v_1 v_1^T X_n\right)^2\bigg|\mathcal{F}_{n-1}\right]\mathbb{E}\left[\left(X_n^T B_{n-1} U U^T B_{n-1}^T v_1 v_1^T B_{n-1} U U^T B_{n-1}^T X_n\right)^2\bigg|\mathcal{F}_{n-1}\right]}\right] \\
&\le 4L^4\sigma^4 \operatorname{Tr}\left(\Sigma v_1 v_1^T\right)\mathbb{E}\left[\operatorname{Tr}\left(\Sigma B_{n-1} U U^T B_{n-1}^T v_1 v_1^T B_{n-1} U U^T B_{n-1}^T\right)\right], \text{ using Lemma } A.2.2 \text{ with } p=2 \\
&= 4L^4\sigma^4\lambda_1^2\alpha_{n-1} + 4L^4\sigma^4\lambda_1\mathbb{E}\left[\operatorname{Tr}\left(V_\perp\Lambda_2 V_\perp^T B_{n-1} U U^T B_{n-1}^T v_1 v_1^T B_{n-1} U U^T B_{n-1}^T\right)\right] \\
&\le 4L^4\sigma^4\lambda_1^2\alpha_{n-1} + 4L^4\sigma^4\lambda_1\lambda_2\mathbb{E}\left[\operatorname{Tr}\left(V_\perp V_\perp^T B_{n-1} U U^T B_{n-1}^T v_1 v_1^T B_{n-1} U U^T B_{n-1}^T\right)\right] \\
&= 4L^4\sigma^4\lambda_1^2\alpha_{n-1} + 4L^4\sigma^4\lambda_1\lambda_2\mathbb{E}\left[v_1^T B_{n-1} U U^T B_{n-1}^T V_\perp V_\perp^T B_{n-1} U U^T B_{n-1}^T v_1\right] \\
&\le 4L^4\sigma^4\lambda_1^2\alpha_{n-1} + 4L^4\sigma^4\lambda_1\lambda_2\mathbb{E}\left[\left(v_1^T B_{n-1} U U^T B_{n-1}^T v_1\right)\operatorname{Tr}\left(U^T B_{n-1}^T V_\perp V_\perp^T B_{n-1} U\right)\right] \\
&\le 4L^4\sigma^4\lambda_1^2\alpha_{n-1} + 4L^4\sigma^4\lambda_1\lambda_2\sqrt{\alpha_{n-1}\beta_{n-1}}
\end{aligned}
$$

For $T_2$,

$$
\begin{aligned}
T_2 &= \mathbb{E}\left[\left(v_1^T B_{n-1} U U^T B_{n-1}^T v_1\right)\left(v_1^T A_n B_{n-1} U U^T B_{n-1}^T A_n v_1\right)\right] \\
&= \mathbb{E}\left[\left(v_1^T B_{n-1} U U^T B_{n-1}^T v_1\right)\mathbb{E}\left[\left(v_1^T A_n B_{n-1} U U^T B_{n-1}^T A_n v_1\right)|\mathcal{F}_{n-1}\right]\right] \\
&= \mathbb{E}\left[\left(v_1^T B_{n-1} U U^T B_{n-1}^T v_1\right)\mathbb{E}\left[\left(X_n^T v_1 v_1^T X_n\right)\left(X_n^T B_{n-1} U U^T B_{n-1}^T X_n\right)|\mathcal{F}_{n-1}\right]\right] \\
&\le \mathbb{E}\left[\left(v_1^T B_{n-1} U U^T B_{n-1}^T v_1\right)\sqrt{\mathbb{E}\left[\left(X_n^T v_1 v_1^T X_n\right)^2\bigg|\mathcal{F}_{n-1}\right]\mathbb{E}\left[\left(X_n^T B_{n-1} U U^T B_{n-1}^T X_n\right)^2\bigg|\mathcal{F}_{n-1}\right]}\right] \\
&\le 4L^4\sigma^4\mathbb{E}\left[\left(v_1^T B_{n-1} U U^T B_{n-1}^T v_1\right)\operatorname{Tr}\left(\Sigma v_1 v_1^T\right)\operatorname{Tr}\left(\Sigma B_{n-1} U U^T B_{n-1}^T\right)\right], \text{ using Lemma } A.2.2 \text{ with } p=2 \\
&= 4L^4\sigma^4\lambda_1^2\alpha_{n-1} + 4L^4\sigma^4\lambda_1\mathbb{E}\left[\left(v_1^T B_{n-1} U U^T B_{n-1}^T v_1\right)\operatorname{Tr}\left(V_\perp\Lambda_2 V_\perp^T B_{n-1} U U^T B_{n-1}^T\right)\right] \\
&\le 4L^4\sigma^4\lambda_1^2\alpha_{n-1} + 4L^4\sigma^4\lambda_1\lambda_2\mathbb{E}\left[\left(v_1^T B_{n-1} U U^T B_{n-1}^T v_1\right)\operatorname{Tr}\left(V_\perp V_\perp^T B_{n-1} U U^T B_{n-1}^T\right)\right] \\
&\le 4L^4\sigma^4\lambda_1^2\alpha_{n-1} + 4L^4\sigma^4\lambda_1\lambda_2\sqrt{\alpha_{n-1}\beta_{n-1}}
\end{aligned}
$$

For $T_3$,

$$
\begin{aligned}
T_3 &= \mathbb{E}\left[\left(v_1^T A_n B_{n-1} U U^T B_{n-1}^T v_1\right)\left(v_1^T A_n B_{n-1} U U^T B_{n-1}^T A_n v_1\right)\right] \\
&= \mathbb{E}\left[\left(X_n^T v_1\right)^3\left(X_n^T B_{n-1} U U^T B_{n-1}^T v_1\right)\left(X_n^T B_{n-1} U U^T B_{n-1}^T X_n\right)\right] \\
&\le \mathbb{E}\left[\left(X_n^T v_1 v_1^T X_n\right)^{\frac{3}{2}}\left(X_n^T B_{n-1} U U^T B_{n-1}^T X_n\right)^{\frac{3}{2}}\left\|U^T B_{n-1}^T v_1\right\|_2\right] \\
&= \mathbb{E}\left[\mathbb{E}\left[\left(X_n^T v_1 v_1^T X_n\right)^{\frac{3}{2}}\left(X_n^T B_{n-1} U U^T B_{n-1}^T X_n\right)^{\frac{3}{2}}\bigg|\mathcal{F}_{n-1}\right]\left\|U^T B_{n-1}^T v_1\right\|_2\right] \\
&\le \mathbb{E}\left[\sqrt{\mathbb{E}\left[\left(X_n^T v_1 v_1^T X_n\right)^3\bigg|\mathcal{F}_{n-1}\right]}\sqrt{\mathbb{E}\left[\left(X_n^T B_{n-1} U U^T B_{n-1}^T X_n\right)^3\bigg|\mathcal{F}_{n-1}\right]}\left\|U^T B_{n-1}^T v_1\right\|_2\right] \\
&\le \left(3L^2\sigma^2\right)^3\lambda_1^{\frac{3}{2}}\mathbb{E}\left[\operatorname{Tr}\left(B_{n-1} U U^T B_{n-1}^T\Sigma\right)^{\frac{3}{2}}\left(v_1^T B_{n-1} U U^T B_{n-1}^T v_1\right)^{\frac{1}{2}}\right], \text{ using Lemma } A.2.2 \text{ with } p=3 \\
&\le 2\left(3L^2\sigma^2\right)^3\lambda_1^{\frac{3}{2}}\left(\lambda_1^{\frac{3}{2}}\alpha_{n-1} + \lambda_2^{\frac{3}{2}}\mathbb{E}\left[\left(v_1^T B_{n-1} U U^T B_{n-1}^T v_1\right)^{\frac{1}{2}}\operatorname{Tr}\left(V_\perp^T B_{n-1} U U^T B_{n-1}^T V_\perp\right)^{\frac{3}{2}}\right]\right)
\end{aligned}
$$

$$= 2 \left(3L^2\sigma^2\right)^3 \lambda_1^3 \alpha_{n-1} + 2 \left(3L^2\sigma^2\right)^3 \times$$

$$\mathbb{E}\left[\frac{\sqrt{\lambda_1\lambda_2 \left\|U^T B_{n-1}^T v_1\right\|_2^2 \mathrm{Tr}\left(V_\perp^T B_{n-1} U U^T B_{n-1}^T V_\perp\right)}}{\sqrt{3\eta L^2\sigma^2}} \sqrt{3\eta L^2\sigma^2} \lambda_1\lambda_2 \mathrm{Tr}\left(V_\perp^T B_{n-1} U U^T B_{n-1}^T V_\perp\right)\right]$$

$$\leq 2 \left(3L^2\sigma^2\right)^3 \lambda_1^3 \alpha_{n-1} + \left(3L^2\sigma^2\right)^3 \mathbb{E}\left[\frac{\lambda_1\lambda_2 v_1^T B_{n-1} U U^T B_{n-1}^T v_1 \mathrm{Tr}\left(V_\perp^T B_{n-1} U U^T B_{n-1}^T V_\perp\right)}{3\eta L^2\sigma^2}\right]$$

$$+ \eta \left(3L^2\sigma^2\right)^4 \mathbb{E}\left[\lambda_1^2\lambda_2^2 \mathrm{Tr}\left(V_\perp^T B_{n-1} U U^T B_{n-1}^T V_\perp\right)^2\right]$$

$$= 2 \left(3L^2\sigma^2\right)^3 \lambda_1^3 \alpha_{n-1} + \frac{\left(3L^2\sigma^2\right)^2 \lambda_1\lambda_2}{\eta} \mathbb{E}\left[v_1^T B_{n-1} U U^T B_{n-1}^T v_1 \mathrm{Tr}\left(V_\perp^T B_{n-1} U U^T B_{n-1}^T V_\perp\right)\right]$$

$$+ \eta \left(3L^2\sigma^2\right)^4 \lambda_1^2\lambda_2^2 \mathbb{E}\left[\mathrm{Tr}\left(V_\perp^T B_{n-1} U U^T B_{n-1}^T V_\perp\right)^2\right]$$

$$\leq 2 \left(3L^2\sigma^2\right)^3 \lambda_1^3 \alpha_{n-1} + \frac{\left(3L^2\sigma^2\right)^2 \lambda_1\lambda_2}{\eta} \sqrt{\alpha_{n-1}\beta_{n-1}} + \eta \left(3L^2\sigma^2\right)^4 \lambda_1^2\lambda_2^2 \beta_{n-1}$$

For $T_4$,

$$T_4 = \mathbb{E}\left[\left(v_1^T A_n B_{n-1} U U^T B_{n-1}^T A_n v_1\right)^2\right]$$

$$= \mathbb{E}\left[\left(X_n^T v_1 v_1^T X_n\right)^2 \left(X_n^T B_{n-1} U U^T B_{n-1}^T X_n\right)^2\right]$$

$$= \mathbb{E}\left[\mathbb{E}\left[\left(X_n^T v_1 v_1^T X_n\right)^2 \left(X_n^T B_{n-1} U U^T B_{n-1}^T X_n\right)^2 \Big| \mathcal{F}_{n-1}\right]\right]$$

$$\leq \left(4L^2\sigma^2\right)^4 \mathbb{E}\left[\sqrt{\mathbb{E}\left[\left(v_1^T X_n X_n^T v_1\right)^4 \Big| \mathcal{F}_{n-1}\right] \mathbb{E}\left[\mathrm{Tr}\left(B_{n-1} U U^T B_{n-1}^T \Sigma\right)^4 \Big| \mathcal{F}_{n-1}\right]}\right]$$

$$\leq \left(4L^2\sigma^2\right)^4 \mathbb{E}\left[\sqrt{\left(v_1^T \Sigma v_1\right)^4 \mathbb{E}\left[\mathrm{Tr}\left(B_{n-1} U U^T B_{n-1}^T \Sigma\right)^4 \Big| \mathcal{F}_{n-1}\right]}\right], \text{ using Lemma } A.2.2 \text{ with } p = 4$$

$$= \left(4L^2\sigma^2\right)^4 \lambda_1^2 \mathbb{E}\left[\sqrt{\left(\mathrm{Tr}\left(B_{n-1} U U^T B_{n-1}^T \Sigma\right)\right)^4}\right]$$

$$= \left(4L^2\sigma^2\right)^4 \lambda_1^2 \mathbb{E}\left[\left(\mathrm{Tr}\left(B_{n-1} U U^T B_{n-1}^T V_\perp \Lambda_2 V_\perp^T\right) + \lambda_1 \mathrm{Tr}\left(B_{n-1} U U^T B_{n-1}^T v_1 v_1^T\right)\right)^2\right]$$

$$\leq 2 \left(4L^2\sigma^2\right)^4 \lambda_1^2 \left(\lambda_2^2 \beta_{n-1} + \lambda_1^2 \alpha_{n-1}\right)$$

Substituting in Eq A.31 along with using $\eta L^2\sigma^2\lambda_1 \leq \frac{1}{4}$ we have,

$$\alpha_n \leq \left(1 + 4\eta\lambda_1 + 100\eta^2 L^4\sigma^4\lambda_1^2\right)\alpha_{n-1} + 100\eta^2 L^4\sigma^4\lambda_1^2 \sqrt{\alpha_{n-1}\beta_{n-1}} + 600\eta^4 L^8\sigma^8\lambda_1^4\beta_{n-1}$$

Hence proved. $\qquad\square$

**Lemma A.2.10.** *For all $t > 0$, under subgaussianity (Definition 2.1), let $U \in \mathbb{R}^{d\times m}$, $\alpha_n := \mathbb{E}\left[\left(v_1^T B_n U U^T B_n^T v_1\right)^2\right]$, $\beta_n := \mathbb{E}\left[\mathrm{Tr}\left(V_\perp^T B_n U U^T B_n^T V_\perp\right)^2\right]$ and $\eta L^2\sigma^2 \leq \frac{1}{4}\min\left\{\frac{1}{\lambda_1}, \frac{1}{\mathrm{Tr}(\Sigma)}, \frac{1}{\sqrt{\lambda_1 \mathrm{Tr}(\Sigma)}}\right\}$ then*

$$\beta_n \leq \left(1 + 4\eta\lambda_2 + 100\eta^2 \log(n) L^4\sigma^4\lambda_2 \mathrm{Tr}(\Sigma)\right)\beta_{n-1} + 100\eta^2 \log(n) L^2\sigma^2\lambda_1 \mathrm{Tr}(\Sigma) \sqrt{\alpha_{n-1}\beta_{n-1}}$$
$$+ 600\eta^2 \log^2(n) L^4\sigma^4\lambda_1^2\alpha_{n-1}$$

*where $B_n$ is defined in Eq 3.*

*Proof.* Let $A_n := X_n X_n^T$ and $\mathcal{F}_n$ denote the filtration for observations $i \in [n]$.

Let

$$a_{n-1} := \mathrm{Tr}\left(V_\perp^T B_{n-1} U U^T B_{n-1}^T V_\perp\right), b_{n-1} := \mathrm{Tr}\left(V_\perp^T A_n B_{n-1} U U^T B_{n-1}^T V_\perp\right),$$
$$c_{n-1} := \mathrm{Tr}\left(V_\perp^T A_n B_{n-1} U U^T B_{n-1}^T A_n V_\perp\right)$$

Then,

$$
\begin{aligned}
\beta_n &= \mathbb{E}\left[\operatorname{Tr}\left(V_\perp^T B_n U U^T B_n^T V_\perp\right)^2\right] \\
&= \mathbb{E}\left[\left(a_{n-1} + 2\eta b_{n-1} + \eta^2 c_{n-1}\right)^2\right] \\
&\leq \beta_{n-1}\left(1 + 4\eta\lambda_2\right) + 4\eta^2 \underbrace{\mathbb{E}\left[\operatorname{Tr}\left(V_\perp^T A_n B_{n-1} U U^T B_{n-1}^T V_\perp\right)^2\right]}_{T_1} \\
&\quad + 2\eta^2 \underbrace{\mathbb{E}\left[\operatorname{Tr}\left(V_\perp^T B_{n-1} U U^T B_{n-1}^T V_\perp\right)\operatorname{Tr}\left(V_\perp^T A_n B_{n-1} U U^T B_{n-1}^T A_n V_\perp\right)\right]}_{T_2} \\
&\quad + 4\eta^3 \underbrace{\mathbb{E}\left[\operatorname{Tr}\left(V_\perp^T A_n B_{n-1} U U^T B_{n-1}^T V_\perp\right)\operatorname{Tr}\left(V_\perp^T A_n B_{n-1} U U^T B_{n-1}^T A_n V_\perp\right)\right]}_{T_3} \\
&\quad + \eta^4 \underbrace{\mathbb{E}\left[\operatorname{Tr}\left(V_\perp^T A_n B_{n-1} U U^T B_{n-1}^T A_n V_\perp\right)^2\right]}_{T_4}
\end{aligned}
\tag{A.32}
$$

For $T_1$,

$$
\begin{aligned}
T_1 &= \mathbb{E}\left[\operatorname{Tr}\left(V_\perp^T A_n B_{n-1} U U^T B_{n-1}^T V_\perp\right)^2\right] \\
&= \mathbb{E}\left[\left(X_n^T B_{n-1} U U^T B_{n-1}^T V_\perp V_\perp^T X_n\right)^2\right] \\
&\leq \mathbb{E}\left[\left\|V_\perp^T B_{n-1} U U^T B_{n-1}^T X_n\right\|_2^2 \left\|V_\perp^T X_n\right\|_2^2\right] \\
&= \mathbb{E}\left[\mathbb{E}\left[\left\|V_\perp^T B_{n-1} U U^T B_{n-1}^T X_n\right\|_2^2 \left\|V_\perp^T X_n\right\|_2^2 \,\Big|\, \mathcal{F}_{n-1}\right]\right] \\
&\leq \mathbb{E}\left[\sqrt{\mathbb{E}\left[\left\|V_\perp^T B_{n-1} U U^T B_{n-1}^T X_n\right\|_2^2 \,\Big|\, \mathcal{F}_{n-1}\right]\mathbb{E}\left[\left\|V_\perp^T X_n\right\|_2^2 \,\Big|\, \mathcal{F}_{n-1}\right]}\right] \\
&\leq \left(2L^2\sigma^2\right)^2 \mathbb{E}\left[\operatorname{Tr}\left(V_\perp^T B_{n-1} U U^T B_{n-1}^T \Sigma B_{n-1} U U^T B_{n-1}^T V_\perp\right)\operatorname{Tr}\left(V_\perp^T \Sigma V_\perp\right)\right], \ \text{using Lemma } A.2.2 \text{ with } p = 2 \\
&\leq \left(2L^2\sigma^2\right)^2 \mathbb{E}\left[\operatorname{Tr}\left(V_\perp^T B_{n-1} U U^T B_{n-1}^T \left(\lambda_1 v_1 v_1^T + V_\perp \Lambda_2 V_\perp^T\right) B_{n-1} U U^T B_{n-1}^T V_\perp\right)\operatorname{Tr}\left(V_\perp^T \Sigma V_\perp\right)\right] \\
&\leq \left(2L^2\sigma^2\right)^2 \operatorname{Tr}\left(\Lambda_2\right)\left(\lambda_2 \beta_{n-1} + \lambda_1 \sqrt{\alpha_{n-1}\beta_{n-1}}\right)
\end{aligned}
$$

For $T_2$,

$$
\begin{aligned}
T_2 &= \mathbb{E}\left[\operatorname{Tr}\left(V_\perp^T B_{n-1} U U^T B_{n-1}^T V_\perp\right)\operatorname{Tr}\left(V_\perp^T A_n B_{n-1} U U^T B_{n-1}^T A_n V_\perp\right)\right] \\
&= \mathbb{E}\left[\operatorname{Tr}\left(V_\perp^T B_{n-1} U U^T B_{n-1}^T V_\perp\right)\left(X_n^T B_{n-1} U U^T B_{n-1}^T X_n\right)\left(X_n^T V_\perp V_\perp^T X_n\right)\right] \\
&= \mathbb{E}\left[\operatorname{Tr}\left(V_\perp^T B_{n-1} U U^T B_{n-1}^T V_\perp\right)\mathbb{E}\left[\left(X_n^T B_{n-1} U U^T B_{n-1}^T X_n\right)\left(X_n^T V_\perp V_\perp^T X_n\right)\,\Big|\, \mathcal{F}_{n-1}\right]\right] \\
&\leq \mathbb{E}\left[\operatorname{Tr}\left(V_\perp^T B_{n-1} U U^T B_{n-1}^T V_\perp\right)\sqrt{\mathbb{E}\left[\left(X_n^T B_{n-1} U U^T B_{n-1}^T X_n\right)^2\,\Big|\, \mathcal{F}_{n-1}\right]\mathbb{E}\left[\left(X_n^T V_\perp V_\perp^T X_n\right)^2\,\Big|\, \mathcal{F}_{n-1}\right]}\right] \\
&\stackrel{(i)}{\leq} \left(2L^2\sigma^2\right)^2 \mathbb{E}\left[\operatorname{Tr}\left(V_\perp^T B_{n-1} U U^T B_{n-1}^T V_\perp\right)\operatorname{Tr}\left(U^T B_{n-1}^T \Sigma B_{n-1} U\right)\operatorname{Tr}\left(V_\perp^T \Sigma V_\perp\right)\right], \\
&= \left(2L^2\sigma^2\right)^2 \operatorname{Tr}\left(\Lambda_2\right)\mathbb{E}\left[\operatorname{Tr}\left(V_\perp^T B_{n-1} U U^T B_{n-1}^T V_\perp\right)\operatorname{Tr}\left(U^T B_{n-1}^T \left(\lambda_1 v_1 v_1^T + V_\perp \Lambda_2 V_\perp^T\right) B_{n-1} U\right)\right] \\
&\leq \left(2L^2\sigma^2\right)^2 \operatorname{Tr}\left(\Lambda_2\right)\left(\lambda_2 \beta_{n-1} + \lambda_1 \sqrt{\alpha_{n-1}\beta_{n-1}}\right)
\end{aligned}
$$

where in $(i)$ we used Lemma $A.2.2$ with $p = 2$. For $T_3$,

$$
\begin{aligned}
T_3 &= \mathbb{E}\left[\operatorname{Tr}\left(V_\perp^T A_n B_{n-1} U U^T B_{n-1}^T V_\perp\right) \operatorname{Tr}\left(V_\perp^T A_n B_{n-1} U U^T B_{n-1}^T A_n V_\perp\right)\right] \\
&= \mathbb{E}\left[\left(X_n^T B_{n-1} U U^T B_{n-1}^T V_\perp V_\perp^T X_n\right)\left(X_n^T B_{n-1} U U^T B_{n-1}^T X_n\right)\left(X_n^T V_\perp V_\perp^T X_n\right)\right] \\
&\leq \mathbb{E}\left[\left\|U^T B_{n-1}^T X_n\right\|_2^3 \left\|V_\perp^T X_n\right\|_2^3 \left\|U^T B_{n-1}^T V_\perp\right\|_2\right] \\
&= \mathbb{E}\left[\mathbb{E}\left[\left\|U^T B_{n-1}^T X_n\right\|_2^3 \left\|V_\perp^T X_n\right\|_2^3 \Big| \mathcal{F}_{n-1}\right] \left\|U^T B_{n-1}^T V_\perp\right\|_2\right] \\
&\leq \mathbb{E}\left[\sqrt{\mathbb{E}\left[\left\|U^T B_{n-1}^T X_n\right\|_2^6 \Big| \mathcal{F}_{n-1}\right] \mathbb{E}\left[\left\|V_\perp^T X_n\right\|_2^6 \Big| \mathcal{F}_{n-1}\right]} \left\|U^T B_{n-1}^T V_\perp\right\|_2\right] \\
&= \mathbb{E}\left[\sqrt{\mathbb{E}\left[\left(X_n^T B_{n-1} U U^T B_{n-1}^T X_n\right)^3 \Big| \mathcal{F}_{n-1}\right] \mathbb{E}\left[\left(X_n^T V_\perp V_\perp^T X_n\right)^3 \Big| \mathcal{F}_{n-1}\right]} \left\|U^T B_{n-1}^T V_\perp\right\|_2\right] \\
&\leq \left(3L^2\sigma^2\right)^3 \mathbb{E}\left[\operatorname{Tr}\left(U^T B_{n-1}^T \Sigma B_{n-1} U\right)^{\frac{3}{2}} \operatorname{Tr}\left(V_\perp^T \Sigma V_\perp\right)^{\frac{3}{2}} \left\|U^T B_{n-1}^T V_\perp\right\|_2\right], \text{ using Lemma } A.2.2 \text{ with } p = 3 \\
&= \left(3L^2\sigma^2\right)^3 \operatorname{Tr}\left(\Lambda_2\right)^{\frac{3}{2}} \mathbb{E}\left[\operatorname{Tr}\left(U^T B_{n-1}^T \Sigma B_{n-1} U\right)^{\frac{3}{2}} \left\|U^T B_{n-1}^T V_\perp\right\|_2\right] \\
&\leq 2\left(3L^2\sigma^2\right)^3 \operatorname{Tr}\left(\Lambda_2\right)^{\frac{3}{2}} \mathbb{E}\left[\left(\lambda_1^{\frac{3}{2}}\left(v_1^T B_{n-1} U U^T B_{n-1}^T v_1\right)^{\frac{3}{2}} + \lambda_2^{\frac{3}{2}}\operatorname{Tr}\left(V_\perp^T B_{n-1} U U^T B_{n-1}^T V_\perp\right)^{\frac{3}{2}}\right) \left\|U^T B_{n-1}^T V_\perp\right\|_2\right] \\
&\leq 2\left(3L^2\sigma^2\right)^3 \lambda_1^{\frac{3}{2}} \operatorname{Tr}\left(\Lambda_2\right)^{\frac{3}{2}} \mathbb{E}\left[\left(v_1^T B_{n-1} U U^T B_{n-1}^T v_1\right)^{\frac{3}{2}} \operatorname{Tr}\left(V_\perp^T B_{n-1} U U^T B_{n-1}^T V_\perp\right)^{\frac{1}{2}}\right] \\
&\quad + 2\left(3L^2\sigma^2\right)^3 \lambda_2^{\frac{3}{2}} \operatorname{Tr}\left(\Lambda_2\right)^{\frac{3}{2}} \mathbb{E}\left[\operatorname{Tr}\left(V_\perp^T B_{n-1} U U^T B_{n-1}^T V_\perp\right)^2\right] \\
&= 2\left(3L^2\sigma^2\right)^3 \lambda_1^{\frac{3}{2}} \operatorname{Tr}\left(\Lambda_2\right)^{\frac{3}{2}} \mathbb{E}\left[\left(v_1^T B_{n-1} U U^T B_{n-1}^T v_1\right)^{\frac{3}{2}} \operatorname{Tr}\left(V_\perp^T B_{n-1} U U^T B_{n-1}^T V_\perp\right)^{\frac{1}{2}}\right] \\
&\quad + 2\left(3L^2\sigma^2\right)^3 \lambda_2^{\frac{3}{2}} \operatorname{Tr}\left(\Lambda_2\right)^{\frac{3}{2}} \mathbb{E}\left[\operatorname{Tr}\left(V_\perp^T B_{n-1} U U^T B_{n-1}^T V_\perp\right)^2\right] \\
&= 2\left(3L^2\sigma^2\right)^3 \lambda_1^{\frac{3}{2}} \operatorname{Tr}\left(\Lambda_2\right)^{\frac{3}{2}} \mathbb{E}\left[\sqrt{\left(v_1^T B_{n-1} U U^T B_{n-1}^T v_1\right) \operatorname{Tr}\left(V_\perp^T B_{n-1} U U^T B_{n-1}^T V_\perp\right)}\left(v_1^T B_{n-1} U U^T B_{n-1}^T v_1\right)\right] \\
&\quad + 2\left(3L^2\sigma^2\right)^3 \lambda_2^{\frac{3}{2}} \operatorname{Tr}\left(\Lambda_2\right)^{\frac{3}{2}} \mathbb{E}\left[\operatorname{Tr}\left(V_\perp^T B_{n-1} U U^T B_{n-1}^T V_\perp\right)^2\right] \\
&\leq 2\left(3L^2\sigma^2\right)^2 \lambda_1 \operatorname{Tr}\left(\Lambda_2\right)^2 \sqrt{\alpha_{n-1}\beta_{n-1}} + 2\left(3L^2\sigma^2\right)\lambda_1^2 \operatorname{Tr}\left(\Lambda_2\right)\alpha_{n-1} + 2\left(3L^2\sigma^2\right)^3 \lambda_2^{\frac{3}{2}} \operatorname{Tr}\left(\Lambda_2\right)^{\frac{3}{2}}\beta_{n-1}
\end{aligned}
$$

For $T_4$,

$$
\begin{aligned}
T_4 &= \mathbb{E}\left[\operatorname{Tr}\left(V_\perp^T A_n B_{n-1} U U^T B_{n-1}^T A_n V_\perp\right)^2\right] \\
&= \mathbb{E}\left[\left(X_n^T B_{n-1} U U^T B_{n-1}^T X_n\right)^2 \left(X_n^T V_\perp V_\perp^T X_n\right)^2\right] \\
&= \mathbb{E}\left[\mathbb{E}\left[\left(X_n^T B_{n-1} U U^T B_{n-1}^T X_n\right)^2 \left(X_n^T V_\perp V_\perp^T X_n\right)^2 \Big| \mathcal{F}_{n-1}\right]\right] \\
&\leq \mathbb{E}\left[\sqrt{\mathbb{E}\left[\left(X_n^T B_{n-1} U U^T B_{n-1}^T X_n\right)^4 \Big| \mathcal{F}_{n-1}\right] \mathbb{E}\left[\left(X_n^T V_\perp V_\perp^T X_n\right)^4 \Big| \mathcal{F}_{n-1}\right]}\right] \\
&\leq \left(4L^2\sigma^2\right)^4 \mathbb{E}\left[\operatorname{Tr}\left(U^T B_{n-1}^T \Sigma B_{n-1} U\right)^2 \operatorname{Tr}\left(V_\perp^T \Sigma V_\perp\right)^2\right] \\
&\leq \left(4L^2\sigma^2\right)^4 \operatorname{Tr}\left(\Lambda_2\right)^2 \mathbb{E}\left[\operatorname{Tr}\left(U^T B_{n-1}^T \left(\lambda_1 v_1 v_1^T + V_\perp \Lambda_2 V_\perp^T\right)\right) B_{n-1} U\right]^2 \\
&\leq 2\left(4L^2\sigma^2\right)^4 \operatorname{Tr}\left(\Lambda_2\right)^2 \left(\lambda_1^2 \alpha_{n-1} + \lambda_2^2 \beta_{n-1}\right)
\end{aligned}
$$

Substituting in Eq A.32 along with using $\eta L^2\sigma^2 \leq \frac{1}{4}\min\left\{\frac{1}{\lambda_1}, \frac{1}{\operatorname{Tr}(\Sigma)}, \frac{1}{\sqrt{\lambda_1 \operatorname{Tr}(\Sigma)}}\right\}$ we have,

$$
\begin{aligned}
\beta_n &\leq \left(1 + 4\eta\lambda_2 + 100\eta^2 L^4\sigma^4 \lambda_2 \operatorname{Tr}\left(\Sigma\right)\right)\beta_{n-1} + 100\eta^2 L^2\sigma^2 \lambda_1 \operatorname{Tr}\left(\Sigma\right)\sqrt{\alpha_{n-1}\beta_{n-1}} \\
&\quad + 600\eta^2 L^4\sigma^4 \lambda_1^2 \alpha_{n-1}
\end{aligned}
$$

Hence proved. $\qquad\square$

**Lemma A.2.11.** *Let $U \in \mathbb{R}^{d \times m}$ and $u_0 \sim \mathcal{N}(0, I_d)$, then for all $n \geq 0$ we have*

$$\mathbb{E}\left[u_0^T B_n^T U U^T B_n u_0\right] = \mathbb{E}\left[v_1^T B_n^T U U^T B_n v_1\right] + \mathbb{E}\left[\text{Tr}\left(V_\perp^T B_n^T U U^T B_n V_\perp\right)\right]$$

*Proof.*

$$\begin{aligned}
\mathbb{E}\left[u_0^T B_n^T U U^T B_n u_0\right] &= \mathbb{E}\left[\text{Tr}\left(u_0^T B_n^T U U^T B_n u_0\right)\right] \\
&= \mathbb{E}\left[\mathbb{E}\left[\text{Tr}\left(U^T B_n u_0 u_0^T B_n^T U\right) | B_n\right]\right] \\
&= \mathbb{E}\left[\text{Tr}\left(U^T B_n \mathbb{E}\left[u_0 u_0^T | B_n\right] B_n^T U\right)\right] \\
&= \mathbb{E}\left[\text{Tr}\left(U^T B_n \mathbb{E}\left[u_0 u_0^T\right] B_n^T U\right)\right] \\
&= \mathbb{E}\left[\text{Tr}\left(U^T B_n B_n^T U\right)\right] \\
&= \mathbb{E}\left[\text{Tr}\left(U^T B_n v_1 v_1^T B_n^T U\right)\right] + \mathbb{E}\left[\text{Tr}\left(U^T B_n V_\perp V_\perp^T B_n^T U\right)\right] \\
&= \mathbb{E}\left[v_1^T B_n^T U U^T B_n v_1\right] + \mathbb{E}\left[\text{Tr}\left(V_\perp^T B_n^T U U^T B_n V_\perp\right)\right]
\end{aligned}$$

$\square$

**Lemma A.2.12.** *Let $U \in \mathbb{R}^{d \times m}$ and $u_0 \sim \mathcal{N}(0, I_d)$, then for all $n \geq 0$ we have*

$$\mathbb{E}\left[\left(u_0^T B_n^T U U^T B_n u_0\right)^2\right] \leq 6\mathbb{E}\left[\left(v_1^T B_n^T U U^T B_n v_1\right)^2\right] + 6\mathbb{E}\left[\text{Tr}\left(V_\perp^T B_n^T U U^T B_n V_\perp\right)^2\right]$$

*Proof.* Let the eigendecomposition of $B_n^T U U^T B_n$ for a fixed $B_n$ be given as $P \Lambda P^T$ such that $PP^T = P^T P = I$ and $\Lambda \succeq 0$. Denote $u_0 \equiv u$ and $y := P^T u$. Therefore,

$$\begin{aligned}
\mathbb{E}\left[\left(u_0^T B_n^T U U^T B_n u_0\right)^2\right] &= \mathbb{E}\left[\left(u^T P \Lambda P^T u\right)^2\right] \\
&= \mathbb{E}\left[\left(\sum_{i=1}^d \lambda_i y_i^2\right)^2\right] \\
&= \sum_{i=1}^d \lambda_i^2 \mathbb{E}\left[y_i^4\right] + \sum_{i \neq j} \lambda_i \lambda_j \mathbb{E}\left[y_i^2 y_j^2\right] \quad\quad\quad (\text{A.33})
\end{aligned}$$

Note that $\mathbb{E}[y] = P^T \mathbb{E}[u] = 0$, $\mathbb{E}\left[yy^T\right] = P^T \mathbb{E}\left[uu^T\right] P = P^T P = I$. Therefore, $y \sim \mathcal{N}(0, I_d)$. Therefore, $\mathbb{E}\left[y_i^4\right] = 3$ and $\mathbb{E}\left[y_i^2 y_j^2\right] = \mathbb{E}\left[y_i^2\right] \mathbb{E}\left[y_j^2\right] = 1$. Therefore,

$$\mathbb{E}\left[\left(u_0^T B_n^T U U^T B_n u_0\right)^2\right] = 3\sum_{i=1}^d \lambda_i^2 + \sum_{i \neq j} \lambda_i \lambda_j \quad\quad\quad (\text{A.34})$$

Substituting in Eq A.33, we have

$$\begin{aligned}
\mathbb{E}\left[\left(u_0^T B_n^T U U^T B_n u_0\right)^2\right] &= 3\sum_{i=1}^d \lambda_i^2 + \sum_{i \neq j} \lambda_i \lambda_j \\
&\leq 3\mathbb{E}\left[\left(\sum_{i=1}^d \lambda_i\right)^2\right] \\
&= 3\mathbb{E}\left[\text{Tr}\left(B_n^T U U^T B_n\right)^2\right] \\
&= 3\mathbb{E}\left[\left(\text{Tr}\left(v_1^T B_n^T U U^T B_n v_1\right) + \text{Tr}\left(V_\perp^T B_n^T U U^T B_n V_\perp\right)\right)^2\right] \\
&\leq 6\left(\mathbb{E}\left[\left(v_1^T B_n^T U U^T B_n v_1\right)^2\right] + \mathbb{E}\left[\text{Tr}\left(V_\perp^T B_n^T U U^T B_n V_\perp\right)^2\right]\right)
\end{aligned}$$

$\square$

## A.3  Proofs of entrywise deviation of Oja's vector

We first state some useful results here. Let $a_0 = v_1(i)^2$, $b_0 = \text{Tr}\left(V_\perp^T e_i e_i^T V_\perp\right) = 1 - v_1(i)^2$. Let the learning rate, $\eta$, be set according to Lemma A.2.4. Note that $(1+x) \le \exp(x)$, $\forall x \in \mathbb{R}$. From Lemma A.2.7, we have

$$
\mathbb{E}\left[\left(v_1^T B_n e_i\right)^2\right] \le \left(1 + 2\eta\lambda_1 + 8L^4\sigma^4\eta^2\lambda_1^2\right)^n \left[a_0 + \eta\lambda_1\left(\frac{4\lambda_1}{(\lambda_1 - \lambda_2)}\right)(b_0 + a_0)\right]
$$
$$
\le \exp\left(2\eta n\lambda_1 + 8L^4\sigma^4\eta^2 n\lambda_1^2\right)\left(a_0 + \eta\lambda_1\left(\frac{4\lambda_1}{(\lambda_1 - \lambda_2)}\right)(b_0 + a_0)\right),
$$
(A.35)

$$
\mathbb{E}\left[\text{Tr}\left(V_\perp^T B_n e_i e_i^T B_n^T V_\perp\right)\right] \le b_0\left(1 + 2\eta\lambda_2 + 4L^4\sigma^4\eta^2\lambda_2\text{Tr}(\Sigma) + 4L^4\sigma^4\eta^2\lambda_1^2\right)^n
$$
$$
+ \left[\eta\lambda_1\left(\frac{4\lambda_1}{(\lambda_1 - \lambda_2)}\right)\left(a_0\frac{\text{Tr}(\Sigma)}{\lambda_1} + b_0\right)\right]\left(1 + 2\eta\lambda_1 + 8L^4\sigma^4\eta^2\lambda_1^2\right)^n
$$
$$
\le b_0\exp\left(2\eta n\lambda_2 + 4L^4\sigma^4\eta^2 n\lambda_2\text{Tr}(\Sigma) + 4L^4\sigma^4\eta^2 n\lambda_1^2\right)
$$
$$
+ \left[\eta\lambda_1\left(\frac{4\lambda_1}{(\lambda_1 - \lambda_2)}\right)\left(a_0\frac{\text{Tr}(\Sigma)}{\lambda_1} + b_0\right)\right]\exp\left(2\eta n\lambda_1 + 8L^4\sigma^4\eta^2 n\lambda_1^2\right)
$$
(A.36)

Similarly, from Lemma A.2.8 and using $(a+b)^2 \le 2a^2 + 2b^2$, we have

$$
\mathbb{E}\left[\left(v_1^T B_n e_i\right)^4\right] \le \left(1 + 2\eta\lambda_1 + 100L^4\sigma^4\eta^2\lambda_1^2\right)^{2n}\left[a_0 + \eta\lambda_1\left(\frac{4\lambda_1}{(\lambda_1 - \lambda_2)}\right)(b_0 + a_0)\right]^2
$$
$$
\le \exp\left(4\eta n\lambda_1 + 200L^4\sigma^4\eta^2 n\lambda_1^2\right)\left[a_0 + \eta\lambda_1\left(\frac{4\lambda_1}{(\lambda_1 - \lambda_2)}\right)(b_0 + a_0)\right]^2,
$$
(A.37)

$$
\mathbb{E}\left[\text{Tr}\left(V_\perp^T B_n e_i e_i^T B_n^T V_\perp\right)^2\right]
$$
$$
\le 2b_0^2\left(1 + 2\eta\lambda_2 + 50\eta^2 L^4\sigma^4\lambda_2\text{Tr}(\Sigma) + 50L^4\sigma^4\eta^2\lambda_1^2\right)^{2n}
$$
$$
+ 2\left[\eta\lambda_1\left(\frac{2\lambda_1}{\theta(\lambda_1 - \lambda_2)}\right)\left(a_0\frac{\text{Tr}(\Sigma)}{\lambda_1} + b_0\left(\frac{1-\theta}{\theta}\right)\right)\right]^2\left(1 + 2\eta\lambda_1 + 100L^4\sigma^4\eta^2\lambda_1^2\right)^{2n}
$$
$$
\le 2b_0^2\exp\left(4\eta n\lambda_2 + 100\eta^2 nL^4\sigma^4\lambda_2\text{Tr}(\Sigma) + 100L^4\sigma^4\eta^2 n\lambda_1^2\right)
$$
$$
+ 2\left[\eta\lambda_1\left(\frac{4\lambda_1}{(\lambda_1 - \lambda_2)}\right)\left(a_0\frac{\text{Tr}(\Sigma)}{\lambda_1} + b_0\right)\right]^2\exp\left(4\eta n\lambda_1 + 200L^4\sigma^4\eta^2 n\lambda_1^2\right)
$$
(A.38)

Finally, noting that $(1+x) \ge \exp\left(x - x^2\right) \forall x \ge 0$, we have

$$
\mathbb{E}\left[\left(v_1^T B_n e_i\right)^2\right] \ge \left(\mathbb{E}\left[v_1^T B_n e_i\right]\right)^2 = v_1(i)^2\left(1 + \eta\lambda_1\right)^{2n} \ge v_1(i)^2\exp\left(2\eta n\lambda_1 - 2\eta^2 n\lambda_1^2\right)
$$
(A.39)

Now we are ready to provide proofs of Lemmas 3.11 and 3.12.

**Lemma 3.11** (Tail bound in support). *Fix a $\delta \in (0.1, 1)$. Define the event $\mathcal{G} := \left\{|v_1^T u_0| \ge \frac{\delta}{\sqrt{e}}\right\}$ and threshold $\tau_n := \frac{\delta}{\sqrt{2e}}\min_{i \in S}|v_1(i)|(1 + \eta\lambda_1)^n$. Let the learning rate be set as in Lemma 3.1. Then, for an absolute constant $C_H > 0$,*

$$
\forall i \in S, \quad \mathbb{P}\left(|r_i| \le \tau_n \,\middle|\, \mathcal{G}\right) \le C_H\left[\eta\lambda_1\log(n) + \eta\lambda_1\left(\frac{\lambda_1}{\lambda_1 - \lambda_2}\right)\frac{1}{v_1(i)^2}\right]
$$

*Proof of Lemma 3.11.* Let $u_0 = av_1 + bv_\perp$ for $v_\perp^T v_1 = 0$ and $a = u_0^T v_1$, and $b = u_0^T v_\perp \in \mathbb{R}$ for some vector $v_\perp$ orthogonal to $v_1$. Then $\forall i \in S$,

$$
\begin{aligned}
|r_i| \le \tau_n &\iff r_i^2 \le \tau_n^2 \iff \left(e_i^T B_n y_0\right)^2 \le \tau_n^2 \\
&\iff a^2 \left(e_i^T B_n v_1\right)^2 + b^2 \left(e_i^T B_n v_\perp\right)^2 \le \tau_n^2 \\
&\implies a^2 \left(e_i^T B_n v_1\right)^2 \le \tau_n^2
\end{aligned}
\tag{A.40}
$$

Then,

$$
\begin{aligned}
\mathbb{P}\left(|r_i| \le \tau_n \,\middle|\, \mathcal{G}\right) &\le \mathbb{P}\left(a^2 \left(e_i^T B_n v_1\right)^2 \le \tau_n^2 \,\middle|\, \mathcal{G}\right), \text{ using Eq } A.40 \\
&\le \mathbb{P}\left(\left(e_i^T B_n v_1\right)^2 \le \frac{e\tau_n^2}{\delta^2} \,\middle|\, \mathcal{G}\right) \\
&= \mathbb{P}\left(\left(e_i^T B_n v_1\right)^2 \le \frac{e\tau_n^2}{\delta^2}\right)
\end{aligned}
\tag{A.41}
$$

For convenience of notation, define $\gamma_n := \frac{\sqrt{e}}{\delta}\tau_n$ and $q_i := |e_i^T B_n v_1|$. Then,

$$
\begin{aligned}
\mathbb{P}\left(\left(e_i^T B_n v_1\right)^2 \le \frac{e\tau_n^2}{\delta^2}\right) &= \mathbb{P}\left(q_i \le \gamma_n\right) \\
&\le \mathbb{P}\left(|q_i - \mathbb{E}[q_i]| \ge |\mathbb{E}[q_i]| - \gamma_n\right), \\
&= \mathbb{P}\left(|q_i - \mathbb{E}[q_i]| \ge |\mathbb{E}[q_i]| - \gamma_n\right), \\
&\le \frac{\mathbb{E}\left[q_i^2\right] - \mathbb{E}[q_i]^2}{\left(|\mathbb{E}[q_i]| - \gamma_n\right)^2}, \text{ using Chebyshev's inequality} \\
&= \frac{\mathbb{E}\left[\left(v_1^T B_n^T e_i e_i^T B_n v_1\right)\right] - \mathbb{E}\left[v_1^T B_n^T e_i\right]^2}{\underbrace{\left(|\mathbb{E}\left[v_1^T B_n^T e_i\right]| - \frac{1}{\sqrt{2}}\left(\min_{i \in S}|v_1(i)|\right)\left(1 + \eta\lambda_1\right)^n\right)^2}_{T_i}}
\end{aligned}
\tag{A.42}\tag{A.43}
$$

We now bound $T_i$ using Eq A.35 and Eq A.39 as -

$$
\begin{aligned}
T_i &\le \frac{\exp\left(2\eta n\lambda_1 + 8L^4\sigma^4\eta^2 n\lambda_1^2\right)\left[v_1(i)^2 + \eta\lambda_1\left(\frac{4\lambda_1}{(\lambda_1-\lambda_2)}\right)\right] - v_1(i)^2\left(1 + \eta\lambda_1\right)^{2n}}{\left(|v_1(i)| - \frac{1}{2}\min_{i \in S}|v_1(i)|\right)^2\left(1 + \eta\lambda_1\right)^{2n}} \\
&\le \frac{\exp\left(2\eta n\lambda_1 + 8L^4\sigma^4\eta^2 n\lambda_1^2\right)\left[v_1(i)^2 + \eta\lambda_1\left(\frac{4\lambda_1}{(\lambda_1-\lambda_2)}\right)\right] - v_1(i)^2\exp\left(2\eta n\lambda_1 - 2\eta^2 n\lambda_1^2\right)}{\left(|v_1(i)| - \frac{1}{2}\min_{i \in S}|v_1(i)|\right)^2\exp\left(2\eta n\lambda_1 - 2\eta^2 n\lambda_1^2\right)} \\
&= \frac{\exp\left(2\left(4L^4\sigma^4 + 1\right)\eta^2 n\lambda_1^2\right)\left[v_1(i)^2 + \eta\lambda_1\left(\frac{4\lambda_1}{(\lambda_1-\lambda_2)}\right)\right] - v_1(i)^2}{\left(|v_1(i)| - \frac{1}{2}\min_{i \in S}|v_1(i)|\right)^2}
\end{aligned}
$$

The second inequality follows from the fact that $1 + x \ge \exp(x - x^2)$, for $x \ge 0$. Note that for $x \in (0, 1)$, $\exp x \le 1 + 2x$. Therefore, for $\left(8L^4\sigma^4 + 2\right)\eta^2 n\lambda_1^2 \le \frac{1}{4}$, we have

$$
\begin{aligned}
T_i &\le \frac{\left(\exp\left(2\left(4L^4\sigma^4 + 1\right)\eta^2 n\lambda_1^2\right) - 1\right)v_1(i)^2 + 3\eta\lambda_1\left(\frac{4\lambda_1}{(\lambda_1-\lambda_2)}\right)}{\left(|v_1(i)| - \frac{1}{2}\min_{i \in S}|v_1(i)|\right)^2} \\
&\le 4\left(\frac{4\left(4L^4\sigma^4 + 1\right)\eta^2 n\lambda_1^2 v_1(i)^2 + 3\eta\lambda_1\left(\frac{4\lambda_1}{(\lambda_1-\lambda_2)}\right)}{v_1(i)^2}\right) \\
&= 4\left(4\left(4L^4\sigma^4 + 1\right)\eta^2 n\lambda_1^2 + \frac{3\eta\lambda_1\left(\frac{4\lambda_1}{(\lambda_1-\lambda_2)}\right)}{v_1(i)^2}\right)
\end{aligned}
$$

The result then follows by using Claim(2) in Lemma A.2.4. $\qquad\square$

We next provide the proof of Lemma 3.12.

**Lemma 3.12** (Tail bound outside support). *Fix a $\delta \in (0.1, 1)$. Let the learning rate be set as in Lemma 3.1 and define the threshold $\tau_n := \frac{\delta}{\sqrt{2e}} \min_{i \in S} |v_1(i)| (1 + \eta \lambda_1)^n$. Then, for $\min_i |v_1(i)| = \widetilde{\Omega}\left(\frac{\lambda_1}{\lambda_1 - \lambda_2} \left(\frac{d}{n^2}\right)^{\frac{1}{4}}\right)$ and an absolute constant $C_T > 0$ we have,*

$$\forall i \notin S, \quad \mathbb{P}(|r_i| > \tau_n) \leq C_T \left[\eta^2 \lambda_1^2 \left(\frac{\lambda_1}{\lambda_1 - \lambda_2}\right)^2 \left(\frac{1}{\delta^2 \min_{i \in S_{hi}} v_1(i)^2}\right)^2\right]$$

*Proof of Lemma 3.12.* Note that for $i \notin S$, $a_0 = 0, b_0 = 1$. Therefore, we have,

$$\mathbb{E}\left[\left(v_1^T B_n^T e_i e_i^T B_n v_1\right)^2\right] \leq \exp\left(4\eta n \lambda_1 + 200 L^4 \sigma^4 \eta^2 n \lambda_1^2\right) \left[\eta \lambda_1 \left(\frac{4\lambda_1}{(\lambda_1 - \lambda_2)}\right)\right]^2, \text{ using Eq } A.37,$$

(A.44)

$$\mathbb{E}\left[\text{Tr}\left(V_\perp^T B_n^T e_i e_i^T B_n V_\perp\right)^2\right] \leq 2\exp\left(4\eta n \lambda_2 + 100 \eta^2 n \log(n) L^4 \sigma^4 \lambda_2 \text{Tr}(\Sigma) + 100 L^4 \sigma^4 \eta^2 n \lambda_1^2\right)$$
$$+ 2\left[\eta \lambda_1 \left(\frac{4\lambda_1}{(\lambda_1 - \lambda_2)}\right)\right]^2 \exp\left(4\eta n \lambda_1 + 200 L^4 \sigma^4 \eta^2 n \lambda_1^2\right), \text{ using Eq } A.38$$

(A.45)

$$\mathbb{E}\left[v_1^T B_n^T e_i e_i^T B_n v_1\right] \leq \exp\left(2\eta n \lambda_1 + 8 L^4 \sigma^4 \eta^2 n \lambda_1^2\right) \left(\eta \lambda_1 \left(\frac{4\lambda_1}{(\lambda_1 - \lambda_2)}\right)\right), \text{ using Eq } A.35$$

(A.46)

$$\mathbb{E}\left[\text{Tr}\left(V_\perp^T B_n^T e_i e_i^T B_n V_\perp\right)\right] \leq \exp\left(2\eta n \lambda_2 + 4 L^4 \sigma^4 \eta^2 n \lambda_2 \text{Tr}(\Sigma) + 4 L^4 \sigma^4 \eta^2 n \lambda_1^2\right)$$
$$+ \left[\eta \lambda_1 \left(\frac{4\lambda_1}{(\lambda_1 - \lambda_2)}\right)\right] \exp\left(2\eta n \lambda_1 + 8 L^4 \sigma^4 \eta^2 n \lambda_1^2\right), \text{ using Eq } A.36$$

(A.47)

$$\frac{\delta^2}{2e}\left(\min_{i \in S} v_1(i)^2\right)(1 + \eta \lambda_1)^n \geq \frac{\delta^2}{2e}\left(\min_{i \in S} v_1(i)^2\right) \exp\left(2\eta n \lambda_1 - 2\eta^2 n \lambda_1^2\right) \text{ using Eq } A.39$$

(A.48)

Define

$$g_i := \tau_n^2 - \mathbb{E}\left[r_i^2\right]$$

(A.49)

Note that using Assumptions 2,

$$\frac{16 e \eta \lambda_1 \left(\frac{4\lambda_1}{(\lambda_1 - \lambda_2)}\right)}{\min_{i \in S} v_1(i)^2} \leq \frac{3\log(n)}{n(\lambda_1 - \lambda_2)} \times \frac{128 e \lambda_1}{\min_{i \in S} v_1(i)^2} \leq \frac{768e}{\min_{i \in S} v_1(i)^2} \times \frac{\log(n)}{n} \leq \delta^2$$

where the last inequality follows for sufficiently large $n$ mentioned in the theorem statement. Therefore, using Eq A.46 and A.47 along with Claim (3) from Lemma A.2.4, $g_i$ is bounded as

$$g_i \geq \left(\frac{\delta^2}{4e}\left(\min_{i \in S} v_1(i)^2\right) \exp\left(2\eta n \lambda_1 - 2\eta^2 n \lambda_1^2\right) - \exp\left(2\eta n \lambda_2 + 4 L^4 \sigma^4 \eta^2 n \lambda_2 \text{Tr}(\Sigma) + 4 L^4 \sigma^4 \eta^2 n \lambda_1^2\right)\right)$$

$$\geq \exp\left(2\eta n \lambda_1 - 2\eta^2 n \lambda_1^2\right)\left(\frac{\delta^2}{4e}\left(\min_{i \in S} v_1(i)^2\right) - \exp\left(-\theta \eta n (\lambda_1 - \lambda_2)\right)\right)$$

$$\geq \frac{2\delta^2}{9e^2}\left(\min_{i \in S} v_1(i)^2\right) \exp\left(2\eta n \lambda_1 - 2\eta^2 n \lambda_1^2\right)$$

(A.50)

where the last inequality used Claim (4) from Lemma A.2.4. Therefore, we have,

$$\mathbb{P}\left(|r_i| > \tau_n\right) = \mathbb{P}\left(r_i^2 > \tau_n^2\right)$$
$$= \mathbb{P}\left(r_i^2 - \mathbb{E}\left[r_i^2\right] > \tau_n^2 - \mathbb{E}\left[r_i^2\right]\right),$$
$$\leq \mathbb{P}\left(|r_i^2 - \mathbb{E}\left[r_i^2\right]| > \tau_n^2 - \mathbb{E}\left[r_i^2\right]\right), \text{ since from Eq A.50 } g_i \geq 0$$
$$\leq \frac{\mathbb{E}\left[\left(r_i^2 - \mathbb{E}\left[r_i^2\right]\right)^2\right]}{\left(\tau_n^2 - \mathbb{E}\left[r_i^2\right]\right)^2}$$
$$= \frac{\mathbb{E}\left[r_i^4\right] - \mathbb{E}\left[r_i^2\right]^2}{\left(\tau_n^2 - \mathbb{E}\left[r_i^2\right]\right)^2}$$
$$= \frac{\mathbb{E}\left[\left(y_0^T B_n^T e_i e_i^T B_n y_0\right)^2\right] - \mathbb{E}\left[y_0^T B_n^T e_i e_i^T B_n y_0\right]^2}{\left(\frac{\delta^2}{2e}\left(\min_{i \in S} v_1\left(i\right)^2\right)\left(1 + \eta\lambda_1\right)^n - \left(\mathbb{E}\left[y_0^T B_n^T e_i e_i^T B_n y_0\right]\right)\right)^2} =: R_i \quad (A.51)$$

We now bound $R_i$. Therefore,

$$R_i \leq \frac{\mathbb{E}\left[\left(y_0^T B_n^T e_i e_i^T B_n y_0\right)^2\right]}{\left(\frac{\delta^2}{2e}\left(\min_{i \in S} v_1\left(i\right)^2\right)\left(1 + \eta\lambda_1\right)^n - \left(\mathbb{E}\left[y_0^T B_n^T e_i e_i^T B_n y_0\right]\right)\right)^2}$$
$$\overset{(i)}{\leq} \frac{6\left(\mathbb{E}\left[\left(v_1^T B_n^T e_i e_i^T B_n v_1\right)^2\right] + \mathbb{E}\left[\text{Tr}\left(V_\perp^T B_n^T e_i e_i^T B_n V_\perp\right)^2\right]\right)}{\left(\frac{\delta^2}{2e}\left(\min_{i \in S} v_1\left(i\right)^2\right)\left(1 + \eta\lambda_1\right)^n - \mathbb{E}\left[v_1^T B_n^T e_i e_i^T B_n v_1\right] - \mathbb{E}\left[\text{Tr}\left(V_\perp^T B_n^T e_i e_i^T B_n V_\perp\right)\right]\right)^2}$$

where (i) uses Lemmas A.2.11 and A.2.12. Denote the numerator and denominator of $R_i$ as $N\left(R_i\right)$ and $D\left(R_i\right)$. For the numerator $N\left(R_i\right)$ using Eq A.44 and A.45, we have

$$N\left(R_i\right) \leq 18\left[\eta\lambda_1\left(\frac{4\lambda_1}{\left(\lambda_1 - \lambda_2\right)}\right)\right]^2 \exp\left(4\eta n\lambda_1 + 200L^4\sigma^4\eta^2 n\lambda_1^2\right)\left(1 + \frac{2}{3}\exp\left(-\theta\eta n\left(\lambda_1 - \lambda_2\right)\right)\right)$$
$$\leq 20\left[\eta\lambda_1\left(\frac{4\lambda_1}{\left(\lambda_1 - \lambda_2\right)}\right)\right]^2 \exp\left(4\eta n\lambda_1 + 200L^4\sigma^4\eta^2 n\lambda_1^2\right)$$

where the last inequality follows from Claim (4) in Lemma A.2.4. For the denominator $D\left(R_i\right)$, using Eq A.50

$$D\left(R_i\right) = g_i^2 \geq \frac{\delta^4}{20e^2}\left(\min_{i \in S} v_1\left(i\right)^2\right)^2 \exp\left(4\eta n\lambda_1 - 4\eta^2 n\lambda_1^2\right)$$

Recall that for $x \in (0, 1)$, $\exp\left(x\right) \leq 1 + 2x$. Therefore, for $\left(100L^4\sigma^4 + 2\right)\eta^2 n\lambda_1^2 \leq \frac{1}{4}$ which holds due to Claim (2) from Lemma A.2.4, substituting in Eq A.51, we have

$$\mathbb{P}\left(|r_i| > \tau_n\right) \leq \left(\frac{400e^2\left(4\lambda_1\right)^2}{\delta^4\left(\min_{i \in S} v_i^4\right)\left(\lambda_1 - \lambda_2\right)^2}\right)\eta^2\lambda_1^2 \quad (A.52)$$

□

## A.4 Proof of convergence for support recovery (Lemma 3.1, Theorem 3.2)

We start with the proof of Lemma 3.1.

**Lemma 3.1** (s-Agnostic Recovery). *Under Assumptions 1,2, for $\min_i |v_1\left(i\right)| = \widetilde{\Omega}\left(\frac{\lambda_1}{\lambda_1 - \lambda_2}\left(\frac{d}{n^2}\right)^{\frac{1}{4}}\right)$,*
$\widehat{S} \leftarrow \mathsf{OjaSupportRecovery}\left(\{X_i\}_{i \in [n]}, k, \eta := \frac{3\log(n)}{n(\lambda_1 - \lambda_2)}\right)$ *with $k \geq s$ satisfies, $\mathbb{P}\left(S \subseteq \widehat{S}\right) \geq 0.9$.*

*Proof.* Let $\mathcal{E} := \left\{S \subseteq \widehat{S}\right\}$ and set $\delta := \frac{1}{50}$ for this proof. We upper bound $\mathbb{P}\left(\mathcal{E}^c\right)$. Define $r_i := e_i^T B_n u_0$, $i \in [d]$. Observe that

$$\mathcal{E} \iff \exists \tau_n > 0 \text{ such that } \{\forall i \in S, |r_i| \geq \tau_n\}\bigcap\{|\{i : i \notin S, |r_i| \geq \tau_n\}| \leq k - s\}$$

or equivalently,

$$\mathcal{E}^c \iff \forall \tau_n > 0, \{\exists i \in S, |r_i| \le \tau_n\} \bigcup \{|\{i : i \notin S, |r_i| \ge \tau_n\}| > k - s\}$$

Therefore, for any fixed $\tau_n > 0$

$$\mathcal{E}^c \implies \{\exists i \in S, |r_i| \le \tau_n\} \bigcup \{|\{i : i \notin S, |r_i| \ge \tau_n\}| > k - s\}$$

Let $\mathcal{G} := \left\{ |v_1^T u_0| \ge \frac{\delta}{\sqrt{e}} \right\}$ and threshold $\tau_n := \frac{\delta}{\sqrt{2e}} \min_{i \in S} |v_1(i)| (1 + \eta \lambda_1)^n$. Using a union-bound,

$$\mathbb{P}(\mathcal{E}^c) \le \mathbb{P}(\{\exists i \in S, |r_i| \le \tau_n\}) + \mathbb{P}(|\{i : i \notin S, |r_i| \ge \tau_n\}| > k - s)$$

$$\le \mathbb{P}(\{\exists i \in S, |r_i| \le \tau_n\}) + \frac{\sum_{i \notin S} \mathbb{P}(|r_i| \ge \tau_n)}{k - s + 1}, \text{ using Markov's inequality}$$

$$= \mathbb{P}(\mathcal{G}) \mathbb{P}\left(\exists i \in S, |r_i| \le \tau_n \Big| \mathcal{G}\right) + \mathbb{P}(\mathcal{G}^c) \mathbb{P}\left(\exists i \in S, |r_i| \le \tau_n \Big| \mathcal{G}^c\right) +$$

$$+ \frac{\sum_{i \notin S} \mathbb{P}(|r_i| \ge \tau_n)}{k - s + 1}$$

$$\le \mathbb{P}(\mathcal{G}^c) + \mathbb{P}\left(\exists i \in S, |r_i| \le \tau_n \Big| \mathcal{G}\right) + \frac{\sum_{i \notin S} \mathbb{P}(|r_i| \ge \tau_n)}{k - s + 1}$$

$$\le \mathbb{P}(\mathcal{G}^c) + \sum_{i \in S} \mathbb{P}\left(|r_i| \le \tau_n \Big| \mathcal{G}\right) + \frac{\sum_{i \notin S} \mathbb{P}(|r_i| \ge \tau_n)}{k - s + 1}$$

$$\le \mathbb{P}(\mathcal{G}^c) + \underbrace{\sum_{i \in S} \mathbb{P}\left(|r_i| \le \tau_n \Big| \mathcal{G}\right)}_{T_1} + \underbrace{\frac{\sum_{i \notin S} \mathbb{P}(|r_i| \ge \tau_n)}{k - s + 1}}_{T_2}$$

Using Lemma A.2.1, we have $\mathbb{P}(\mathcal{G}^c) \le \delta$. We bound $T_1$ and $T_2$ using Lemmas 3.11 and 3.12 respectively. Therefore,

$$\mathbb{P}(\mathcal{E}^c) \le \delta + C_H \left[ \eta \lambda_1 s \log(n) + \eta \lambda_1 \left( \frac{\lambda_1}{(\lambda_1 - \lambda_2)} \right) \sum_{i \in S} \frac{1}{v_1(i)^2} \right]$$

$$+ C_T \left[ \eta^2 \lambda_1^2 \left( \frac{\lambda_1}{\lambda_1 - \lambda_2} \right)^2 \left( \frac{1}{\delta^2 \min_{i \in S_{\text{hi}}} v_1(i)^2} \right)^2 \right] (d - s)$$

$$\le 5\delta$$

where the last inequality follows by using the bound on $n$. $\qquad\qquad\square$

Next, using Lemma 3.1, we prove Theorem 3.2.

**Theorem 3.2** (High probability support recovery)**.** *Let Assumptions 1, 2 hold. For dataset* $\mathcal{D} := \{X_i\}_{i \in [n]}$, *let* $\mathcal{A}$ *be the randomized algorithm which computes* $\widehat{S} \leftarrow$ OjaSupportRecovery$\left(\{X_i\}_{i \in [n]}, k, \eta\right)$, *where* $\eta := \frac{3 \log(n)}{n(\lambda_1 - \lambda_2)}$ *and* $k = s$. *Then, for* $\delta \in (0, 1)$, $\min_i |v_1(i)| = \widetilde{\Omega}\left( \frac{\lambda_1}{\lambda_1 - \lambda_2} \left( \frac{d}{n^2} \right)^{\frac{1}{4}} \right)$, $\tilde{S} \leftarrow$ SuccessBoost$\left(\{X_i\}_{i \in [n]}, \mathcal{A}, \delta\right)$ *satisfies,*

$$\mathbb{P}\left(\tilde{S} = S\right) \ge 1 - \delta$$

*Proof.* Consider the set $\mathcal{T} := \{\mathcal{S} : \mathcal{S} \subseteq [d], |\mathcal{S}| = s\}$ with the associated metric $\rho(\mathcal{S}, \mathcal{S}') := \mathbb{1}(\mathcal{S} \ne \mathcal{S}')$.

Then, Lemma 3.1 shows that the randomized algorithm, $\mathcal{A}$, is a ConstantSuccessOracle$(\mathcal{D}, \theta, \mathcal{T}, \rho, S, 0)$ (Definition 3.9).

Therefore, the result follows from Lemma 3.10.

$\qquad\qquad\square$

## A.5 Proof of convergence for sparse PCA (Theorems 3.5,3.7)

We start by providing the proof of Theorem 3.5 in Section A.5.1, and then provide the proof of Theorem 3.7 in Section A.5.2.

### A.5.1 Proof of theorem 3.5

Let $\widehat{v} := \frac{B_n w_0}{\|B_n w_0\|_2}$. Then, for any subset $\hat{S} \subseteq S$ (obtained from a support recovery procedure such as Algorithm 1), the corresponding output of a truncation procedure with respect to $\widehat{S}$ is given as:

$$v_{\text{trunc}} := \frac{\lfloor \widehat{v} \rfloor_{\widehat{S}}}{\left\| \lfloor \widehat{v} \rfloor_{\widehat{S}} \right\|_2} = \frac{I_{\widehat{S}} \widehat{v}}{\left\| I_{\widehat{S}} \widehat{v} \right\|_2} = \frac{I_{\widehat{S}} B_n w_0}{\left\| I_{\widehat{S}} B_n w_0 \right\|_2} \tag{A.53}$$

We first prove a general and flexible result that bounds the $\sin^2$ error as a function of $B_n$ (see Eq 3) and analyze the performance of $v_{\text{trunc}}$ by viewing it as a power method on $w_0$ followed by a truncation using the set $\widehat{S}$ in the following result.

**Lemma A.5.1.** *Let $B_n$ and $v_{trunc}$ be defined as in Eq 3 and Eq A.53 respectively. For $\widehat{S} \subseteq [d]$ such that $\widehat{S} \perp\!\!\!\perp B_n, w_0$, then with probability at least $1 - \delta$*

$$\sin^2 (v_{trunc}, v_1) \leq \frac{C \log \left( \frac{1}{\delta} \right)}{\delta^2} \frac{\text{Tr} \left( B_n^T \left( I_{\widehat{S}} - I_{\widehat{S}} v_1 v_1^T I_{\widehat{S}} \right) B_n \right)}{v_1^T B_n^T I_{S \cap \widehat{S}} B_n v_1}$$

*where $C$ is an absolute constant and $\delta \in (0, 1)$.*

*Proof.* Using the definition of $\sin^2$ error,

$$\sin^2 (v_{\text{trunc}}, v_1) = 1 - \left( \frac{v_1^T I_{\widehat{S}} B_n w_0}{\left\| I_{\widehat{S}} B_n w_0 \right\|_2} \right)^2 = \frac{w_0^T B_n^T \left( I_{\widehat{S}} - I_{\widehat{S}} v_1 v_1^T I_{\widehat{S}} \right) B_n w_0}{w_0^T B_n^T I_{\widehat{S}} B_n w_0} \tag{A.54}$$

For the denominator, with probability at least $(1 - \delta)$, we have

$$w_0^T B_n^T I_{\widehat{S}} B_n w_0 \geq w_0^T B_n^T I_{S \cap \widehat{S}} B_n w_0 \overset{(i)}{\geq} \frac{\delta^2}{e} \text{Tr} \left( B_n^T I_{S \cap \widehat{S}} B_n \right) \geq \frac{\delta^2}{e} v_1^T B_n^T I_{S \cap \widehat{S}} B_n v_1 \tag{A.55}$$

where $(i)$ follows from Lemma A.2.1. For the numerator, using $\zeta_2$ from Lemma 3.1 of [JJK+16], with probability at least $(1 - \delta)$, we have

$$w_0^T B_n^T \left( I_{\widehat{S}} - I_{\widehat{S}} v_1 v_1^T I_{\widehat{S}} \right) B_n w_0 \leq C' \log \left( \frac{1}{\delta} \right) \text{Tr} \left( B_n^T \left( I_{\widehat{S}} - I_{\widehat{S}} v_1 v_1^T I_{\widehat{S}} \right) B_n \right) \tag{A.56}$$

Combining Eq A.55 and Eq A.56 with Eq A.54 completes our proof. □

Lemma A.5.1 provides an intuitive sketch of our proof strategy. Following the recipe proposed in [JJK+16], we show how to upper-bound $\epsilon_n := \frac{C \log \left( \frac{1}{\delta} \right)}{\delta^2} \frac{\text{Tr} \left( B_n^T \left( I_{\widehat{S}} - I_{\widehat{S}} v_1 v_1^T I_{\widehat{S}} \right) B_n \right)}{v_1^T B_n^T I_{S \cap \widehat{S}} B_n v_1}$. For upper-bounding the numerator, we bound $\mathbb{E} \left[ \text{Tr} \left( B_n^T \left( I_{\widehat{S}} - I_{\widehat{S}} v_1 v_1^T I_{\widehat{S}} \right) B_n \right) \right]$ and use Markov's inequality. To lower-bound the denominator, we lower-bound $\mathbb{E} \left[ v_1^T B_n^T I_{S \cap \widehat{S}} B_n v_1 \right]$, upper-bound the variance $\mathbb{E} \left[ \left( v_1^T B_n^T I_{S \cap \widehat{S}} B_n v_1 \right)^2 \right]$ and finally use Chebyshev's inequality. A formal analysis is provided in the following theorem -

**Theorem A.5.2** (Convergence of Truncated Oja's Algorithm). *Let $\widehat{S} \subseteq [d]$ be the estimated support set, such that $\widehat{S} \perp\!\!\!\perp B_n, w_0$ (see Algorithm 2). Consider any event $\mathcal{E}$ solely dependent on the randomness of $\widehat{S}$. Define:*

$$W_{\widehat{S}} := \mathbb{E} \left[ I_{\widehat{S}} - I_{\widehat{S}} v_1 v_1^T I_{\widehat{S}} \mid \mathcal{E} \right], \ G_{\widehat{S}} := \mathbb{E} \left[ I_{S \cap \widehat{S}} \mid \mathcal{E} \right]$$

$$\alpha_0 := v_1^T W_{\widehat{S}} v_1, \ \beta_0 := \text{Tr} \left( V_\perp^T W_{\widehat{S}} V_\perp \right), \ p_0 := v_1^T G_{\widehat{S}} v_1, \ q_0 := \text{Tr} \left( V_\perp^T G_{\widehat{S}} V_\perp \right)$$

*Fix $\delta \in (0.1, 1)$. Set the learning rate as $\eta := \frac{3\log(n)}{n(\lambda_1 - \lambda_2)}$. Then, under Assumption 2, for $n = \Omega\left(s\left(\frac{\lambda_1 \log(n)}{\lambda_1 - \lambda_2}\right)^2\right)$ and $p_0\left(1 + \frac{\delta}{16}\right) \leq 1 + 2\eta\lambda_1 s\left(\frac{4\lambda_1}{(\lambda_1 - \lambda_2)}\right)$, we have with probability at least $1 - \delta - \mathbb{P}\left(\mathcal{E}^c\right)$,*

$$\sin^2\left(v_{trunc}, v_1\right) \leq \frac{C'\log\left(\frac{1}{\delta}\right)}{\delta^3}\frac{\lambda_1}{\lambda_1 - \lambda_2}\frac{\alpha_0\left(1 + 2\eta\operatorname{Tr}\left(\Sigma\right)\right) + 2\eta\lambda_1\beta_0}{p_0}$$

*where $v_{trunc}$ is defined in Eq A.53 and $C' > 0$ is an absolute constant.*

*Proof of Theorem A.5.2.* We first note that from Lemma A.5.1, with probability at least $1 - \delta$,

$$\sin^2\left(v_{\text{trunc}}, v_1\right) \leq \frac{C\log\left(\frac{1}{\delta}\right)}{\delta^2}\frac{\operatorname{Tr}\left(B_n^T\left(I_{\widehat{S}} - I_{\widehat{S}}v_1 v_1^T I_{\widehat{S}}\right)B_n\right)}{v_1^T B_n^T I_{S\cap\widehat{S}}B_n v_1} =: \chi \tag{A.57}$$

Next, we bound $\chi$, conditioned on the event $\mathcal{E}$. Using Markov's inequality, we have with probability at least $1 - \delta$,

$$\operatorname{Tr}\left(B_n^T\left(I_{\widehat{S}} - I_{\widehat{S}}v_1 v_1^T I_{\widehat{S}}\right)B_n\right)$$
$$\leq \frac{\mathbb{E}\left[\operatorname{Tr}\left(B_n^T\left(I_{\widehat{S}} - I_{\widehat{S}}v_1 v_1^T I_{\widehat{S}}\right)B_n\right)\Big|\mathcal{E}\right]}{\delta}$$
$$= \frac{\mathbb{E}\left[v_1^T B_n^T\left(I_{\widehat{S}} - I_{\widehat{S}}v_1 v_1^T I_{\widehat{S}}\right)B_n v_1\Big|\mathcal{E}\right] + \mathbb{E}\left[\operatorname{Tr}\left(V_{\perp}^T B_n^T\left(I_{\widehat{S}} - I_{\widehat{S}}v_1 v_1^T I_{\widehat{S}}\right)B_n V_{\perp}\right)\Big|\mathcal{E}\right]}{\delta}$$
$$= \frac{\mathbb{E}\left[v_1^T B_n^T W_{\widehat{S}}B_n v_1\right] + \mathbb{E}\left[\operatorname{Tr}\left(V_{\perp}^T B_n^T W_{\widehat{S}}B_n V_{\perp}\right)\right]}{\delta}, \text{ using } \widehat{S} \perp\!\!\!\perp B_n \tag{A.58}$$

Note that $(1 + x) \leq \exp(x) \,\forall x \in \mathbb{R}$. From Lemma A.2.7, we have

$$\mathbb{E}\left[\operatorname{Tr}\left(v_1^T B_n^T W_{\widehat{S}}B_n v_1\right)\right] \leq \left(1 + 2\eta\lambda_1 + 8L^4\sigma^4\eta^2\lambda_1^2\right)^n\left[\alpha_0 + \eta\lambda_1\left(\frac{4\lambda_1}{(\lambda_1 - \lambda_2)}\right)(\beta_0 + \alpha_0)\right] \tag{A.59}$$

$$\mathbb{E}\left[\operatorname{Tr}\left(V_{\perp}^T B_n^T W_{\widehat{S}}B_n V_{\perp}\right)\right]$$
$$\leq \beta_0\left(1 + 2\eta\lambda_2 + 4L^4\sigma^4\eta^2\lambda_2\operatorname{Tr}\left(\Sigma\right) + 4L^4\sigma^4\eta^2\lambda_1^2\right)^n$$
$$+ \left[\eta\lambda_1\left(\frac{4\lambda_1}{(\lambda_1 - \lambda_2)}\right)\left(\alpha_0\frac{\operatorname{Tr}\left(\Sigma\right)}{\lambda_1} + \beta_0\right)\right]\left(1 + 2\eta\lambda_1 + 8L^4\sigma^4\eta^2\lambda_1^2\right)^n$$
$$\leq \exp\left(2\eta n\lambda_1 + 8L^4\sigma^4\eta^2 n\lambda_1^2\right)\left[\eta\lambda_1\left(\alpha_0\frac{\operatorname{Tr}\left(\Sigma\right)}{\lambda_1} + \beta_0\right)\left(\frac{4\lambda_1}{(\lambda_1 - \lambda_2)}\right) + \beta_0\exp\left(-2\theta\eta n\left(\lambda_1 - \lambda_2\right)\right)\right]$$
$$\leq 2\exp\left(2\eta n\lambda_1 + 8L^4\sigma^4\eta^2 n\lambda_1^2\right)\left[\eta\lambda_1\left(\alpha_0\frac{\operatorname{Tr}\left(\Sigma\right)}{\lambda_1} + \beta_0\right)\left(\frac{4\lambda_1}{(\lambda_1 - \lambda_2)}\right)\right] \tag{A.60}$$

where the last inequality follows due to Lemma A.2.4.

Substituting Eq A.59 and Eq A.60 in Eq A.58, we have with probability at least $(1 - \delta)$, conditioned on the event $\mathcal{E}$,

$$\operatorname{Tr}\left(B_n^T\left(I_{\widehat{S}} - I_{\widehat{S}}v_1 v_1^T I_{\widehat{S}}\right)B_n\right) \leq \frac{\left(\alpha_0\left(1 + 2\eta\operatorname{Tr}\left(\Sigma\right)\right) + 2\eta\lambda_1\beta_0\right)\left(\frac{12\lambda_1}{(\lambda_1 - \lambda_2)}\right)\exp\left(2\eta n\lambda_1 + 8L^4\sigma^4\eta^2 n\lambda_1^2\right)}{\delta} \tag{A.61}$$

Similarly, for the denominator we have with probability at least $1 - \delta$ using Chebyshev's inequality, conditioned on the event $\mathcal{E}$,

$$v_1^T B_n I_{S \bigcap \widehat{S}} B_n^T v_1 \geq \mathbb{E}\left[v_1^T B_n I_{S \bigcap \widehat{S}} B_n^T v_1 \Big| \mathcal{E}\right]\left[1 - \frac{1}{\sqrt{\delta}}\sqrt{\frac{\mathbb{E}\left[\left(v_1^T B_n I_{S \bigcap \widehat{S}} B_n^T v_1\right)^2 \Big| \mathcal{E}\right]}{\mathbb{E}\left[v_1^T B_n I_{S \bigcap \widehat{S}} B_n^T v_1 \Big| \mathcal{E}\right]^2} - 1}\right]$$

(A.62)

Recall that $p_0 := v_1^T \mathbb{E}\left[I_{S \bigcap \widehat{S}} \Big| \mathcal{E}\right] v_1$. Using the argument from Lemma 11 from [JJK$^+$16] and $\widehat{S} \perp\!\!\!\perp B_n$,

$$\mathbb{E}\left[v_1^T B_n I_{S \bigcap \widehat{S}} B_n^T v_1 \Big| \mathcal{E}\right] = \mathbb{E}\left[v_1^T B_n \mathbb{E}\left[I_{S \bigcap \widehat{S}} \Big| \mathcal{E}\right] B_n^T v_1\right] \geq p_0 \exp\left(2\eta n \lambda_1 - 4\eta^2 n \lambda_1^2\right) \text{ (A.63)}$$

This is since the base case of their recursion, [JJK$^+$16] has $v_1^T I v_1$ which is 1, but we have $v_1^T \mathbb{E}\left[I_{S \bigcap \widehat{S}} \Big| \mathcal{E}\right] v_1$ which is defined as $p_0$.

Next, using Lemma A.2.8 and noting that $v_1^T I_{S \bigcap \widehat{S}} v_1 + \text{Tr}\left(V_\perp^T I_{S \bigcap \widehat{S}} V_\perp\right) = \text{Tr}\left(I_{S \bigcap \widehat{S}}\right)$ we have,

$$\mathbb{E}\left[\left(v_1^T B_n I_{S \bigcap \widehat{S}} B_n^T v_1\right)^2 \Big| \mathcal{E}\right]$$

$$\leq \left(1 + 2\eta\lambda_1 + 100L^4\sigma^4\eta^2\lambda_1^2\right)^{2n} \mathbb{E}\left[\left(v_1^T I_{S \bigcap \widehat{S}} v_1 + \eta\lambda_1\left(\frac{4\lambda_1}{(\lambda_1 - \lambda_2)}\right)\text{Tr}\left(I_{S \bigcap \widehat{S}}\right)\right)^2 \Big| \mathcal{E}\right]$$

$$\leq \left(1 + 2\eta\lambda_1 + 100L^4\sigma^4\eta^2\lambda_1^2\right)^{2n} \mathbb{E}\left[\left(v_1^T I_{S \bigcap \widehat{S}} v_1 + \eta\lambda_1 s\left(\frac{4\lambda_1}{(\lambda_1 - \lambda_2)}\right)\right)^2 \Big| \mathcal{E}\right]$$

$$\leq \exp\left(4\eta n \lambda_1 + 200L^4\sigma^4\eta^2 n \lambda_1^2\right)\left[p_0 + 2\eta\lambda_1 s p_0\left(\frac{4\lambda_1}{(\lambda_1 - \lambda_2)}\right) + \left(\eta\lambda_1 s\left(\frac{4\lambda_1}{(\lambda_1 - \lambda_2)}\right)\right)^2\right],$$

(A.64)

where in the last inequality, we used $\left(v_1^T I_{S \bigcap \widehat{S}} v_1\right)^2 \leq v_1^T I_{S \bigcap \widehat{S}} v_1 \leq 1$. For convenience of notation, we define

$$\phi := p_0 + 2\eta\lambda_1 s p_0\left(\frac{4\lambda_1}{(\lambda_1 - \lambda_2)}\right) + \left(\eta\lambda_1 s\left(\frac{4\lambda_1}{(\lambda_1 - \lambda_2)}\right)\right)^2 \leq 4$$

where we used $\eta\lambda_1 s\left(\frac{4\lambda_1}{(\lambda_1 - \lambda_2)}\right) \leq \frac{1}{2}$ (due to Lemma A.2.4) and $p_0 \leq 1$.

Substituting Eq A.63 and Eq A.64 in Eq A.62, and we have with probability at least $(1 - \delta)$,

conditioned on $\mathcal{E}$,

$$v_1^T B_n G_{\widehat{S}} B_n^T v_1$$

$$\geq p_0 \exp\left(2\eta n\lambda_1 - 4\eta^2 n\lambda_1^2\right) \left(1 - \frac{1}{\sqrt{\delta}}\sqrt{\exp\left(\left(100L^4\sigma^4 + 4\right)\eta^2 n\lambda_1^2\right)\frac{\phi}{p_0^2} - 1}\right),$$

$$\overset{(i)}{\geq} p_0 \exp\left(2\eta n\lambda_1 - 4\eta^2 n\lambda_1^2\right) \left(1 - \frac{1}{\sqrt{\delta}}\sqrt{\left(1 + 2\left(100L^4\sigma^4 + 4\right)\eta^2 n\lambda_1^2\right)\frac{\phi}{p_0^2} - 1}\right)$$

$$\geq p_0 \exp\left(2\eta n\lambda_1 - 4\eta^2 n\lambda_1^2\right) \left(1 - \frac{1}{\sqrt{\delta}}\sqrt{\frac{\phi - p_0^2}{p_0^2} + 2\frac{\phi}{p_0^2}\left(100L^4\sigma^4 + 4\right)\eta^2 n\lambda_1^2}\right)$$

$$\overset{(ii)}{\geq} \frac{p_0}{2}\exp\left(2\eta n\lambda_1 - 4\eta^2 n\lambda_1^2\right) \tag{A.65}$$

where in $(i)$ we used $\left(100L^4\sigma^4 + 4\right)\eta^2 n\lambda_1^2 \leq 1$ and $x \in (0,1)$, $\exp(x) \leq 1 + 2x$. For $(ii)$, it suffices to have

$$\frac{\phi - p_0^2}{p_0^2} \leq \frac{\delta}{8}, \quad \frac{\phi}{p_0^2}\left(100L^4\sigma^4 + 4\right)\eta^2 n\lambda_1^2 \leq \frac{\delta}{16}$$

which is further ensured by,

$$p_0 \geq \max\left\{\frac{1 + 2\eta\lambda_1 s\left(\frac{4\lambda_1}{(\lambda_1 - \lambda_2)}\right)}{1 + \frac{\delta}{16}}, \frac{4\eta\lambda_1 s\left(\frac{4\lambda_1}{(\lambda_1 - \lambda_2)}\right)}{\sqrt{\delta}}, 8\sqrt{\frac{\left(100L^4\sigma^4 + 4\right)\eta^2 n\lambda_1^2}{\delta}}\right\}$$

Note that for the choice of $\eta$, and $\delta \geq \frac{1}{10}$, we have using Lemma A.2.4,

$$\frac{1 + 2\eta\lambda_1 s\left(\frac{4\lambda_1}{(\lambda_1 - \lambda_2)}\right)}{1 + \frac{\delta}{16}} \geq \max\left\{\frac{4\eta\lambda_1 s\left(\frac{4\lambda_1}{(\lambda_1 - \lambda_2)}\right)}{\sqrt{\delta}}, 8\sqrt{\frac{\left(100L^4\sigma^4 + 4\right)\eta^2 n\lambda_1^2}{\delta}}\right\}$$

for sufficiently large $n$. Therefore, we only ensure that $p_0$ is greater than the first term in the theorem statement. Finally, let

$$\xi := \frac{C'\log\left(\frac{1}{\delta}\right)}{\delta^3}\frac{\lambda_1}{\lambda_1 - \lambda_2}\frac{\alpha_0\left(1 + 2\eta\operatorname{Tr}(\Sigma)\right) + 2\eta\lambda_1\beta_0}{p_0}$$

Using Eq A.61 and Eq A.65 and substituting in Eq A.57, we have with probability at least $1 - 2\delta$, conditioned on $\mathcal{E}$, $\chi \leq \xi$, or equivalently $\mathbb{P}\left(\chi \geq \xi \big| \mathcal{E}\right) \leq 2\delta$. Therefore,

$$\mathbb{P}(\chi \geq \xi) = \mathbb{P}(\mathcal{E})\mathbb{P}\left(\chi \geq \xi \big| \mathcal{E}\right) + \mathbb{P}(\mathcal{E}^c)\mathbb{P}\left(\chi \geq \xi \big| \mathcal{E}^c\right) \leq \mathbb{P}\left(\chi \geq \xi \big| \mathcal{E}\right) + \mathbb{P}(\mathcal{E}^c) \leq 2\delta + \mathbb{P}(\mathcal{E}^c)$$

The proof follows by making $\delta$ smaller by a constant factor. $\qquad\square$

With Theorem A.5.2 in place, we are ready to finally provide the proof of one of our main results, Theorem 3.5.

**Theorem 3.5** (Vector Truncation). *Let Assumptions 1 and 2 hold and $k \geq s$. For dataset $D := \{X_i\}_{i\in[n]}$ and $w_0 \sim \mathcal{N}(0, I)$, let $\mathcal{A}$ be the randomized algorithm which computes $\hat{v}_{\mathsf{truncvec}} \leftarrow$ $\mathsf{TruncateOja}\left(\{X_i\}_{i\in\left(\frac{n}{2},n\right]}, \widehat{S}, \mathsf{Oja}, \{\eta, w_0\}\right)$, where $\eta := \frac{3\log(n)}{n(\lambda_1 - \lambda_2)}$. Then, for $\min_i |v_1(i)| = \widetilde{\Omega}\left(\left(\frac{d}{n^2}\right)^{\frac{1}{8}}\right)$, $\tilde{v} \leftarrow \mathsf{SuccessBoost}\left(\{X_i\}_{i\in[n]}, \mathcal{A}, d^{-10}\right)$ satisfies,*

$$\sin^2(\tilde{v}, v_1) \leq C''\left(\frac{\lambda_1}{\lambda_1 - \lambda_2}\right)^2\frac{k\log^2(d)}{n}$$

*with probability at least $1 - d^{-10}$, where $C'' \geq 0$ is an absolute constant.*

*Proof.* Let $\mathcal{E} := \left\{ S_{\text{hi}} \subseteq \widehat{S} \right\}$ and set $\delta := \frac{1}{4}$ for this proof. Consider the following variables from Theorem A.5.2:

$$W_{\widehat{S}} := \mathbb{E}\left[ I_{\widehat{S}} - I_{\widehat{S}} v_1 v_1^T I_{\widehat{S}} \mid \mathcal{E} \right], \ G_{\widehat{S}} := \mathbb{E}\left[ I_{S \cap \widehat{S}} \mid \mathcal{E} \right]$$

$$\alpha_0 := v_1^T W_{\widehat{S}} v_1, \ \beta_0 := \text{Tr}\left( V_\perp^T W_{\widehat{S}} V_\perp \right), \ p_0 := v_1^T G_{\widehat{S}} v_1, \ q_0 := \text{Tr}\left( V_\perp^T G_{\widehat{S}} V_\perp \right)$$

Since $|\widehat{S}| = k$, therefore,

$$\beta_0 = \text{Tr}\left( V_\perp^T W_{\widehat{S}} V_\perp \right) \leq \text{Tr}\left( W_{\widehat{S}} \right) \leq \text{Tr}\left( \widehat{S} \right) = k \tag{A.66}$$

Furthermore, under event $\mathcal{E}$, $S_{\text{hi}} \subseteq \left\{ S \cap \widehat{S} \right\}$. Therefore,

$$p_0 = v_1^T G_{\widehat{S}} v_1 \geq \sum_{i \in S_{\text{hi}}} v_1(i)^2 = 1 - \sum_{i \notin S_{\text{hi}}} v_1(i)^2 \geq 1 - \frac{s \log(n)}{n}, \ \text{using definition of } S_{\text{hi}} \tag{A.67}$$

To verify the assumption on $p_0$ mentioned in Theorem A.5.2, it is sufficient to ensure

$$\eta \lambda_1 s \left( \frac{20 \lambda_1}{\lambda_1 - \lambda_2} \right) \leq \left( 1 - \frac{s \log(n)}{n} \right) \left( 1 + \frac{\delta}{4} \right) - 1$$

which is true by the definition of $\eta$ and $n$ (see Lemma A.2.4). Lastly,

$$\alpha_0 = v_1^T W_{\widehat{S}} v_1 = \mathbb{E}\left[ v_1^T I_{\widehat{S}} v_1 - \left( v_1^T I_{\widehat{S}} v_1 \right)^2 \mid \mathcal{E} \right]$$

$$\leq 1 - \mathbb{E}\left[ v_1^T I_{\widehat{S}} v_1 \mid \mathcal{E} \right], \ \text{using } v_1^T I_{\widehat{S}} v_1 \leq 1$$

$$\leq 1 - \sum_{i \in S_{\text{hi}}} v_1(i)^2, \ \text{since } S_{\text{hi}} \subseteq \widehat{S}$$

$$= \sum_{i \notin S_{\text{hi}}} v_1(i)^2 \leq \frac{s \log(n)}{n} \tag{A.68}$$

Therefore, using bounds on $\beta_0, p_0$ and $\alpha_0$ from Eqs A.66, A.67 and A.68 respectively, in conjunction with Theorem A.5.2, with probability at least $1 - \delta - \mathbb{P}(\mathcal{E}^c)$,

$$\sin^2(v_{\text{oja}}, v_1) \leq \frac{C' \log\left( \frac{1}{\delta} \right)}{\delta^3} \frac{5 \lambda_1}{\lambda_1 - \lambda_2} \eta \lambda_1 k \tag{A.69}$$

Using Lemma 3.1, $\mathbb{P}(\mathcal{E}^c) \leq 5\delta$. The result then follows using Eq A.69 and setting $\delta$ smaller by a constant. $\qquad \square$

### A.5.2 Proof of theorem 3.7

We first state the result from [Lia23] achieving the optimal $\sin^2$ error rate for Oja's Algorithm.

**Proposition A.5.3** (Optimal Rate for Oja's Algorithm with Subgaussian Data (Theorem 3.1, [Lia23]))**.** *Let $\{X_i\}_{i \in [n]}$ be i.i.d samples from a subgaussian distribution (Definition 2.1) with covariance matrix, $\Sigma$, leading eigenvector $v_1$ and eigengap, $\lambda_1 - \lambda_2 > 0$. Then, there exists an algorithm* OptimalOja *which operates in $O(d)$ space, $O(nd)$ time, processes one datapoint at a time, and returns an estimate $\hat{v}$ which satisfies, with probability at least $1 - \delta$, $\delta \in (0, 1)$*

$$\sin^2(\hat{v}, v_1) \leq C \frac{\lambda_1 \lambda_2}{(\lambda_1 - \lambda_2)^2} \frac{d \log\left( \frac{1}{\delta} \right)}{n}$$

*where $C > 0$ is an absolute constant.*

**Theorem 3.7** (Data Truncation)**.** *Let Assumptions 1 and 2 hold and $k \geq s$. For dataset $\mathcal{D} := \{X_i\}_{i \in [n]}$ and $w_0 \sim \mathcal{N}(0, I)$, let $\mathcal{A}$ be the randomized algorithm which computes $\hat{v}_{\text{truncvec}} \leftarrow$*

$\mathsf{TruncateOja}\left(\left\{\lfloor X_i\rfloor_{\widehat{S}}\right\}_{i\in\left(\frac{n}{2},n\right]},\widehat{S},\mathsf{OptimalOja},\{\{\eta_t\}_{t\in\left[\frac{n}{2}\right]},w_0\}\right)$. *Then for* $\min_i|v_1(i)|=$

$\widetilde{\Omega}\left(\frac{\lambda_1}{\lambda_1-\lambda_2}\left(\frac{d}{n^2}\right)^{\frac{1}{4}}\right)$, $\tilde{v}\leftarrow\mathsf{SuccessBoost}\left(\{X_i\}_{i\in[n]},\mathcal{A},d^{-10}\right)$ *satisfies,*

$$\sin^2(\tilde{v},v_1)\leq C''\frac{\lambda_1\lambda_2}{(\lambda_1-\lambda_2)^2}\frac{k\log(d)}{n}$$

*with probability at least* $1-d^{-10}$, *where* $C''\geq 0$ *is an absolute constant.*

*Proof.* Let $\delta:=\frac{1}{3}$. Define the event $\mathcal{E}=\left\{S\subseteq\widehat{S}\right\}$. Using Lemma 3.1, we have that
$$\mathbb{P}(\mathcal{E})\geq 1-\delta$$

Let $\chi:=\sin^2(\hat{v}_{\mathsf{truncvec}},v_1)$ and $\xi:=C\frac{\lambda_1\lambda_2}{(\lambda_1-\lambda_2)^2}\frac{k\log\left(\frac{1}{\delta}\right)}{n}$ for an absolute constant $C>0$. Therefore,

$$\mathbb{P}(\chi\geq\xi)=\mathbb{P}(\mathcal{E})\,\mathbb{P}\left(\chi\geq\xi\Big|\mathcal{E}\right)+\mathbb{P}(\mathcal{E}^c)\,\mathbb{P}\left(\chi\geq\xi\Big|\mathcal{E}^c\right)$$

$$\leq\mathbb{P}\left(\chi\geq\xi\Big|\mathcal{E}\right)+\mathbb{P}(\mathcal{E}^c)$$

$$\leq\mathbb{P}\left(\chi\geq\xi\Big|\mathcal{E}\right)+\delta \tag{A.70}$$

Therefore, next we bound $\mathbb{P}\left(\sin^2(\hat{v}_{\mathsf{truncvec}},v_1)\Big|\mathcal{E}\right)$. Therefore, we seek to bound the $\sin^2$ error after truncating the data using the true support, $S$. Note that
$$\mathbb{E}\left[I_S X X^\top I_S\right]=\lambda_1 v_1 v_1^\top+I_S V_\perp\Lambda_2 V_\perp^\top I_S$$
Therefore, after truncation, the leading eigenvector and eigenvalue are preserved, and the second largest eigenvalue is at most $\lambda_2$. Furthermore, the truncated distribution is still subgaussian, and therefore Proposition A.5.3 is applicable here and we have with probability at least $1-\delta$,
$$\sin^2(\hat{v}_{\mathsf{truncvec}},v_1)\leq\xi \tag{A.71}$$
Eq (A.70) and (A.71) show that $\mathcal{A}$ is a $\mathsf{ConstantSuccessOracle}\,(\mathcal{D},(\eta,k),\mathcal{T},\rho,v_1,O(k\log(d)/n))$ (Definition 3.9) for the set $\mathcal{T}=\left\{u:u\in\mathbb{R}^d,\|u\|_2=1\right\}$ with the metric $\rho(u,v):=\left\|uu^\top-vv^\top\right\|_F=\frac{1}{2}|\sin(u,v)|$. The result then follows from Lemma 3.10. $\qquad\square$

## A.6 Alternate method for truncation

In this section, we present another algorithm for truncation, based on a value-based thresholding, complementary to the technique described in Section 3. The proof technique uses the same tools as the ones described in Section 3. Both Algorithm 2 and 4 may be of independent interest depending on the particular use-case and constraints of the particular problem. Theorem A.6.1 provides the convergence guarantees for Algorithm 4. Note that compared to Theorem 3.5, Theorem A.6.1 provides a better guarantee for the sample size. However, this comes at the cost of the sparsity of the returned vector, $\hat{v}_{\mathsf{oja\text{-}thresh}}$, not being a controllable parameter. We can however show that the support size of $\hat{v}_{\mathsf{oja\text{-}thresh}}$ is $O(s)$ in expectation. For the purpose of this proof, let $S_{\mathsf{hi}}:=\left\{i:i\in S,|v_1(i)|\geq\sqrt{\frac{\log(d)}{n}}\right\}$.

**Theorem A.6.1** (Convergence of Oja-Thresholded). *Let* $\hat{v}_{\mathsf{oja\text{-}thresh}}$, $\widehat{S}$ *be obtained from Algorithm 4. Set the learning rate as* $\eta:=\frac{3\log(n)}{n(\lambda_1-\lambda_2)}$. *Define threshold* $\gamma_n:=\frac{3}{4\sqrt{2}e}\min_{i\in S_{hi}}|v_1(i)|(1+\eta\lambda_1)^n$. *Then for* $n=\widetilde{\Omega}\left(\frac{1}{s\min_{i\in S_{hi}}v_1(i)^4}\left(\frac{\lambda_1}{\lambda_1-\lambda_2}\right)^2\right)$, *we have* $\mathbb{E}\left[|\widehat{S}|\right]\leq C's$ *and with probability at least* $\frac{3}{4}$,

$$\sin^2\left(\hat{v}_{\mathsf{oja\text{-}thresh}},v_1\right)\leq C''\left(\frac{\lambda_1}{\lambda_1-\lambda_2}\right)^2\frac{\max\{s,\log(d)\}\log(d)}{n},\quad S_{hi}\subseteq\widehat{S}$$

*where* $C',C''>0$ *are absolute constants.*

---

**Algorithm 4** Oja-Thresholded $\left( \{X_i\}_{i \in [n]}, \gamma_n, \eta \right)$

---

1: **Input** : Dataset $\{X_i\}_{i \in [n]}$, learning rate $\eta > 0$, truncation threshold $\gamma_n$
2: Set $b_n \leftarrow 0$ and choose $y_0, w_0 \sim \mathcal{N}(0, I)$ independently
3: **for** t in range$[1, \frac{n}{2}]$ **do**
4: $\quad y_t \leftarrow (I + \eta X_t X_t^T) y_{t-1}$
5: $\quad b_n \leftarrow b_n + \log(\|y_t\|_2)$
6: $\quad y_t \leftarrow \frac{y_t}{\|y_t\|_2}$
7: **end for**
8: $\widehat{S} \leftarrow$ Set of indices, $i \in [d]$, such that $\log\left(|e_i^T y_n|\right) + b_n - \log(\gamma_n) \geq 0$.
9: $\widehat{v} \leftarrow \mathsf{Oja}\left( \{X_i\}_{i \in \{n/2+1,\dots,n\}}, \eta, w_0 \right)$
10: $\hat{v}_{\text{oja-thresh}} \leftarrow \frac{[\widehat{v}]_{\widehat{S}}}{\left\| [\widehat{v}]_{\widehat{S}} \right\|_2}$
11: **return** $\left[ \hat{v}_{\text{oja-thresh}}, \widehat{S} \right]$

---

*Proof.* Consider the setting of Theorem A.5.2. Set $\delta := \frac{1}{4}$ for this proof and let $\mathcal{E}$ be the event $\left\{ |v_1^T y_0| \geq \frac{\delta}{\sqrt{e}} \right\}$. By Lemma A.2.1, $\mathbb{P}(\mathcal{E}) \geq 1 - \delta$. Recall the definitions,

$$W_{\widehat{S}} := \mathbb{E}\left[ I_{\widehat{S}} - I_{\widehat{S}} v_1 v_1^T I_{\widehat{S}} \Big| \mathcal{E} \right], \ G_{\widehat{S}} := \mathbb{E}\left[ I_{S \cap \widehat{S}} \Big| \mathcal{E} \right]$$

$$\alpha_0 := v_1^T W_{\widehat{S}} v_1, \ \beta_0 := \mathrm{Tr}\left( V_\perp^T W_{\widehat{S}} V_\perp \right), \ p_0 := v_1^T G_{\widehat{S}} v_1$$

We upper bound $\alpha_0, \beta_0$ and lower bound $p_0$ under the setting of Algorithm 4. Define $r_i := e_i^T B_n y_0, i \in [d]$. For $\alpha_0, p_0$, we have

$$\begin{aligned}
\alpha_0 = v_1^T W_{\widehat{S}} v_1 &= \mathbb{E}\left[ v_1^T I_{\widehat{S}} v_1 - \left( v_1^T I_{\widehat{S}} v_1 \right)^2 \Big| \mathcal{E} \right] \\
&= \mathbb{E}\left[ v_1^T I_{\widehat{S}} v_1 \left( 1 - v_1^T I_{\widehat{S}} v_1 \right) \Big| \mathcal{E} \right] \\
&\leq 1 - \mathbb{E}\left[ v_1^T I_{\widehat{S}} v_1 \Big| \mathcal{E} \right], \ \text{using } v_1^T I_{\widehat{S}} v_1 \leq 1 \\
&= 1 - \sum_{i \in S} v_1(i)^2 \mathbb{P}\left( i \in \widehat{S}; i \in S \Big| \mathcal{E} \right), \\
&= \sum_{i \in S} v_1(i)^2 \mathbb{P}\left( i \notin \widehat{S}; i \in S \Big| \mathcal{E} \right)
\end{aligned} \tag{A.72}$$

$$\begin{aligned}
p_0 &= v_1^T G_{\widehat{S}} v_1 \\
&= v_1^T \mathbb{E}\left[ I_{S \cap \widehat{S}} \right] v_1 \\
&= \sum_{i \in S} v_1(i)^2 \mathbb{P}\left( i \in \widehat{S}; i \in S \Big| \mathcal{E} \right) \\
&= 1 - \sum_{i \in S} v_1(i)^2 \mathbb{P}\left( i \notin \widehat{S}; i \in S \Big| \mathcal{E} \right)
\end{aligned} \tag{A.73}$$

Therefore, for both $\alpha_0, p_0$, we seek to upper bound $\sum_{i\in S} v_1(i)^2 \, \mathbb{P}\left(i \notin \widehat{S}; i \in S \middle| \mathcal{E}\right)$. We have

$$
\begin{aligned}
\sum_{i\in S} v_1(i)^2 \, \mathbb{P}\left(i \notin \widehat{S}; i \in S \middle| \mathcal{E}\right) \quad &= \sum_{i\in S_{\mathrm{hi}}} v_1(i)^2 \, \mathbb{P}\left(i \notin \widehat{S}; i \in S \middle| \mathcal{E}\right) + \sum_{i\in S\backslash S_{\mathrm{hi}}} v_1(i)^2 \, \mathbb{P}\left(i \notin \widehat{S}; i \in S \middle| \mathcal{E}\right) \\
&\leq \frac{s\log(n)}{n} + \sum_{i\in S\backslash S_{\mathrm{hi}}} v_1(i)^2 \, \mathbb{P}\left(i \notin \widehat{S}; i \in S \middle| \mathcal{E}\right) \\
&= \frac{s\log(n)}{n} + \sum_{i\in S\backslash S_{\mathrm{hi}}} v_1(i)^2 \, \mathbb{P}\left(|r_i|< \gamma_n; i \in S \middle| \mathcal{E}\right) \\
&\overset{(i)}{\leq} \frac{s\log(n)}{n} + C_H \sum_{i\in S\backslash S_{\mathrm{hi}}} v_1(i)^2 \left[\eta\lambda_1 \log(n) + \eta\lambda_1 \left(\frac{\lambda_1}{\lambda_1-\lambda_2}\right)\frac{1}{v_1(i)^2}\right] \\
&= \frac{s\log(n)}{n} + C_H\eta\lambda_1 \log(n) + C_H\eta\lambda_1 s'\left(\frac{\lambda_1}{(\lambda_1-\lambda_2)}\right) \\
&\leq C_H\eta\lambda_1 \{s, \log(n)\} \leq \frac{1}{2}, \ \text{ using } |v_1(i)|\geq \sqrt{\frac{\log(n)}{n}}, i\in S_{\mathrm{hi}}
\end{aligned}
$$

For $\beta_0$ we have

$$
\begin{aligned}
\beta_0 &\leq \mathbb{E}\left[\mathrm{Tr}\left(W_{\widehat{S}}\right)\middle|\mathcal{E}\right] = \sum_{i\in[d]} \mathbb{P}\left(i\in\widehat{S}\middle|\mathcal{E}\right) - \sum_{i\in S} v_1(i)^2 \, \mathbb{P}\left(i\in\widehat{S}\middle|\mathcal{E}\right) \\
&\leq \sum_{i\notin S}\mathbb{P}\left(i\in\widehat{S}\middle|\mathcal{E}\right) + \sum_{i\in S}\left(1-v_1(i)^2\right)\mathbb{P}\left(i\in\widehat{S}\middle|\mathcal{E}\right) = \sum_{i\notin S}\mathbb{P}\left(i\in\widehat{S}\middle|\mathcal{E}\right) + s - 1 \\
&\leq \sum_{i\notin S}\mathbb{P}\left(|r_i|\geq \gamma_n\middle|\mathcal{E}\right) + s - 1 = \sum_{i\notin S}\frac{\mathbb{P}\left(|r_i|\geq \gamma_n\right)}{\mathbb{P}\left(\mathcal{E}\right)} + s - 1 \\
&\leq 2\sum_{i\notin S}\mathbb{P}\left(|r_i|\geq \gamma_n\right) + s - 1, \ \text{ since } \mathbb{P}\left(\mathcal{G}\right)\geq 1-\delta \\
&\leq 2C_T\left[\left(\frac{\lambda_1}{\lambda_1-\lambda_2}\right)^2\left(\frac{1}{\delta^2 \min_{i\in S_{\mathrm{hi}}} v_1(i)^2}\right)^2\right]\eta^2\lambda_1^2\left(d-s\right) + s - 1, \ \text{ using Lemma 3.12} \\
&\leq 2s, \ \text{ using bound on } n
\end{aligned}
$$

The result then follows using Theorem A.5.2 and substituting the bounds on $\alpha_0$, $\beta_0$ and $p_0$. Finally, note that using a similar argument as Theorem 3.5, we have

$$
\begin{aligned}
\mathbb{P}\left(S_{\mathrm{hi}}\not\subseteq \widehat{S}|\mathcal{E}\right) &\leq \sum_{i\in S_{\mathrm{hi}}}\mathbb{P}\left(i\notin\widehat{S}; i\in S_{\mathrm{hi}}\middle|\mathcal{E}\right) \\
&= \sum_{i\in S_{\mathrm{hi}}}\mathbb{P}\left(|r_i|< \gamma_n; i\in S\middle|\mathcal{E}\right) \leq C_H\sum_{i\in S_{\mathrm{hi}}}\eta\lambda_1\log(n) + \eta\lambda_1\left(\frac{\lambda_1}{\lambda_1-\lambda_2}\right)\frac{1}{v_1(i)^2} \\
&\leq C_H\eta\lambda_1 s\log(n) + C_H\eta\lambda_1\left(\frac{\lambda_1}{\lambda_1-\lambda_2}\right)\sum_{i\in S_{\mathrm{hi}}}\frac{1}{v_1(i)^2} \leq \delta
\end{aligned}
$$

using the sample size bound on $n$. $\qquad\square$

