# OpenReview forum: "Oja's Algorithm for Streaming Sparse PCA"
_NeurIPS.cc/2024/Conference — NeurIPS 2024 poster_

### Official Review · Reviewer_qSPb · 2024-07-07

**Soundness:** 3
**Presentation:** 3
**Contribution:** 3
**Rating:** 5
**Confidence:** 2

**Summary:**

This paper studies the problem of finding the top eigenvector from samples.
Given $n$ samples drawn from a distribution whose mean is $0$ and covariance matrix is $\Sigma\in\mathbb{R}^{d\times d}$, we would like to find a vector $\hat{v}$ based on these $n$ samples such that $\hat{v}$ and the top eigenvector of $\Sigma$, $v$, have a small $\sin^2$ error which is defined as $1-\langle\hat{v},v\rangle^2$.
Furthermore, we assume that the sparsity of $v$ is $s$ and hence we expect the output $\hat{v}$ also has sparsity $s$.
The goal in this paper is to design a single pass algorithm using $O(nd)$ time and $O(d)$ space such that the error is minimized.
Oja's algorithm is a well known algorithm for finding the top eigenvector and works as follows.
In the $t$-step, we iteratively update our current solution towards the $t$-th sample.
The authors applied Oja's algorithm to achieve the goal and showed that the error is at most $O(\frac{\sigma_*^2 s\log d}{n})$ where $\sigma_*^2$ depends on the top two eigenvalues.

**Strengths:**

- The problem seems to be a natural question and well-motivated.

- The general presentation is good.
The readers of all levels of expertise should be able to follow the main idea in this paper.

**Weaknesses:**

- In terms of techniques, the main idea is to apply the known Oja' algorithm.
I am not sure if there are any fundamental new ideas introduced in this paper.

**Questions:**

Note:

- Line 31: $r_{\textsf{eff}}$ is not defined yet in the main text.

**Limitations:**

.

---

> ### Author Rebuttal · Authors · 2024-08-07
>
> Thank you for your kind words regarding the motivation of our problem and the simplicity of the presentation. We address your primary concerns below:
>
> **[Re: Novelty of fundamental ideas]:** While our algorithm builds on top of Oja’s algorithm, hard thresholding of Oja’s vector for Sparse PCA has not been proposed or analyzed before. We introduce several novel ideas in this analysis and respectfully disagree with the claim of the lack of fundamental new ideas in our work. In what follows, we describe the motivation for the problem considered in our work, how it compares with relevant works under the computational and statistical regime of interest, and bring out the new ideas involved in our analysis.
>
> ### Importance of the Problem Setting
> *Motivation*: Sparse PCA has been a long-standing problem, with many algorithms being proposed in the literature, motivated both by theoretical ideas and practical applications. It has seen a lot of interest from the Computer Science and Statistics communities. Table 1 in our paper provides a detailed summary of various important contributions over the last two decades, along with their statistical performance as a function of various computational and model parameters.
>
> *Problem Statement*: In this paper, we consider a variant of the problem that has been relatively unexplored in the literature - *Can Sparse PCA be performed at the statistically optimal rate in linear time and space, without placing strong structural assumptions on the population covariance matrix?*. The only papers which operate under this tight statistical and computational budget, to the best of our knowledge, are Johnstone and Lu [JL09], Yang and Xu [YX15], and Wang and Lu [WL16]. However, all of them require a spiked population covariance model for their analysis. Figure 1(a) compares the performance of these algorithms on a population covariance model which slightly deviates from the spiked model and shows how critical this assumption is for their performance.
>
> *Our Contribution*: This question on the efficiency of PCA, without sparsity, has been asked before (see e.g. Jain et. al [2016]) and Oja's algorithm has emerged as one of the clear winners, achieving the statistically optimal rate under tight computational and space constraints. However, its extension under the sparsity assumption is challenging and has not been analyzed before. In fact, this algorithm, despite its simplicity, has not been proposed in its current form in the literature before. We therefore believe that this algorithm and its analysis provide a valuable step in the direction of **computationally and space-efficient Sparse PCA** algorithms which work without strong assumptions on the data distribution.
>
> ### Novel Ideas in Our Analysis
>
> 1. We would like to point out that the analysis of thresholding, as pointed out by all other reviewers, is very different and novel, diverging significantly from analyzing regular Oja’s algorithm. In particular, it introduces a tight analysis of a system of linear recurrences, which is novel, as pointed out by Reviewer zu7o.
>
> 2. Our analysis further offers theoretical insights into the behavior of the entries of the Oja vector, which, to the best of our knowledge, has not been attempted before. In particular, Lemma 3.13 and 3.14 provide the first results of their kind and show that although the entries of the Oja vector do not concentrate, it is indeed possible to show a tail-bound on their deviation.
>
> 3. The closest algorithm to our work that we are aware of analyzes soft-thresholding of the Oja vector under a stronger assumption on the covariance model and an initialization close to the population eigenvector (see Wang and Lu, 2016). Furthermore, they provide a PDE-based asymptotic analysis of the problem, whereas our work provides a sharp and non-asymptotic rate of convergence.
>
> **[Re: Definition of $r_\mathsf{eff}$]:** We define $r_\mathsf{eff}$ in Lines 4-5.

---

> > ### Comment · Reviewer_qSPb · 2024-08-10
> >
> > Thanks for the detailed response. I will take this into consideration during the AC-reviewer discussion.
> >
> > Definition of $r_{\text{eff}}$: It may be helpful to formally define it after the abstract.

---

> ### Author Response · Authors · 2024-08-10
>
> Dear Reviewer qSPb,
>
> Thank you for your response. As per your suggestion, we will define $r_\mathsf{eff}$ in an equation after the abstract to make it easy to spot. We would also like to take this opportunity to reiterate that the proof techniques in our paper are completely different from those in the previous papers on streaming PCA and matrix products which were done under bounded $r_\mathsf{eff}$. We hope that our response has conveyed the novelty of our work. If you have any further questions that we can answer, please let us know.

---

### Official Review · Reviewer_jbhG · 2024-07-12

**Soundness:** 4
**Presentation:** 4
**Contribution:** 4
**Rating:** 7
**Confidence:** 4

**Summary:**

The work proposes a one pass Ojas' algorithm which can achieve minimax error bound for high dimensional sparse PCA under standard technical conditions.

**Strengths:**

The paper is extremely well written. The proposed one-pass Oja's algorithm is novel with detailed convergence analysis and convincing numerical experiments to support author's claims. I appreciate the thorough literature review highlighting how the current paper improves on over the previous works (in particular Table 1). The relaxation on effective rank assumption is particularly notable in the results. The mathematical ideas in the proofs are easy to follow and contain several new techniques.

**Weaknesses:**

In general, I like the paper as it is quite clear and mathematically sound. In some minor places, the exposition can be slightly improved by defining terminologies before using them (for instance please define $\sin^2$ error before referring).

**Questions:**

Q: Can something be said along the lines of these results for other top eigenvectors of $\Sigma$ (not just $v_1$)?

---

> ### Author Rebuttal · Authors · 2024-08-07
>
> Thank you for your kind words regarding our presentation, the novelty of our algorithm, the motivation of our problem, and the contribution to literature. We address your primary concerns below:
>
> **[Re: Top-k principal components]** Recent results provide a black-box way to obtain k-PCA given an algorithm to extract the top eigenvector (see [a]) which could be employed treating our algorithm as a 1-PCA oracle. This deflation-styled approach has also been proposed in [b] in the context of Sparse PCA.
>
> We also believe that an analysis such as [c] can be extended to the sparse setting to obtain top-k principal components simultaneously via QR decomposition and thresholding. This could be an interesting direction for future work.
>
> References:
>
> [a] Jambulapati, A., Kumar, S., Li, J., Pandey, S., Pensia, A. &amp; Tian, K.. (2024). Black-Box k-to-1-PCA Reductions: Theory and Applications. Proceedings of Thirty-Seventh Conference on Learning Theory, in Proceedings of Machine Learning Research 247:2564-2607.
>
> [b] Mackey, Lester. "Deflation methods for sparse PCA." Advances in neural information processing systems 21 (2008).
>
> [c] Allen-Zhu, Zeyuan, and Yuanzhi Li. "First efficient convergence for streaming k-pca: a global, gap-free, and near-optimal rate." In 2017 IEEE 58th Annual Symposium on Foundations of Computer Science (FOCS), pp. 487-492. IEEE, 2017.

---

> > ### Comment · Reviewer_jbhG · 2024-08-09
> >
> > I thank the authors for their insightful explanations.

---

### Official Review · Reviewer_zu7o · 2024-07-15

**Soundness:** 4
**Presentation:** 4
**Contribution:** 3
**Rating:** 7
**Confidence:** 4

**Summary:**

The paper studies the problem of streaming sparse PCA under iid data. That is, we have $x_1,...,x_n \sim \mathcal D$ iid vectors in $\mathbb R^d$ which are revealed to us in an online fashion. We want to estimate the top eigenvector of $\Sigma = \mathbb E[xx^\intercal]$. We assume that this top eigenvector $v_1$ is $s$-sparse, for $s = O(\frac{n}{\log(n)})$. The error between the real and estimated top eigenvector is computed using the $\sin^2$ error metric $(1-\langle v_1, \tilde v_1\rangle^2)$.

This paper shows that Oja's algorithm can be used to both find the support of $v_1$ and the values in $v_1$ (assuming we are given some value $k \geq s$ and correctly choose a step size parameter).
This algorithm runs in $O(nd)$ time and $O(d)$ space.
The key technical assumptions made are:
- The data is iid subgaussian
- The sparsity of $v_1$ is $s = O(\frac{n}{\log (n)})$
- The effective dimension (ratio of trace to spectral norm) of the covariance matrix is at most $O(\frac{n}{\log(n) \sigma^2})$ where $\sigma^2 = \frac{\lambda_1}{\lambda_1 - \lambda_2} \cdot \frac{\lambda_2}{\lambda_1 - \lambda_2}$ measures the (square) of the singular value gap between the top two eigvals of $\Sigma$.
- The smallest nonzero entries of the (unit vector) top eigevnector $v_1$ is $\tilde\Omega(\frac{d^{1/8}}{n^{1/4}})$.

In contrast to prior works, this work achieves better $\sin^2$ error that prior $O(nd)$ time algorithms and $O(d)$ space algorithms. It also makes no assumption about the quality of the starting vector (i.e. it's not a local convergence result, it's a global convergence result).

The results are essentially all theoretical, with one experiments used to show that Oja works well here, and another used to elucidate which theoretical bounds are loose.

**Strengths:**

The paper is a nice contribution to the literature, and it's well written. Sparse PCA is an important problem, and handling it in low space and time is important as well. The error metrics and assumptions are all reasonable. The paper is written clearly. It's just sorta all around solid.

The result is original in that it has a clear goal: achieve the $\sin^2$ error of large-space or large-time algorithms (those that use $\omega(nd)$ time or $\omega(d)$ space), but only using $O(nd)$ time and $O(d)$ space. It's especially nice that a pretty naive application of Oja's algorithm can achieve this.

There's also some nice novelty in the proof technique, which bounds the 2nd and 4th moments of the entries of the output of Oja's algorithm. In particular, the authors tighten a bound from the prior work by designing and solving a system of linear recurrences. A cool math setup that I do not often see.

I'm not an expert in the sparse pca world, not the streaming pca world, and certainly not streaming sparse pca. That said, assuming the authors are not omitting any relevant prior works, this results on low error in very low space and time seems pretty cool. It seems like a particularly nice step in the sparse PCA literature.

**Weaknesses:**

There's a few notation inconsistency issues, some minor gripes. Nothing I'm really worried about. I'll push it all to the "questions" section below.

I accept this paper for publication.

**Questions:**

None of these are game-breaking, and many of these are minor typos. Feel free to ignore whatever feels unfair to you. But, do at least make the notation for the initial vector consistent and make the figures easy to read.

1. Annoyingly, the authors __very often__ swap the symbols $y_0$, $u_0$, $w_0$, and $z_0$. Please fix this.
1. Figure 1 is too hard to read. The letters are too small. The axes have no labels. The error is negative somehow (if it's plotting $\pm1$ standard deviation, which is making the error bars negative, consider using 25th and 75th quantiles instead?).
1. It's not clear if Lemma 3.1 and Theorem 3.2 allow us to use over/underestimates of $\eta$, or if the theorems are very tied to that exact value of $eta$. Seems worth discussing.
1. It's not clear why Theorem 3.2 requires $k=s$ instead of $k\geq s$. Discussing this would be nice. Returning a vector whose support is $\log(1/\delta)$ times larger than $s$ seems fine to me, if that's the issue here.
1. Prop 3.4 is written a bit unclearly, namely around the "with sparsity parameter, $n=$" part, since $n$ is not a sparsity parameter.
1. Theorems 3.5 and 3.7 uses failure probability $d^{-10}$, which is fine I guess? But why not just use a $\delta$?
1. Remark 3.6 seems like it might make more sense to have back around assumption 2
1. Section 3.4 strikes me as a very standard analysis style. Not sure why you point to [KLL+23] specifically. You can say it's standard, and point to [KLL+23] as an example of this, maybe? If I'm missing something and it's not standard, lemme know.
1. Figure 2 is also too hard to read, especially in print. The letters are too small. The series look too much like each-other. The y-axis should be more specific in the error. It's not really clear what the dotted / population lines are showing -- is it $\log(E[e_i^\intercal B_n u_0])$, or is it $E[\log(e_i^\intercal B_n u_0)]$? Is it something else? What exactly does "error" mean here?
1. Line 221 can use $E[r_i | u_0]$ imo
1. Idk why equation (4) uses absolute values and $\pm$ on the non-top-eigenvalue terms. Isn't $E[B_n]$ PSD, and thus the second term guaranteed to be nonnegative?
1. Line 221 maybe mention that $E[B_n] = (1+\eta \Sigma)^t^?
1. Line 232 this line about [SSM11] taking $\lambda_1 = d^\alpha \rightarrow \infty$ is kinda confusing because it's not a scale-invariant claim. Is it like a condition number that's getting large?
1. Line 238 should mention that Section 4 explains the technique a bit more in detail
1. Line 247 should really point to Section 4 as well, explaining the technique in a bit more detail
1. [Line 256] If you have space, it'd be nice to understand why Theorem 3.5 needs a more general argument that $U = e_i$.

---

> ### Author Rebuttal · Authors · 2024-08-07
>
> Thank you for your kind and detailed feedback and comments about the clarity and presentation of our work, the novelty of our analysis, and the significance of our contribution to the Sparse PCA literature. We will correct all the typographical issues pointed out and will not address them here individually. We answer your primary concerns and suggestions below:
>
> **[Re: $y_0, u_0, w_0$, and $z_0$]:** Thank you for pointing it out. We will take care to fix them in the final manuscript.
>
> **[Re: Over/under estimates of $\eta$]:**: For step-sizes, we follow the convention in (Balsubramani et al. (2015), De Sa et al. (2015), Jain et al. (2016), Li et al (2017), Allen-Zhu et al. (2016)), Huang et al. (2021), where Oja’s algorithm is analyzed without sparsity and the optimal learning rate requires knowledge of the gap, $\lambda_{1} - \lambda_{2}$, and other model parameters to get the statistically optimal rate. Our sin-squared error roughly is of the form $O\left(\eta\lambda_1+\exp(-n\eta(\lambda_1-\lambda_2)\right)$. The optimal eta ensures that the first term dominates yielding our optimal error rates. However, a small $\eta$ resulting from plugging in an upper bound on the eigengap $\lambda_1-\lambda_2$ may make the second term dominate leading to a suboptimal sin-squared error. We will clarify this further.
>
> **[Re: Figure 1(a) and 2(a), (b)]:** We have provided revised figures in the global author rebuttal document along with detailed explanations of each axis and the legend followed. Figure 1(a) plots the sine-squared error with iterations of the algorithm. The error bars currently represent standard deviations across 100 runs, leading to a negative error. We have now fixed that to plot the 25th and 75th percentile bars in the revised figure and added labels to the axes.
> Figures 2 (a), and (b) have also been revised with a larger font size and clear axis labels. The y-axis in Figure 2(a) has been corrected to read the “value” of the referenced quantity instead of “error” and dotted lines have been removed to avoid confusion. The line width has been increased to enhance clarity. The lines labelled “sample” plot $\log(|e_{i}^{\top}B_{n}u_{0}|)$, whereas the “population” curves plot $\log(|\mathbb{E}[e_{i}^{\top}B_{n}u_{0}]|)$.
>
> **[Re: $k \geq s$ in Theorem 3.2]:** Our probability boosting argument for Algorithm 3 requires a distance metric, such that when the true support $S\in \hat{S}$, $d(S,\hat{S})\leq \epsilon$ for some small $\epsilon>0$. And not only that, we also crucially need that $d(S,\hat{S})\leq \epsilon$ implies that $\hat{S}$ contains $S$. This is easily done for $k=s$ since the metric is just the indicator function, which returns 0 if two sets are equal and 1 otherwise. With $k \geq s$, we were as of yet unable to create such a metric that would be amenable to boosting to high probability.
>
> **[Re: Prop 3.4]:** Thank you for pointing that out. It should read “with sparsity parameter s, such that n = …”
>
> **[Re: Failure probability and $d^{-10}$]:** This is primarily to show that the failure probability can be $\frac{1}{\mathsf{poly}(d)}$ without affecting the sample complexity or the error by more than a constant multiplicative factor.
>
> **[Re: Section 3.4 and [KLL+23]]:** Although Section 3.4 is reminiscent of a standard median-of-means type analysis, we were unaware of other existing algorithms that extended this framework to vectors, apart from techniques such as the geometric median(also suggested by Jain et al. (2016) for probability boosting), which may or may not return a sparse vector. One advantage of [KLL+23] is that they choose a vector from the given set.
>
> **[Re: Eq(4) and absolute value terms]:** As described on Line 221, the $\pm$ notation is used here to describe the bound on the deviation. That’s why the second term has an absolute value, which is an upper bound on the deviation.
>
> **[Re: Line 232 involving [SSM11]]:** Thank you for pointing this out. Yes, you are indeed correct, this is indeed a condition number since the second eigenvalue is 1 here. We will clarify this in the revised manuscript.
>
> **[Re: General argument for $U=e_i$]:** We use other matrices in place of U, such as U=I_S, in the proof to handle truncation assuming knowledge of the true support. Lemma A.5.1 provides a sketch of which terms come into play and the corresponding values of U which are important. We included that in the appendix in the interest of space and can provide a brief sketch in the extended manuscript after revision.

---

### Official Review · Reviewer_pD6C · 2024-07-18

**Soundness:** 3
**Presentation:** 3
**Contribution:** 3
**Rating:** 7
**Confidence:** 2

**Summary:**

The paper studies Principal Component Analysis with O(d) space and O(nd) time, where n is the number of datapoints and d is their dimensionality.

The authors provide the first single-pass algorithm that under a general \Sigma matrix whose top principal vector is s-sparse, manages to find a close enough vector in the sense of the sinus-squared error. Their main theorem is Th. 1.1. which states that under a structural assumption on the effective rank (ratio of the trace and the principal eigenvalue of the population covariance matrix Σ) not being too large (the function involves the spectral gap and the top eigenvalue).

Their algorithm relies on Oja's algorithm and the authors show that w.h.p. the Oja vector when initialized by a random unit vector, will actually converge to an output whose top k entries in terms of magnitude will include the true support of the s-sparse v_0. Then, the authors can ue the recovered support and achieve minimiza optimal sparse PCA, thus improving upon several prior works.

**Strengths:**

+very interesting and well-motivated problem

+clean framework and clean algorithm

+novel analysis and simple algorithm that improves upon several prior works

**Weaknesses:**

-no serious weaknesses

**Questions:**

Some typos:
-matrix multiplication constant is not 2.732 as written.

**Limitations:**

-

---

> ### Author Rebuttal · Authors · 2024-08-07
>
> Thank you for your kind words regarding the problem statement considered, the simplicity and performance of our algorithm, and the novelty of our analysis. We will correct the matrix multiplication constant to ~2.372 based on recent developments (see [a]).
>
> References:
>
> [a] Williams, Virginia Vassilevska, Yinzhan Xu, Zixuan Xu, and Renfei Zhou. "New bounds for matrix multiplication: from alpha to omega." In Proceedings of the 2024 Annual ACM-SIAM Symposium on Discrete Algorithms (SODA), pp. 3792-3835. Society for Industrial and Applied Mathematics, 2024.

---

> > ### Comment · Reviewer_pD6C · 2024-08-11
> > **post-rebuttal**
> >
> > The reviewer has read the author's response and keeps the score unchanged.

---

### Author Rebuttal · Authors · 2024-08-06

We want to first thank all the reviewers for their valuable suggestions and insightful feedback. We believe we have addressed nearly all of their main technical questions. In what follows, we will address some important points each reviewer has raised. We will correct all the typographical issues pointed out and will not address them here.

### **Re: Over/under estimates of $\eta$ (Reviewer zu7o)**

 For step-sizes, we follow the convention in (Balsubramani et al. (2015), De Sa et al. (2015), Jain et al. (2016), Li et al (2017), Allen-Zhu et al. (2016)), Huang et al. (2021), where Oja’s algorithm is analyzed without sparsity and the optimal learning rate requires knowledge of the gap, $\lambda_{1} - \lambda_{2}$, and other model parameters to get the statistically optimal rate. Our sin-squared error roughly is of the form $O\left(\eta\lambda_1+\exp(-n\eta(\lambda_1-\lambda_2)\right)$. The optimal eta ensures that the first term dominates yielding our optimal error rates. However, a small $\eta$ resulting from plugging in an upper bound on the eigengap $\lambda_1-\lambda_2$ may make the second term dominate leading to a suboptimal sin-squared error. We will clarify this further.

### **Re: $k \geq s$ in Theorem 3.2 (Reviewer zu7o)**

 Our probability boosting argument for Algorithm 3 requires a distance metric, such that when the true support $S\in \hat{S}_1$, $d(S,\hat{S}_1)\leq \epsilon$ for some small $\epsilon>0$. And not only that, we also crucially need that $d(S,\hat{S})\leq \epsilon$ implies that $\hat{S}$ contains $S$. This is easily done for $k=s$ since the metric is just the indicator function which returns 0 if two sets are equal and 1 otherwise. With $k \geq s$, we were as yet unable to create such a metric that would be amenable to boosting to high probability.

### **Re: Top-k principal components (Reviewer jbhG)**

 Recent results provide a black-box way to obtain k-PCA given an algorithm to extract the top eigenvector (see [a]) which could be employed treating our algorithm as a 1-PCA oracle. This has also been proposed in [b]. We also believe that an analysis such as [c] can be extended to the sparse setting to obtain top-k principal components simultaneously via QR decomposition and thresholding. This could be an interesting direction for future work.

### **Re: Novelty of fundamental ideas] (Reviewer qSPb)**

 While our algorithm builds on top of Oja’s algorithm, hard-thresholding of Oja’s vector for Sparse PCA has not been proposed or analyzed before. The closest algorithm analyzes soft-thresholding under a stronger assumption on the covariance model and an initialization close to the population eigenvector (see Wang and Lu, 2016). We would also like to point out that the analysis of thresholding, as pointed out by all other reviewers, is significantly different from analyzing regular Oja’s algorithm. In particular, it introduces a tight analysis of a system of linear recurrences, which is novel, as pointed out by Reviewer zu7o.  Our analysis offers theoretical insights into the behavior of the entries of the Oja vector, which, to the best of our knowledge, has not been attempted before. Please refer to the reviewer rebuttal for a more detailed response.

References:

[a] Jambulapati, A., Kumar, S., Li, J., Pandey, S., Pensia, A. & Tian, K.. (2024). Black-Box k-to-1-PCA Reductions: Theory and Applications. Proceedings of Thirty-Seventh Conference on Learning Theory, in Proceedings of Machine Learning Research 247:2564-2607 Available from https://proceedings.mlr.press/v247/jambulapati24a.html.

[b] Mackey, Lester. "Deflation methods for sparse PCA." Advances in neural information processing systems 21 (2008).

[c] Allen-Zhu, Zeyuan, and Yuanzhi Li. "First efficient convergence for streaming k-pca: a global, gap-free, and near-optimal rate." In 2017 IEEE 58th Annual Symposium on Foundations of Computer Science (FOCS), pp. 487-492. IEEE, 2017.

---

### Comment · Area_Chair_tfpc · 2024-08-07
**rebuttal**

Dear authors (and reviewers),

Many of the reviewers were positive, but reviewer qSPb thought there were not enough new ideas.  At the end of the day, we don't just average reviewer scores, so this paper is not a guaranteed "accept".  Authors, I'd suggest you write a rebuttal to reviewer qSPb.

For all the reviewers, please take a look at each others' reviews and see if that changes your mind.

As a reminder, the OpenReview framework allows us to have a dialogue, rather than requiring infrequent formal comments. So let's please use it! Just make sure to select the correct visibility for comments (either just reviewers, or reviewers and authors) as appropriate.

Best regards,

Area Chair

---

### Decision · Program_Chairs · 2024-09-25

**Decision:**

Accept (poster)

**Comment:**

All four reviewers were positive, though one of them barely so who was a little concerned that the algorithm was straightforward and the contribution is fairly theoretical (and for a fairly narrow task).  Taking that into account, it still appears to be a solid contribution, and certainly theoretical analyses are welcome in NeurIPS. The theoretical contribution itself is clearly described, and at least one reviewer was impressed by the techniques.  Overall, the reviewers thought it was a good paper with no major flaws, and thus I'm happy to recommend acceptance.